# AN INFORMATION THEORETIC EVALUATION METRIC FOR STRONG UNLEARNING

## ABSTRACT

Machine unlearning (MU) aims to remove the influence of specific data from trained models, addressing privacy concerns and ensuring compliance with regulations such as the "right to be forgotten." Evaluating strong unlearning, where the unlearned model is indistinguishable from one retrained without the forgetting data, remains a significant challenge in deep neural networks (DNNs). Common black-box metrics, such as variants of membership inference attacks and accuracy comparisons, primarily assess model outputs but often fail to capture residual information in intermediate layers. To bridge this gap, we introduce the Information Difference Index (IDI), a novel white-box metric inspired by information theory. IDI quantifies retained information in intermediate features by measuring mutual information between those features and the labels to be forgotten, offering a more comprehensive assessment of unlearning efficacy. Our experiments demonstrate that IDI effectively measures the degree of unlearning across various datasets and architectures, providing a reliable tool for evaluating strong unlearning in DNNs.

## 1 INTRODUCTION

Machine unlearning (MU) seeks to remove the impact of specific data samples from a trained model, addressing privacy issues such as "right to be forgotten" (Cao and Yang, 2015; Voigt and Von dem Bussche, 2017). In addition to privacy, MU is also emerging as a tool to eliminate the influence of corrupted or outdated data used during training (Nguyen et al., 2022; Kurmanji et al., 2023). The most straightforward approach to MU is *exact unlearning*, where the model is retrained from scratch, excluding the data that need to be forgotten. Although this method ensures complete data removal, it is computationally expensive and not scalable (Aldaghri et al., 2021; Bourtoule et al., 2021; Yan et al., 2022). Consequently, research has shifted towards *approximate unlearning*, which aims to replicate the effects of retraining in a more efficient manner.

The goal of MU is to create an unlearned model that is indistinguishable from a model retrained from scratch, referred to as strong unlearning. This objective has become particularly crucial with the rise of open-source models like Stable Diffusion (Rombach et al., 2022) and LLaMA (Touvron et al., 2023), which are widely used and fine-tuned by various users. For unlearning algorithms to be practically useful, they must be capable of fully eliminating traces of private data and preventing potential exploitation. While $(\epsilon, \delta)$-certified unlearning methods (Zhang et al., 2024b; Mu and Klabjan, 2024) provide theoretical guarantees, they are often impractical for large-scale models. As a result, most approximate unlearning methods rely on heuristic approaches, lacking formal guarantees. Thus, these methods must undergo empirical evaluation to demonstrate their effectiveness.

However, current evaluations, primarily based on black-box approaches such as membership inference attacks (MIA) (Shokri et al., 2017; Carlini et al., 2022) and accuracy comparisons, focus on output similarity rather than internal model changes. Although these metrics may capture weak unlearning (Fan et al., 2024; Jia et al., 2023; Chundawat et al., 2023b; Foster et al., 2024; Chen et al., 2023), they may not be sufficient for assessing strong unlearning. In this work, we investigate whether relying solely on outputs can truly reflect complete influence removal, considering that outputs can be superficially adjusted without impacting internal representations (Kirichenko et al., 2023).

Surprisingly, our experiments reveal that even minimal changes to the model, such as modifying only the final layer while preserving all information in the intermediate layers, can still satisfy the black-box evaluation metrics, exposing their limitations in assessing strong unlearning. This finding

also raises critical concerns about whether current MU methods genuinely achieve information removal comparable to retraining from scratch, despite yielding similar model outputs.

Consequently, motivated by the Information Bottleneck principle (Tishby et al., 2000; Tishby and Zaslavsky, 2015), we introduce the **information difference index (IDI)**, a novel white-box metric designed to quantify residual information in intermediate layers after unlearning. IDI measures the mutual information (Shannon, 1948) between intermediate features and the forgetting labels, providing an interpretable value to assess the effectiveness of unlearning algorithms. We observe that IDI remains robust despite the inherent stochasticity of the unlearning process, and is model-agnostic, ensuring adaptability to various architectures. Additionally, estimating IDI with a data subset yields reliable results, enhancing its practicality. To our knowledge, IDI is the first robust white-box metric designed to evaluate unlearning quality, addressing a crucial yet underexplored aspect of the field.

Through the application of IDI, we find that many recent MU methods, despite their strong performance on black-box metrics, still retain significant information about the forgetting data within intermediate layers. Building on the insights gained from IDI, we introduce **COLapse-and-Align (COLA)**, a simple method that first collapses feature representations to be forgotten and then re-aligns retain features to address residual information in unlearning processes. Despite its simplicity, COLA serves as a useful benchmark, demonstrating notable improvements in IDI scores compared to other methods on datasets such as CIFAR-10, CIFAR-100, and ImageNet-1K, as well as architectures like ResNet-18, ResNet-50, and ViT. Notably, COLA achieves this without access to the full training dataset, unlike several existing methods. The ability of IDI to capture COLA's impact on intermediate features further underscores its value as a robust efficacy metric.

We summarize our contributions as follows: First, we identify the limitations of existing black-box metrics, which overlook residual information in intermediate layers. Second, we introduce the information difference index (IDI), an interpretable white-box metric that quantifies mutual information between intermediate features and labels. Third, we validate the robustness of IDI through extensive experiments on diverse datasets and model architectures. Finally, using the COLapse-and-Align (COLA) method as a baseline, we show that IDI effectively captures residual information in intermediate features, proving its value as a reliable metric for unlearning quality.

## 2 PROBLEM STATEMENT AND PRELIMINARIES

### 2.1 PROBLEM STATEMENT

Let $D = \{(x_i, y_i)\}_{i=1}^N$ denote a training dataset comprising $N$ image-label pairs $(x_i, y_i)$. In a supervised learning setup, $D$ is partitioned into two subsets: the *forget set* $D_f$, containing the data points to be removed, and the *retain set* $D_r = D \setminus D_f$, containing the data points to be preserved. The initial model $\theta_o$, referred to as the **Original model**, is trained on the full dataset $D$ using empirical risk minimization. The **Retrain model** $\theta_r$ is trained from scratch on only the retain set $D_r$. The **unlearned model** $\theta_u$ is obtained by applying a machine unlearning (MU) algorithm to the Original model $\theta_o$, aiming to remove the influence of $D_f$. The goal of MU is for $\theta_u$ to closely approximate $\theta_r$, ensuring the unlearned model behaves as though $D_f$ had never been used in training, while preserving the training methodology across $\theta_o$, $\theta_r$, and $\theta_u$.

MU is often studied in the context of image classification (Shaik et al., 2023; Nguyen et al., 2022), where it is typically classified into two scenarios based on the nature of the forget set: *class-wise forgetting*, where all samples from a specific class are targeted, and *random data forgetting*, where samples are selected indiscriminately across all classes.

Throughout the paper, within a given model $\theta$, we define the **head** as the last few layers responsible for classification; typically one to three linear layers. The **encoder**, on the contrary, encompasses the remainder of the network, which usually consists of convolutional layers or transformer encoders.

### 2.2 PRELIMINARIES

**Machine Unlearning (MU).** Exact unlearning, which involves creating Retrain, guarantees the information removal from the forget set but is computationally expensive (Bourtoule et al., 2021; Yan et al., 2022; Aldaghri et al., 2021; Brophy and Lowd, 2021). To address this, approximate

unlearning methods have been developed, focusing on efficiency rather than strict theoretical guarantees. Specifically, strong unlearning, where the unlearned model is indistinguishable from Retrain, has been explored through the application of differential privacy (DP) (Dwork and Roth, 2014) inspired techniques, which aim to achieve parameter-level indistinguishability (Dwork and Roth, 2014; Ginart et al., 2019; Neel et al., 2021; Sekhari et al., 2021; Ullah et al., 2021; Guo et al., 2020). However, applying such techniques to deep neural networks (DNNs) remains challenging due to their vast number of parameters and non-convex loss landscapes (Qiao et al., 2024). As a result, recent studies typically assess the similarity of model outputs (*i.e.*, predictions), using weak unlearning as a practical proxy for strong unlearning (Xu et al., 2023). While empirically ensuring strong unlearning is challenging, it is critical for deploying unlearning algorithms to meet legal requirements like GDPR (Voigt and Von dem Bussche, 2017), the "right to be forgotten", and prevent retention of sensitive data, particularly with the growing use of open-source models like CLIP (Radford et al., 2021), Stable-Diffusion (Rombach et al., 2022), and LLaMA (Touvron et al., 2023), where data could unintentionally persist and be exploited. Our work focuses on developing a robust empirical metric to evaluate unlearning algorithms, distinct from verification (Zhang et al., 2024a; Sommer et al., 2022), which focuses on real-world attack scenarios to validate the effectiveness of unlearning.

**Evaluation Criteria in MU.** As the goal of MU is to remove the influence of specific data while preserving the others, the unlearning algorithms are typically evaluated on three criteria: *Efficacy*, *Accuracy*, and *Efficiency* (Hayes et al., 2024). Efficacy measures how closely the unlearned model approximates Retrain, which is key to unlearning quality. Accuracy ensures task performance remains intact after unlearning, while efficiency ensures the unlearning process is faster than retraining. Accuracy and efficiency can be easily evaluated using existing metrics. Accuracy consists of three categories: *unlearning accuracy (UA)*, *remaining accuracy (RA)*, and *testing accuracy (TA)*. UA measures performance on $\mathcal{D}_f$ as $UA(\theta_{\mathbf{u}}) = 1 - \text{Acc}_{\mathcal{D}_f}(\theta_{\mathbf{u}})$, RA on $\mathcal{D}_r$ as $RA(\theta_{\mathbf{u}}) = \text{Acc}_{\mathcal{D}_r}(\theta_{\mathbf{u}})$, and TA measures generalization to unseen data as $TA(\theta_{\mathbf{u}}) = \text{Acc}_{\mathcal{D}_{test}}(\theta_{\mathbf{u}})$. Performance levels comparable to Retrain across these metrics indicate better unlearning. To simplify comparisons, Cotogni et al. (2023) proposed AUS, which combines UA and TA into a single accuracy measure. In terms of efficiency, *runtime efficiency (RTE)* measures the time an algorithm takes to complete unlearning, with lower RTE indicating more efficient unlearning (Fan et al., 2024; Jia et al., 2023). However, assessing unlearning efficacy, or determining whether the unlearned model has fully removed the influence of specific data to the same extent as Retrain, remains a significant challenge in complex DNNs. The efficacy metrics are divided into two categories: *black-box* metrics, which focus solely on model outputs (*i.e.*, predictions), and *white-box* metrics, which examine internal dynamics such as parameters, gradients, and features. While black-box metrics are typically used due to their convenience, no universally accepted standard exists, leaving room for more reliable assessment.

**Black-box Efficacy Metrics.** Variants of membership inference attacks (MIA) (Shokri et al., 2017; Carlini et al., 2022) are the most widely used black-box metrics (Shen et al., 2024; Kim et al., 2024; Fan et al., 2024; Jia et al., 2023; Foster et al., 2024). MIA determines whether specific data were part of the training set by training an auxiliary classifier, with attack success rates on the forget set that are close to those of Retrain being preferred. It is worth noting that recent works often use a combination of UA, RA, TA, MIA and RTE as metrics for evaluating unlearning performance across the three criteria (Chen et al., 2023; Kim et al., 2024; Jia et al., 2023; Fan et al., 2024), collectively referred to as the 'full-stack' evaluation scheme (Jia et al., 2023; Fan et al., 2024). Other metrics, such as Jensen-Shannon divergence (JSD), and ZRF (Chundawat et al., 2023b; Poppi et al., 2024) compare the output logits between the unlearned model and Retrain (or a random model for ZRF). Additionally, time-based metrics like Anamnesis Index (AIN) (Chundawat et al., 2023a; Tarun et al., 2023a) and relearn time (RT) (Tarun et al., 2023b; Golatkar et al., 2020a;b; 2021) measure the time (or epochs) required for the unlearned model to regain performance on the forget set. Black-box metrics, though convenient, overlook internal behaviors and cannot verify strong unlearning by ensuring forgetting data's influence is fully removed. Section 3 highlights their limitations.

**White-box Efficacy Metrics.** In contrast, white-box metrics offer a more detailed evaluation by analyzing internal model dynamics to track residual influence. Previous studies have measured parameter-wise distances (e.g., $\ell_2$-distance, KL-divergence) between the unlearned model and Retrain (Golatkar et al., 2020a; Wu et al., 2020). However, this approach is computationally expensive and unreliable due to the inherent randomness in DNN training (Hayes et al., 2024; Goel et al., 2022).

Becker and Liebig (2022) proposed a Fisher information based metric, but their experimental results were inconsistent with theoretical intuition. Graves et al. (2021) applied model inversion attacks to reconstruct images from the forget set, but this method relies on qualitative comparisons, making it difficult to compare across different algorithms. Although robust white-box metrics are currently lacking and challenging to develop, they are crucial for validating approximate methods, which often lack guarantees of complete information removal. Without such metrics, these methods cannot be trusted in privacy-sensitive applications that demand a high level of confidence in information removal. To address this critical need, our work proposes a reliable and practical white-box metric.

## 3 RETHINKING THE EVALUATION OF UNLEARNING EFFICACY

### 3.1 HEAD DISTILLATION: SIMPLE TECHNIQUE CHALLENGES BLACK-BOX METRICS

In this section, we reveal the limitations of commonly used black-box efficacy metrics by applying our simple unlearning technique to a single-class forgetting task. We reveal how these metrics can misrepresent unlearning efficacy by overlooking residual information in intermediate features, even when the model's output appears similar to that of Retrain.

Drawing inspiration from the teacher-student framework (Chundawat et al., 2023b; Kurmanji et al., 2023), our strategy, termed **head distillation (HD)**, employs logit distillation from Original $\theta_{\mathbf{o}}$. Specifically, the unlearned model $\theta_{\mathbf{u}}$ is initialized from $\theta_{\mathbf{o}}$ with the encoder frozen and only the head remaining trainable. During the unlearning process, the head is finetuned on training dataset $\mathcal{D}$ using KL-divergence loss (Hinton et al., 2014) to follow the masked output from $\theta_{\mathbf{o}}$, where the logit for the forgetting class is set to negative infinity while preserving the logits for the remaining classes, as shown in Figure 1. This approach enables $\theta_{\mathbf{u}}$ to mimic a pseudo-retrained model, as the masked logits closely resemble those of Retrain. By aligning the output behavior with that of Retrain, HD aims to simulate the desired unlearning effect.

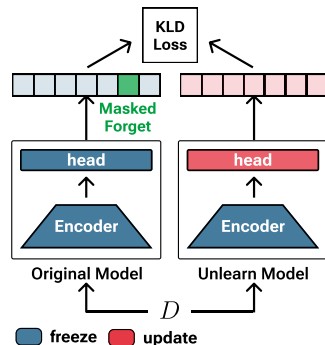

Figure 1: Overview of head distillation (HD): Original distills knowledge into the unlearn model's head by masking the forgetting class logit, with the encoder kept frozen.

We conducted experiments using the CIFAR-10 (Krizhevsky, 2009) dataset and the ResNet-18 architecture (He et al., 2016), where the head consists only of a single linear layer. To evaluate *efficacy*, we used a widely adopted black-box metric, membership inference attack (MIA) and Jensen-Shannon divergence (JSD). Additionally, we measured unlearning accuracy (UA), testing accuracy (TA) and run-time efficiency (RTE) for *accuracy* and *effiency*. For more details on these metrics, please refer to Appendix C.1. We compared HD to recent approximate MU methods, including FT, RL (Golatkar et al., 2020a), GA (Thudi et al., 2022), $\ell_1$-sparse (Jia et al., 2023), and SALUN (Fan et al., 2024). Details on the baselines can be found in Appendix C.2.

Figure 2 presents the experimental results. Despite its simplicity, HD demonstrates remarkable performance across black-box efficacy metrics, outperforming all other methods in MIA and ranking second in JSD. HD achieves this performance in just 6.2 seconds, approximately 30 to 60 times faster than competing methods. Additionally, HD maintains comparable testing accuracy (TA), effectively preserving task performance. All methods achieved perfect unlearning accuracy (100% UA), which is omitted from Figure 2. Notably, HD's strong performance extends to other unlearning scenarios, including multi-class and random data forgetting, as detailed in Appendix D.

The experimental results indicate that HD performs exceptionally well across all black-box evaluation metrics. However, its validity as an effective MU algorithm warrants closer examination. The primary issue is that HD closely resembles Original $\theta_{\mathbf{o}}$, with changes limited to the single-layer head, while the encoder remains identical to $\theta_{\mathbf{o}}$. Consequently, all intermediate features related to the forget set are retained. This raises a critical question:

> *Do black-box metrics truly capture the unlearning quality,*
> *or are they misled by superficial changes while deeper information persists?*

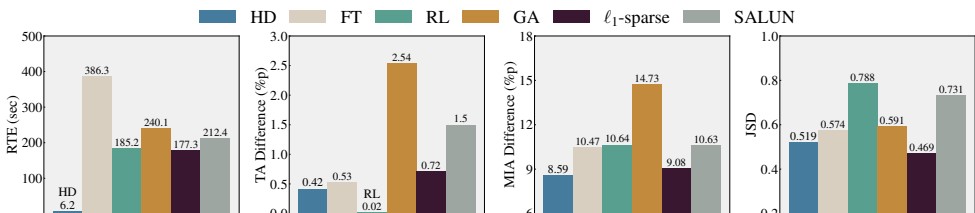

Figure 2: Performance of six methods (HD, FT, RL, GA, $\ell_1$-sparse, SALUN) on (CIFAR-10, ResNet-18), evaluated in efficiency (RTE), accuracy (TA), and efficacy (MIA, JSD). For TA, MIA, and JSD, lower differences from Retrain are preferred, indicating closer similarity to Retrain.

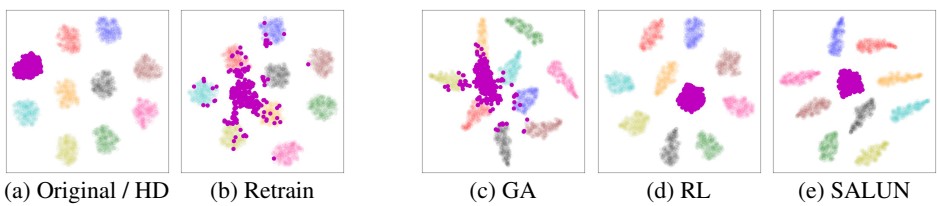

| (a) Original / HD | (b) Retrain | (c) GA | (d) RL | (e) SALUN |

Figure 3: t-SNE visualizations of encoder outputs for Original, Retrain, and unlearned models from three MU methods (GA, RL, SALUN) on single-class forgetting with (CIFAR-10, ResNet-18). In each t-SNE plot, features of the forgetting class are represented in purple. Original and HD have identical feature distribution as they share the same encoder.

## 3.2 RESIDUAL INFORMATION OF FORGETTING DATA: LIMITATIONS OF BLACK-BOX ASSESSMENTS FOR UNLEARNING EFFICACY

To address the above question, we conducted two analyses on recent unlearning methods to determine whether they internally remove information from the forget set, despite their strong performance on black-box efficacy metrics. We note that both analyses are performed on the unlearned models using the same experimental setting discussed in Section 3.1.

We start with a qualitative analysis using t-SNE (van der Maaten and Hinton, 2008) visualizations of intermediate features from model encoders to investigate how Retrain differs from Original and to analyze the internal behavior of different unlearning algorithms, as shown in Figure 3. Figure 3b reveals the scattered distribution of the features corresponding to the forgetting class (represented in purple) in Retrain. These features are dispersed across multiple clusters, indicating the model's difficulty in extracting coherent information from the forgetting class. This scattering reflects an ideal outcome of strong unlearning, suggesting that the unlearned model has successfully 'forgotten' how to represent meaningful semantic information from the forget set.

Notably, while the features from GA (Thudi et al., 2022) appear scattered in a manner similar to Retrain, the features from RL (Golatkar et al., 2020a) and SALUN (Fan et al., 2024) closely resemble those of Original. As expected, HD, which shares the same encoder as Original, produces t-SNE results identical to it. These findings indicate that several unlearned models still retain a significant capacity to recognize the forgetting class, distinguishing them from Retrain.

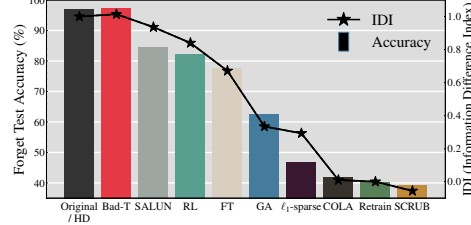

Figure 4: Forget test accuracy and IDI (our metric in Section 4.3) for Original, Retrain, and MU methods (including COLA, our method in Section 5.1) after head retraining with fixed unlearned encoders using 2% of $\mathcal{D}$ in (CIFAR-10, ResNet-18). IDI aligns with the recovered accuracy across models.

To further examine the residual influence in unlearned models, we conducted a follow-up experiment inspired by time-based metrics (*e.g.*, (Chundawat et al., 2023a)). This experiment explores whether unlearned encoders can recover forgotten information with minimal data. Specifically, we replaced the heads of all unlearned models, including Retrain and Original, with randomly initialized ones. The encoders were then frozen, and new heads were trained using $\mathcal{D}'$, a small subset (only 2% of the total) of $\mathcal{D}$, comprising randomly selected samples.

After training, we evaluated the accuracy of the new models on the forget test data. Surprisingly, as shown in Figure 4, while the retrained head of Retrain achieves no more than 41% accuracy, the heads from certain methods, like Bad-T, SALUN, and RL exhibit over 82% accuracy. Notably, the high accuracy observed in SALUN and RL corresponds to the clustered t-SNE plots in Figure 3.

The results from the above analyses demonstrate that unlearned models across various MU algorithms retain substantial residual influence from the forget set internally, unlike Retrain. This highlights incomplete unlearning in those approximate methods. However, a critical concern is that commonly used black-box assessments fail to detect these underlying residuals. If unlearning efficacy metrics cannot ensure strong unlearning, as clearly shown in our results, the reliability of approximate unlearning algorithms, which often lack theoretical guarantees, becomes questionable in real-world applications. Therefore, developing practical white-box approaches that consider internal model behaviors is essential to achieving the fundamental goal of unlearning.

## 4 AN INFORMATION THEORETIC METRIC FOR UNLEARNING EFFICACY USING INTERMEDIATE FEATURES

Current MU efficacy evaluations, which rely primarily on black-box metrics, overlook residual information in intermediate layers, as shown in Section 3. To address this, we measure residual information in the intermediate features of unlearned models using mutual information. Building on this, we propose a novel white-box metric, IDI, that goes beyond output-based evaluations.

### 4.1 QUANTIFYING RESIDUAL INFORMATION WITH MUTUAL INFORMATION

To quantify the relationship between high dimensional intermediate features and data labels, we utilize Shannon's mutual information (MI), a robust measure that effectively captures variable dependencies across various dimensional complexities. For an input $\mathbf{X}$, let $\mathbf{Z}_\ell^{(\mathbf{u})}$ and $\mathbf{Z}_\ell^{(\mathbf{r})}$ denote the features from the $\ell$-th layer of the total $L$-layer encoder in the unlearned model and Retrain, respectively. Let $Y$ be a binary label, where $Y = 1$ indicates that the input $\mathbf{X}$ belongs to the forget set, and $Y = 0$ otherwise. We measure the MI, denoted as $I(\mathbf{Z}_\ell; Y)$, across each layer from 1 to $L$, to determine whether intermediate features retain information about the forget set. To estimate MI, we use the InfoNCE loss (Oord et al., 2018). InfoNCE is widely used in MI estimation of DNNs and shown to be robust and effective (Radford et al., 2021; Jia et al., 2021).

Given a batch $\mathcal{B} = \{(U^{(k)}, V^{(k)}) : 1 \leq k \leq K\}$, sampled from a joint distribution $P_{U,V}$, where $U \in \mathcal{U}$ and $V \in \mathcal{V}$ be random variables. The InfoNCE loss (Poole et al., 2019) is defined as:

$$\mathcal{L}_{\text{NCE}}(\mathcal{B}, \nu, \eta) = \frac{1}{K} \sum_{k=1}^{K} \log \frac{\exp(f_\nu(U^{(k)})^\top g_\eta(V^{(k)}))}{\frac{1}{K} \sum_{k'=1}^{K} \exp(f_\nu(U^{(k)})^\top g_\eta(V^{(k')}))},$$

where $f_\nu : \mathcal{U} \to \mathbb{R}^d$ and $g_\eta : \mathcal{V} \to \mathbb{R}^d$ are critic functions, with an output embedding dimension $d$, parameterized by neural networks with parameters $\nu$ and $\eta$. This neural network parameterization, inspired by Radford et al. (2021), effectively captures complex relationships in contrastive learning through flexible and expressive modeling of the joint distributions of $U$ and $V$.

The InfoNCE loss serves as a lower bound on the MI between $U$ and $V$. In fact, the maximum value of the InfoNCE loss, when using the joint critic functions, equals the mutual information:

$$I(U; V) = \max_{\nu, \eta} \mathcal{L}_{\text{NCE}}(\mathcal{B}, \nu, \eta).$$

Thus, to estimate the mutual information, we maximize the InfoNCE loss over the parameters $\nu$ and $\eta$. By leveraging the flexibility of neural networks, we can effectively capture the underlying structure of the data and accurately quantify the amount of shared information between the variables $U$ and $V$.

To estimate mutual information (MI) for each layer in the network, we design separate critic functions for every layer, denoted as $f_{\nu_\ell}$ and $g_{\eta_\ell}$, where $\ell$ denotes the layer index from 1 to $L$, the total number of layers in the encoder. The critic $g_{\eta_\ell}$ handles the binary variable $Y$, which is parameterized as two trainable $d$-dimensional vectors, $g_{\eta_\ell}(0)$ and $g_{\eta_\ell}(1)$, and selects the appropriate vector based on the value of $Y$. In contrast, $f_{\nu_\ell}$ maps the intermediate features $\mathbf{Z}_\ell$, from the $\ell$-th layer of the encoder, to a $d$-dimensional embedding space. The parameters $\nu_\ell$ define the weights and biases of this neural

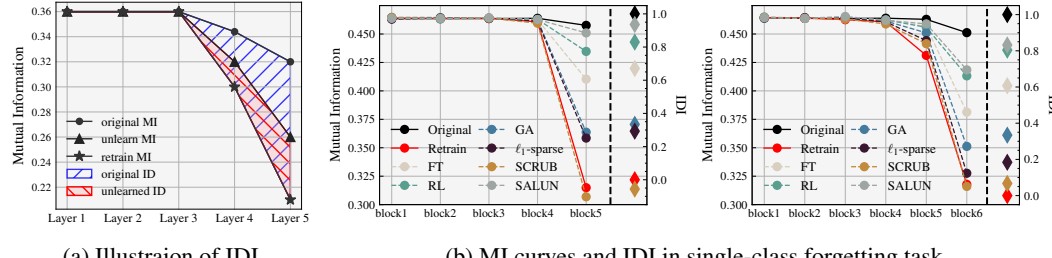

(a) Illustraion of IDI          (b) MI curves and IDI in single-class forgetting task

Figure 5: (a) Conceptual illustration of IDI. Curves show estimated mutual information $I(\mathbf{Z}_\ell; Y)$ for Original ($\bullet$), unlearned ($\blacktriangle$), and Retrain ($\star$). IDI is the ratio $\frac{\mathbf{ID}(\theta_\mathbf{u})}{\mathbf{ID}(\theta_\mathbf{o})}$, corresponding to the red area divided by the blue area. (b) MI curves and IDI values for Original, Retrain, and unlearned models from six methods (FT, RL, GA, $\ell_1$-sparse, SCRUB, SALUN) on CIFAR-10 across ResNet-18 (left) and ResNet-50 (right) blocks, averaged over five trials. See Appendix D.2 for standard deviations.

network. The complexity of $f_{\nu_\ell}$ varies depending on the layer: in earlier layers, $f_{\nu_\ell}$ processes raw, less interpretable features, requiring a more intricate design to effectively capture the relationship between $\mathbf{Z}_\ell$ and $Y$. In later layers, with more refined features, $f_{\nu_\ell}$ can perform the mapping more directly. This design enables the accurate estimation of $I(\mathbf{Z}_\ell; Y)$, capturing the dependency between features and labels at different depths. For details on $f_{\nu_\ell}$ and $g_{\eta_\ell}$, refer to Appendix B.

For $f_{\nu_\ell}$, we propose a model-agnostic approach that reuses the network layers from $\ell + 1$ to $L$, allowing us to approximate the mutual information between the output and intermediate features at layer $\ell$ without requiring network redesign for each layer, thus maintaining flexibility and scalability. To ensure dimensional compatibility between $f$ and $g$, we introduce an additional linear projection layer so that $f_{\nu_\ell}(\mathbf{Z}_\ell)$ outputs a $d$-dimensional feature.

During the optimization of the InfoNCE loss, we freeze the parameters of the model up to the $\ell$-th layer and reuse the architecture of the remaining layers, starting from $\ell + 1$, as $f_{\nu_\ell}$. These remaining layers, along with the projection layer, are randomly initialized and trained from scratch to specifically optimize the InfoNCE loss. We utilize both the retain set and the forget set to have representations for $Y = 0$ and $Y = 1$, ensuring that the model captures information relevant to both outcomes.

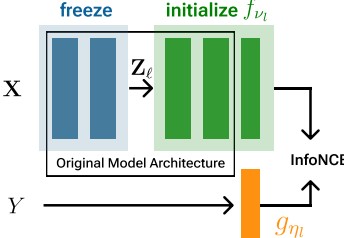

Figure 6: Illustration of estimating MI using InfoNCE. The critic function $f_{\nu_\ell}$ represents a trainable network to capture features from $\mathbf{Z}_\ell$, while the critic function $g_{\eta_\ell}$ handles the binary input $Y$.

This approach allows $f_{\nu_\ell}$ to effectively leverage intermediate features $\mathbf{Z}_\ell$ to classify $Y$, providing deeper insights into the model's internal information processing at each layer. It also reveals the model's capacity to extract and utilize relevant information for distinguishing between output labels, offering a clearer understanding of the information dynamics across the network.

## 4.2 RESIDUAL INFORMATION IN UNLEARNED MODELS

We begin by plotting the estimated MI between the intermediate layers and the binary label indicating whether the data belong to the forget set, as shown in Figure 5b. As expected, MI decreases across layers, aligning with the Information Bottleneck principle (Tishby et al., 2000). This figure reveals the internal behaviors of unlearned models that black-box assessments fail to capture.

In particular, SCRUB and $\ell_1$-sparse, which approximate the MI levels of Retrain, are more likely to achieve the MU objective at the feature level across both ResNet architectures. Their lower MI suggests that their encoders, like Retrain, struggle to differentiate between the forget set and the retain set. Conversely, SALUN and RL show MI curves that are close to that of Original, indicating the opposite. Note that HD produces the identical curve as Original, as its encoder remains unchanged. We observe similar patterns in CIFAR-100 and ImageNet-1K, as well as in ViT. Additionally, extending our experiment to multi-class forgetting tasks (*e.g.*, 20 classes on CIFAR-100) reveals more pronounced MI differences between Retrain and Original. See Appendix E.1 for further results.

### 4.3 INFORMATION DIFFERENCE INDEX (IDI)

Motivated from the above experiment, we define the **information difference (ID)** of $\theta_{\mathbf{u}}$ as the MI difference across intermediate layers between the unlearned model and Retrain, calculated as:

$$\mathbf{ID}(\theta_{\mathbf{u}}) = \sum_{\ell=1}^{L}\big(I(\mathbf{Z}_{\ell}^{(\mathbf{u})}; Y) - I(\mathbf{Z}_{\ell}^{(\mathbf{r})}; Y)\big). \tag{1}$$

ID of $\theta_{\mathbf{u}}$ shows the extent of information retention through ensuing layers of the unlearned encoder. To provide a normalized measure, we introduce the **information difference index (IDI)**:

$$\mathbf{IDI}(\theta_{\mathbf{u}}) = \frac{\mathbf{ID}(\theta_{\mathbf{u}})}{\mathbf{ID}(\theta_{\mathbf{o}})} = \frac{\sum_{\ell=1}^{L}\big(I(\mathbf{Z}_{\ell}^{(\mathbf{u})}; Y) - I(\mathbf{Z}_{\ell}^{(\mathbf{r})}; Y)\big)}{\sum_{\ell=1}^{L}\big(I(\mathbf{Z}_{\ell}^{(\mathbf{o})}; Y) - I(\mathbf{Z}_{\ell}^{(\mathbf{r})}; Y)\big)}, \tag{2}$$

where $\mathbf{Z}_{\ell}^{(\mathbf{o})}$ is the output of the $\ell$-th layer of Original encoder. Figure 5a illustrates IDI, which is conceptually the ratio of the areas between MI curves. However, computing MI for all $L$ layers can be computationally expensive. In practice, we compute MI from the last $n$ selected layers (*i.e.*, the last layers of later blocks), where $n \ll L$, as MI in earlier layers of Retrain and Original show negligible differences, as shown in Figure 5b. Further details are provided in Appendix E.2.

IDI quantifies the information gap between the unlearned model and Retrain. An IDI of 0 denotes that the unlearned model has completely removed all information related to the forget set, achieving indistinguishability from Retrain. In contrast, an IDI of 1 indicates that the encoder retains all the information found in Original. Interestingly, a negative IDI value, termed *over-unlearning*, occurs when the model removes more information than Retrain. While we have demonstrated IDI in the context of class-wise forgetting, its application to random data forgetting is provided in Appendix A.1. We note that as the denominator of IDI ($\mathbf{ID}(\theta_o)$) approaches zero, IDI may yield unexpected values. However, this case indicates that Original and Retrain are nearly identical, suggesting minimal unlearning utility. Thus, $\mathbf{ID}(\theta_o)$ can serve as an indicator for the necessity of unlearning in practice.

## 5 EXPERIMENTS

### 5.1 PROPOSED BASELINE: COLLAPSE AND ALIGN (COLA)

As discussed in both Section 3 and 4.2, we observed residual information in the intermediate layers of several unlearned models, despite their outputs being similar to those of Retrain. To address this, we propose a robust two-step unlearning framework, **COLlapse and Align (COLA)**, consisting of a *collapse phase* and an *alignment phase* to directly remove residual information at the feature level.

During the *collapse phase*, COLA eliminates feature-level information by applying supervised contrastive loss (Khosla et al., 2020) to encoder outputs. Rather than dispersing features from the forget set, which could harm model performance, COLA applies the loss to the retain set, promoting tight intra-class clustering. As these clusters shrink, features from the forget set are forced to collapse into the clusters of the retain set, achieving catastrophic forgetting. After feature collapsing, the *alignment phase* optimizes the entire model using cross-entropy loss on the retain set to align the encoder and head. For an illustration of COLA, as well as COLA+, a method tailored for random data forgetting, and detailed loss formulations, refer to Appendix C.6.

### 5.2 COMPREHENSIVE EVALUATION OF UNLEARNING METHODS WITH COLA AND IDI

We demonstrate the utility of IDI as a valuable efficacy metric and highlight the strong performance of COLA and its variant COLA+ through extensive experiments. Our experiments cover three datasets: CIFAR-10, CIFAR-100 (Krizhevsky, 2009), and ImageNet-1K (Deng et al., 2009), and three model architectures: ResNet-18, ResNet-50 (He et al., 2016), and ViT (Dosovitskiy et al., 2021). For simplicity, we approximate IDI using the features from blocks rather than every layer in ResNet and ViT (see Appendix C.7). Please refer to Appendix C for further experimental details.

Table 1 shows the experimental results on CIFAR-10 and ImageNet-1K in class-forgetting tasks. At first glance, excluding the IDI column, several methods show similar accuracy (UA, RA, TA) but greater deviations in efficacy (MIA) and efficiency (RTE). This suggests that previous unlearning

Table 1: Performance summary of MU methods (including COLA and 14 other baselines) for class-wise forgetting task on (CIFAR-10, ResNet-18) and (ImageNet-1K, ResNet-50). The results are shown as $a \pm b$, with $a$ being the mean and $b$ the standard deviation of five independent trials. A better performance of an MU method corresponds to a smaller performance gap with Retrain (except RTE), with the top method in **bold** and the second best underlined.

| | CIFAR-10 (single class) | | | | | | ImageNet-1K (five classes) | | | | | |
|---|---|---|---|---|---|---|---|---|---|---|---|---|
| Methods | UA | RA | TA | MIA | IDI | RTE (min) | UA | RA | TA | MIA | IDI | RTE (min) |
| Retrain | 100.0 | 100.0 | 95.64 | 10.64 | 0.0 | 154.56 | 100.0 | 88.80 | 75.88 | 9.41 | 0.0 | 2661.90 |
| HD | $\mathbf{100.0}_{\pm0.0}$ | $\mathbf{100.0}_{\pm0.0}$ | $95.22_{\pm0.07}$ | $2.05_{\pm0.11}$ | $1.000_{\pm0.0}$ | $\mathbf{0.10}_{\pm0.01}$ | $\mathbf{100.0}_{\pm0.0}$ | $87.94_{\pm0.16}$ | $\underline{75.60}_{\pm0.07}$ | $7.12_{\pm0.12}$ | $1.000_{\pm0.0}$ | $\mathbf{4.75}_{\pm0.03}$ |
| FT | $\mathbf{100.0}_{\pm0.0}$ | $\mathbf{100.0}_{\pm0.0}$ | $95.12_{\pm0.09}$ | $0.17_{\pm0.05}$ | $0.671_{\pm0.008}$ | $6.44_{\pm0.07}$ | $\mathbf{100.0}_{\pm0.0}$ | $\mathbf{88.52}_{\pm0.0}$ | $\mathbf{76.16}_{\pm0.01}$ | $8.24_{\pm1.23}$ | $0.102_{\pm0.026}$ | $140.04_{\pm1.42}$ |
| RL | $99.93_{\pm0.01}$ | $\mathbf{100.0}_{\pm0.0}$ | $\mathbf{95.66}_{\pm0.05}$ | $0.0_{\pm0.0}$ | $0.830_{\pm0.005}$ | $3.09_{\pm0.03}$ | $\underline{99.96}_{\pm0.03}$ | $86.46_{\pm0.07}$ | $75.23_{\pm0.01}$ | $0.23_{\pm0.01}$ | $1.002_{\pm0.007}$ | $200.73_{\pm1.87}$ |
| GA | $\mathbf{100.0}_{\pm0.0}$ | $99.06_{\pm0.25}$ | $93.10_{\pm0.50}$ | $25.37_{\pm3.24}$ | $0.334_{\pm0.014}$ | $4.00_{\pm0.08}$ | $\mathbf{100.0}_{\pm0.0}$ | $80.77_{\pm0.22}$ | $71.49_{\pm0.10}$ | $4.20_{\pm0.46}$ | $0.328_{\pm0.023}$ | $212.14_{\pm2.61}$ |
| Bad-T | $99.90_{\pm0.14}$ | $\underline{99.99}_{\pm0.0}$ | $94.99_{\pm0.12}$ | $68.17_{\pm42.80}$ | $1.014_{\pm0.004}$ | $4.64_{\pm0.05}$ | $98.01_{\pm0.02}$ | $84.03_{\pm0.03}$ | $73.42_{\pm0.03}$ | $69.13_{\pm12.57}$ | $1.152_{\pm0.011}$ | $211.52_{\pm0.96}$ |
| BoundaryExpand | $71.39_{\pm0.31}$ | $99.20_{\pm0.04}$ | $92.53_{\pm0.02}$ | $7.69_{\pm0.33}$ | $0.892_{\pm0.001}$ | $\underline{0.19}_{\pm0.01}$ | $77.22_{\pm0.11}$ | $82.79_{\pm0.08}$ | $71.78_{\pm0.09}$ | $1.43_{\pm0.51}$ | $0.628_{\pm0.005}$ | $5.14_{\pm0.02}$ |
| BoundaryShrink | $85.16_{\pm0.42}$ | $99.60_{\pm0.17}$ | $93.48_{\pm0.40}$ | $0.25_{\pm0.43}$ | $0.887_{\pm0.009}$ | $0.59_{\pm0.02}$ | $91.20_{\pm0.02}$ | $81.41_{\pm0.17}$ | $70.55_{\pm0.08}$ | $1.45_{\pm0.06}$ | $0.543_{\pm0.011}$ | $\underline{4.81}_{\pm0.01}$ |
| EU-5 | $\mathbf{100.0}_{\pm0.0}$ | $\mathbf{100.0}_{\pm0.0}$ | $95.25_{\pm0.02}$ | $0.06_{\pm0.03}$ | $0.528_{\pm0.005}$ | $1.54_{\pm0.0}$ | $\mathbf{100.0}_{\pm0.0}$ | $79.62_{\pm0.0}$ | $71.22_{\pm0.13}$ | $13.33_{\pm1.53}$ | $0.183_{\pm0.028}$ | $193.38_{\pm0.78}$ |
| CF-5 | $98.13_{\pm1.39}$ | $\mathbf{100.0}_{\pm0.0}$ | $\underline{95.54}_{\pm0.09}$ | $0.0_{\pm0.0}$ | $0.675_{\pm0.027}$ | $1.57_{\pm0.03}$ | $\mathbf{100.0}_{\pm0.0}$ | $84.31_{\pm0.08}$ | $74.16_{\pm0.06}$ | $10.21_{\pm5.33}$ | $0.701_{\pm0.014}$ | $81.53_{\pm0.56}$ |
| EU-10 | $\mathbf{100.0}_{\pm0.0}$ | $99.50_{\pm0.02}$ | $93.61_{\pm0.08}$ | $15.24_{\pm1.08}$ | $-0.349_{\pm0.019}$ | $2.42_{\pm0.11}$ | $\mathbf{100.0}_{\pm0.0}$ | $71.84_{\pm0.03}$ | $65.78_{\pm0.02}$ | $16.65_{\pm1.91}$ | $\underline{-0.051}_{\pm0.021}$ | $193.79_{\pm0.47}$ |
| CF-10 | $\mathbf{100.0}_{\pm0.0}$ | $99.98_{\pm0.0}$ | $94.95_{\pm0.05}$ | $\mathbf{11.61}_{\pm0.91}$ | $-0.060_{\pm0.017}$ | $2.31_{\pm0.03}$ | $\mathbf{100.0}_{\pm0.0}$ | $80.87_{\pm0.04}$ | $72.34_{\pm0.08}$ | $13.99_{\pm5.41}$ | $0.608_{\pm0.012}$ | $82.29_{\pm0.34}$ |
| SCRUB | $\mathbf{100.0}_{\pm0.0}$ | $\mathbf{100.0}_{\pm0.0}$ | $95.37_{\pm0.04}$ | $19.73_{\pm1.92}$ | $\underline{-0.056}_{\pm0.008}$ | $3.49_{\pm0.02}$ | $99.28_{\pm0.07}$ | $\underline{88.39}_{\pm0.04}$ | $76.51_{\pm0.03}$ | $7.42_{\pm0.51}$ | $0.517_{\pm0.011}$ | $426.04_{\pm2.98}$ |
| SALUN | $\underline{99.99}_{\pm0.01}$ | $\mathbf{100.0}_{\pm0.0}$ | $95.42_{\pm0.12}$ | $0.01_{\pm0.0}$ | $0.936_{\pm0.012}$ | $3.54_{\pm0.11}$ | $89.67_{\pm0.27}$ | $88.54_{\pm0.05}$ | $75.54_{\pm0.10}$ | $0.50_{\pm0.09}$ | $0.343_{\pm0.017}$ | $793.82_{\pm3.32}$ |
| $\ell_1$-sparse | $\mathbf{100.0}_{\pm0.0}$ | $99.93_{\pm0.0}$ | $94.90_{\pm0.10}$ | $1.56_{\pm0.09}$ | $0.293_{\pm0.012}$ | $2.96_{\pm0.03}$ | $97.57_{\pm0.61}$ | $85.33_{\pm0.07}$ | $74.77_{\pm0.03}$ | $\underline{8.84}_{\pm1.39}$ | $0.239_{\pm0.031}$ | $226.74_{\pm1.35}$ |
| **COLA** | $\mathbf{100.0}_{\pm0.0}$ | $\mathbf{100.0}_{\pm0.0}$ | $95.36_{\pm0.06}$ | $\underline{12.64}_{\pm0.92}$ | $\mathbf{0.010}_{\pm0.006}$ | $4.91_{\pm0.04}$ | $\mathbf{100.0}_{\pm0.0}$ | $87.93_{\pm0.05}$ | $76.15_{\pm0.04}$ | $\mathbf{9.95}_{\pm1.21}$ | $\mathbf{0.040}_{\pm0.042}$ | $171.44_{\pm0.75}$ |

studies likely ranked MU methods based on MIA and RTE. However, as discussed earlier, relying solely on black-box metrics can be misleading, as they fail to account for residual information.

Indeed, some methods show strong MIA performance but fail to remove forget data from intermediate layers, as reflected by high IDI values. For instance, CF-5 on ImageNet-1K achieves a favorable MIA value (10.21) close to Retrain (9.41) in the shortest time (81.53 min), yet its IDI (0.701) shows significant retention of forget data. Similarly, EU-5 on CIFAR-10, which appears highly efficient (1.54 min), presents a high IDI (0.528), suggesting that its efficiency stems from incomplete unlearning. The discrepancy between black-box metrics (MIA, JSD) and IDI is similarly observed in random data forgetting,

Table 2: Performance summary for random data forgetting on (CIFAR-10, ResNet-18). The notation for **bold** and underline, as well as the number of independent trials, is consistent with Table 1.

| | CIFAR-10 (500 samples per class) | | | | | | |
|---|---|---|---|---|---|---|---|
| Methods | UA | RA | TA | MIA | JSD | IDI | RTE (min) |
| Retrain | 3.94 | 100.0 | 95.26 | 75.12 | 0.0 | 0.0 | 152.87 |
| HD | $\underline{3.64}_{\pm1.66}$ | $97.93_{\pm1.38}$ | $92.80_{\pm1.18}$ | $\underline{77.47}_{\pm4.09}$ | $0.08_{\pm0.04}$ | $1.000_{\pm0.0}$ | $\mathbf{0.30}_{\pm0.05}$ |
| FT | $5.03_{\pm0.40}$ | $98.95_{\pm0.21}$ | $92.94_{\pm0.26}$ | $83.52_{\pm0.58}$ | $0.07_{\pm0.11}$ | $\underline{-0.069}_{\pm0.013}$ | $8.11_{\pm0.03}$ |
| RL | $4.77_{\pm0.27}$ | $\mathbf{99.92}_{\pm0.0}$ | $\mathbf{93.54}_{\pm0.04}$ | $22.47_{\pm1.19}$ | $0.38_{\pm0.02}$ | $0.084_{\pm0.030}$ | $2.75_{\pm0.01}$ |
| GA | $2.86_{\pm0.76}$ | $98.37_{\pm0.71}$ | $91.90_{\pm0.70}$ | $85.49_{\pm2.17}$ | $0.09_{\pm0.01}$ | $0.924_{\pm0.028}$ | $4.31_{\pm0.03}$ |
| Bad-T | $5.47_{\pm1.05}$ | $\underline{99.87}_{\pm0.05}$ | $91.51_{\pm0.61}$ | $39.53_{\pm3.43}$ | $0.27_{\pm0.03}$ | $0.939_{\pm0.053}$ | $4.78_{\pm0.09}$ |
| EU-10 | $3.16_{\pm0.19}$ | $98.68_{\pm0.08}$ | $93.07_{\pm0.12}$ | $83.40_{\pm0.21}$ | $0.06_{\pm0.01}$ | $-0.110_{\pm0.013}$ | $2.13_{\pm0.05}$ |
| CF-10 | $2.71_{\pm0.24}$ | $99.11_{\pm0.06}$ | $\underline{93.47}_{\pm0.15}$ | $84.33_{\pm0.05}$ | $\mathbf{0.05}_{\pm0.01}$ | $0.219_{\pm0.029}$ | $\underline{2.10}_{\pm0.06}$ |
| SCRUB | $4.31_{\pm1.50}$ | $96.21_{\pm1.70}$ | $88.83_{\pm1.86}$ | $37.88_{\pm7.65}$ | $0.56_{\pm0.09}$ | $0.322_{\pm0.016}$ | $3.37_{\pm0.05}$ |
| SALUN | $2.74_{\pm0.30}$ | $97.77_{\pm0.04}$ | $91.68_{\pm0.44}$ | $83.52_{\pm2.20}$ | $0.10_{\pm0.03}$ | $0.861_{\pm0.012}$ | $5.69_{\pm0.04}$ |
| $\ell_1$-sparse | $5.47_{\pm0.22}$ | $96.66_{\pm0.07}$ | $91.31_{\pm0.25}$ | $\mathbf{77.12}_{\pm0.21}$ | $0.09_{\pm0.01}$ | $-0.157_{\pm0.026}$ | $3.03_{\pm0.04}$ |
| **COLA+** | $\mathbf{3.90}_{\pm0.08}$ | $99.24_{\pm0.17}$ | $93.23_{\pm0.09}$ | $83.48_{\pm0.10}$ | $\mathbf{0.06}_{\pm0.01}$ | $\mathbf{0.024}_{\pm0.010}$ | $7.80_{\pm0.02}$ |

as shown in Table 2, particularly for methods like SALUN. By incorporating IDI alongside existing metrics, we gain a more comprehensive and insightful evaluation of MU methods.

In Figure 7, we present the CIFAR-100 results comparing task performance (TA) and unlearning quality (IDI). Methods like Bad-T achieve high accuracy but retain substantial residual information (high IDI). In contrast, EU-5 and $\ell_1$-sparse effectively remove forget set (low IDI) but experience substantial accuracy loss (low TA), indicating damage to essential features for task performance. Despite its simplicity, COLA (and COLA+) achieves state-of-the-art IDI performance across all experiments, as shown in Table 1, Table 2, and Figure 7, effectively eliminating feature-level information. This is further supported by the recovery experiment (Figure 4), where COLA achieves low accuracy comparable to Retrain. In addition to excelling in IDI, COLA (and COLA+) performs well on black-box metrics, preserving task performance and maintaining output similarity (Table 1 and Table 2). While it's computational

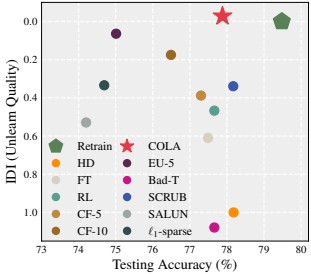

Figure 7: IDI and TA of Retrain and MU methods for single-class forgetting (ResNet-18, CIFAR-100).

cost (RTE) is relatively high, we emphasize the inherent challenge of thoroughly removing the forget set while retaining model utility. Further experimental results are provided in Appendix D.

## 5.3 DISCUSSIONS

**IDI as a Real-World Efficacy Metric.** Accuracy metrics (UA, RA, TA) and efficacy metrics (MIA, JSD), commonly used in recent unlearning studies, require the presence of Retrain as a gold standard to compare model outputs. While this approach is crucial for advancing MU methods in controlled experimental settings, where the field of unlearning for DNNs is still in its infancy, it

becomes impractical in real-world applications where Retrain is unavailable. Similar to current black-box metrics, the original formulation of IDI (see Equations 1 and 2) uses Retrain as a reference to assess unlearning efficacy. However, IDI allows for flexibility by using any available unlearned model as the reference. Although the absence of Retrain changes the interpretation of IDI (*i.e.*, an IDI of zero means complete unlearning as Retrain), it still provides valuable insights relative to the chosen reference. This adaptability enhances the IDI's practicality, making it useful for evaluating unlearned models even in real-world scenarios. A detailed explanation and examples are provided in Appendix E.3.

**Consistency and Scalability of IDI.** Consistency is crucial for unlearning metrics, but many fall short (Chundawat et al., 2023a; Tarun et al., 2023b; Becker and Liebig, 2022). Also, a major issue with using model parameters for white-box efficacy evaluation in DNNs is their inconsistency, as weights can vary significantly due to stochastic factors (*e.g.*, random seeds), making comparisons between the unlearned model and Retrain ambiguous (Yang and Shami, 2020; Goel et al., 2022). In contrast, IDI remains robust, delivering consistent results across models from the same algorithm, as evidenced by low standard deviations in independent trials (see Table 1 and Table 2).

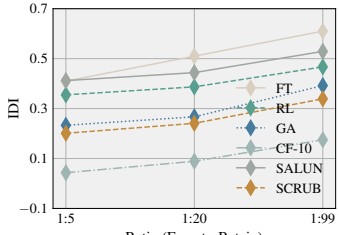

Figure 8: IDI of six methods with varying binary label ratios, in the single-class forgetting on (CIFAR-100, ResNet-18), where $a : b$ denotes the ratio of forgetting to retaining samples.

Furthermore, IDI provides consistent results without requiring the entire dataset $\mathcal{D}$. As shown in Figure 8, the relative rankings of methods remain stable across different data ratios in the classwise forgetting task on CIFAR-100. This efficiency allows for unlearning evaluations with reduced computational cost, where white-box metrics often demand significant resources.

**IDI compare to White-Box MIA.** While black-box MIA, adapted from privacy studies, is widely used as an evaluation tool in unlearning literature, we explore the potential of white-box MIA, which has not traditionally been employed for this purpose, and compare it with IDI. Specifically, we evaluate two white-box MIA methods: one leveraging model activations and another utilizing gradients (Nasr et al., 2019). Table 3 presents the results of white-box MIA and IDI in single-class forgetting scenarios. White-box MIA delivers consistent results on CIFAR-10 but becomes unstable as the dataset scales to CIFAR-100, with significant

Table 3: Performance summary of MU methods on white-box MIAs (Activation, Gradient) and IDI for single-class forgetting on ResNet-18. MIA values represent the attack success rate (%) for distinguishing forgetting samples. "Random" refers to a model randomly initialized without prior training.

| Methods | CIFAR-10 | | | CIFAR-100 | | |
|---|---|---|---|---|---|---|
| | Activation | Gradient | IDI | Activation | Gradient | IDI |
| Original | $99.98_{\pm 0.03}$ | $100.0_{\pm 0.0}$ | $1.000$ | $53.13_{\pm 2.88}$ | $61.34_{\pm 3.23}$ | $1.000$ |
| Retrain | $94.89_{\pm 1.07}$ | $95.13_{\pm 1.12}$ | $0.000$ | $52.87_{\pm 6.15}$ | $59.12_{\pm 4.12}$ | $0.000$ |
| Random | $52.89_{\pm 41.03}$ | $45.23_{\pm 23.04}$ | $-1.281_{\pm 0.018}$ | $53.20_{\pm 5.15}$ | $47.12_{\pm 7.21}$ | $-2.955_{\pm 0.046}$ |
| RL | $100.0_{\pm 0.0}$ | $99.98_{\pm 0.01}$ | $0.830_{\pm 0.005}$ | $93.20_{\pm 3.53}$ | $95.30_{\pm 0.82}$ | $0.467_{\pm 0.010}$ |
| GA | $97.07_{\pm 0.35}$ | $96.01_{\pm 0.13}$ | $0.334_{\pm 0.014}$ | $97.44_{\pm 2.12}$ | $82.44_{\pm 0.95}$ | $0.392_{\pm 0.021}$ |
| EU-10 | $86.13_{\pm 4.78}$ | $89.42_{\pm 2.32}$ | $-0.349_{\pm 0.019}$ | $64.41_{\pm 1.65}$ | $72.13_{\pm 4.13}$ | $-0.221_{\pm 0.009}$ |
| CF-10 | $97.99_{\pm 0.38}$ | $98.33_{\pm 0.23}$ | $-0.060_{\pm 0.017}$ | $21.62_{\pm 0.61}$ | $23.15_{\pm 1.23}$ | $0.175_{\pm 0.040}$ |
| SCRUB | $99.43_{\pm 0.09}$ | $99.15_{\pm 0.05}$ | $-0.056_{\pm 0.008}$ | $46.44_{\pm 1.28}$ | $62.31_{\pm 1.73}$ | $0.339_{\pm 0.069}$ |
| COLA | $92.26_{\pm 0.08}$ | $93.12_{\pm 0.11}$ | $0.010_{\pm 0.006}$ | $61.08_{\pm 0.23}$ | $65.24_{\pm 0.43}$ | $-0.037_{\pm 0.006}$ |

variability in MIA values across algorithms. This instability is further highlighted with a randomly initialized model, which produces MIA values comparable to Retrain despite no actual training. In contrast, IDI provides stable and interpretable results, yielding strongly negative values for randomly initialized models, accurately reflecting their lack of residual information. This underscores IDI's reliability as a robust and interpretable metric for unlearning evaluation.

## 6 CONCLUSION

We highlight the limitations of relying on black-box metrics to assess unlearning efficacy in typical approximate unlearning studies. Although intermediate features capable of reconstructing forgotten information persist, these metrics fail to capture the key aspects required for strong unlearning, often misleading evaluations. To address this, we introduce the Information Difference Index (IDI) from an information-theoretic perspective, alongside the contrastive-based COLA baseline for direct feature-level unlearning. Through extensive experiments, we demonstrate the validity and practicality of IDI, showing that it complements existing metrics for a more comprehensive evaluation of strong unlearning. In addition, we highlight the effectiveness of the COLA baseline, despite its simplicity.

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

## A  RANDOM DATA FORGETTING

Another scenario in machine unlearning (MU) is *random data forgetting*, which involves forgetting a randomly selected subset of data across multiple classes. This differs from the *class-wise forgetting* task, which aims to forget entire data from single or multiple classes.

### A.1  INFORMATION DIFFERENCE INDEX FOR RANDOM DATA FORGETTING

To calculate the information difference index (IDI) for class-wise forgetting, we employ a binary label $Y$ to determine whether a sample belongs to the retain or forget set. However, this approach is inadequate for random data forgetting, where samples span multiple classes and a minor fraction of each class is targeted for forgetting. As a result, no single class is completely removed. To address this, we transform the binary label $Y$ into a multiclass label $Y_C$, which reflects the ground-truth class label of each sample. Consequently, we define the IDI for random data forgetting as follows:

$$\mathbf{IDI}_{random}(\theta_{\mathbf{u}}) = \frac{\mathbf{ID}_{random}(\theta_{\mathbf{u}})}{\mathbf{ID}_{random}(\theta_{\mathbf{0}})}, \tag{3}$$

where $\mathbf{ID}_{random}(\theta_{\mathbf{u}}) = \sum_{\ell=1}^{L} \big( I(\mathbf{Z}_\ell^{(\mathbf{u})}; Y_C) - I(\mathbf{Z}_\ell^{(\mathbf{r})}; Y_C) \big)$. Unlike the ID computed in a class-wise forgetting, $\mathbf{ID}_{random}(\cdot)$ utilizes only the forget set $\mathcal{D}_f$. Intuitively, we expect the mutual information $I(\mathbf{Z}^{(\mathbf{o})}; Y_C)$ to be higher than $I(\mathbf{Z}^{(\mathbf{r})}; Y_C)$ because Original explicitly learned the relationship between the forget samples and their ground truth labels, while Retrain did not. Although the labels have transitioned from binary to multiple classes, the function $f_{\nu_\ell}$ remains unchanged. For $g_{\eta_\ell}$, it now employs the $C$ dimension of vectors, where $C$ represents the total number of data classes.

### A.2  COLA+

The core idea behind COLA is to induce catastrophic forgetting within the model's encoder in the collapse phase, making the influence of the forget set vanish implicitly. This approach is effective for class-wise forgetting tasks, where the forget set includes distinct classes. However, it may be less effective for random data forgetting, where the forget set and retain set samples generally share the same classes and are not easily distinguishable. To address this, we aim to explicitly remove the information of the forget set through pseudo-labeling. This variant, called COLA+, assigns the second-highest predicted label to the forget set samples before unlearning with supervised contrastive loss (Khosla et al., 2020). This pseudo-labeling effectively collapsing the forget set features into the retain set clusters of other classes, while reducing the confusion of the knowledge of the retain set. The results of the COLA+ experiment on the random data forgetting task are presented in Appendix D.5.

## B  NETWORK PARAMETRIZATIONS FOR INFONCE LOSS

This section provides a detailed explanation of the parameterization of the neural network critic functions used in the InfoNCE loss, including layer-specific adaptations.

### B.1  CRITIC FUNCTIONS FOR INFONCE LOSS

To compute the InfoNCE loss, we parameterize two critic functions: $f_\nu$ and $g_\eta$, where $\nu$ and $\eta$ represent the learnable parameters of their respective neural networks. For each layer $\ell$ in the network, these functions are defined as follows:

1. **Critic $f_{\nu_\ell}$:**
   - $f_{\nu_\ell} : \mathcal{Z}_\ell \to \mathbb{R}^d$, where $\mathcal{Z}_\ell$ represents the feature space at layer $\ell$.
   - This function maps raw or intermediate features $\mathbf{Z}_\ell$ to a $d$-dimensional embedding space. In earlier layers, $f_{\nu_\ell}$ must process raw, less interpretable features, making it more complex. For later layers, where features are more structured, $f_{\nu_\ell}$ can leverage the refined representations for better alignment with $Y$.

2. **Critic** $g_{\eta_\ell}$:

- $g_{\eta_\ell} : \{0, 1\} \to \mathbb{R}^d$.
- For the binary variable $Y$, $g_{\eta_\ell}$ is parameterized as a pair of trainable $d$-dimensional vectors: $g_{\eta_\ell}(0)$ and $g_{\eta_\ell}(1)$. Depending on the label $Y$, the corresponding vector is selected to represent the target embedding for contrastive learning.

### B.2 LAYER-SPECIFIC PARAMETERIZATION

Each layer $\ell$ has independent sets of parameters $\nu_\ell$ and $\eta_\ell$. This design allows the model to adapt to the varying complexity of feature representations across the network. Specifically:

- In earlier layers, $f_{\nu_\ell}$ focuses on extracting information from raw features $\mathbf{Z}_\ell$, which are less structured and more challenging to interpret.
- In later layers, $f_{\nu_\ell}$ benefits from more refined features, enabling a more direct alignment with $Y$.

## C EXPERIMENT DETAILS

### C.1 EVALUATION METRICS DETAIL

**UA, RA, TA** We compute accuracy as follows:

$$\text{Acc}_{\mathcal{D}}(\theta) = \frac{1}{|\mathcal{D}|} \sum_{(x,y) \in \mathcal{D}} \mathbb{1}\left[\arg\max\left(f(x;\theta)\right) = y\right], \tag{4}$$

where $f(x;\theta)$ represents the model's output logits for input $x$ with parameters $\theta$, and $y$ is the ground truth label. Unlearning accuracy (UA), which quantifies the model's task performance on forgetting data, is defined as $UA(\theta_{\mathbf{u}}) = 1 - \text{Acc}_{\mathcal{D}_f}(\theta_{\mathbf{u}})$. Remaining accuracy (RA) measures the model's performance on the retain set $\mathcal{D}_r$, which should be preserved after unlearning, and is defined as $RA(\theta_{\mathbf{u}}) = \text{Acc}_{\mathcal{D}_r}(\theta_{\mathbf{u}})$. Finally, testing accuracy (TA) evaluates generalization to unseen data, and is defined as $TA(\theta_{\mathbf{u}}) = \text{Acc}_{\mathcal{D}_{test}}(\theta_{\mathbf{u}})$. It is important to note that better unlearning in terms of accuracy reflects a smaller performance gap between the unlearned model and Retrain, meaning that higher accuracy levels are not necessarily better. Refer to (Jia et al., 2023) for detailed explanation.

**MIA.** Membership Inference Attack (MIA) (Shokri et al., 2017; Carlini et al., 2022) determines whether a specific data record was part of a model's training set by leveraging auxiliary classifiers to distinguish between training and non-training data based on the model's output.

In the context of unlearning, membership inference attack (MIA) is primarily used as an evaluation metric, rather than representing an adversarial scenario where an attacker attempts to extract membership information from the unlearned model. Consequently, a comparable MIA success rate on the forgetting data relative to Retrain signifies a more effective unlearning algorithm. Unlike the original MIA implementation (Shokri et al., 2017), which utilizes multiple shadow models, MIA variants in the unlearning often employ a single auxiliary classifier for each unlearning method (Jia et al., 2023). A detailed comparison of these approaches can be found in (Hayes et al., 2024).

The MIA implementation in our study has two phases: the *training phase* and the *testing phase*.

During the *training phase*, we create a balanced dataset by equally sampling from the retain set ($\mathcal{D}_r$) and the test set, explicitly excluding the forget set ($\mathcal{D}_f$). We then use this balanced dataset to train the MIA predictor with two output categories (train, non-train), allowing it to differentiate between training and non-training samples.

In the *testing phase*, the trained MIA predictor is used to evaluate the efficacy of the unlearning methods. Specifically, the **MIA** metric is calculated by applying the MIA predictor to the unlearned model ($\theta_u$) using the forget set ($\mathcal{D}_f$). The objective is to determine how many samples within $\mathcal{D}_f$ are identified as training samples by the MIA predictor.

Formally, MIA is defined as:

$$\text{MIA} = 1 - \frac{\text{TN}}{|\mathcal{D}_f|} \tag{5}$$

where *TN* represents the number of true negatives (*i.e.*, the number of forget samples correctly predicted as non-training examples by the MIA predictor), and $|\mathcal{D}_f|$ denotes the total number of samples in the forget set. Overall, MIA leverages privacy attack mechanisms to validate the effectiveness of the unlearning process, providing a quantitative measure of how successfully the model has 'forgotten' specific data resembling Retrain.

We consider two widely adopted variants of MIA. The first variant, **C-MIA (Confidence-based MIA)**, assesses membership based on the confidence score, which is the predicted probability of the true class (Fan et al., 2024; Jia et al., 2023). The second variant, **E-MIA (Entropy-based MIA)**, infers membership by examining the entropy of the model's outputs, calculated as $H(x) = -\sum_i \mathbf{p}_i(x) \cdot \log \mathbf{p}_i(x)$ (Chundawat et al., 2023b; Foster et al., 2024; Kurmanji et al., 2023). Higher entropy indicates greater uncertainty in the model's predictions, often signaling non-training samples. We primarily report results using E-MIA due to its more pronounced differences across various baselines compared to C-MIA. It is noteworthy that our head distillation (HD) method achieves similar performance outcomes with both E-MIA and C-MIA.

**U-LiRA**  U-LiRA is a variant of black-box membership inference attack (MIA) designed to evaluate the privacy protection of unlearning algorithms (Hayes et al., 2024). For our LiRA MIA experiments in Appendix D, we followed the U-LiRA methodology from Hayes et al. (2024), training 128 ResNet-18 models on random splits of half the CIFAR-10 training set, ensuring that each sample is included in 64 and excluded from 64 models on average. We applied the unlearning algorithm to 40 random forget sets (200 samples each) per model, resulting in 5,120 unlearned models. For evaluation, we used 2,560 shadow and 2,560 target models, focusing on class 4 samples. Testing each method required 300–500 GPU hours, highlighting the cost-intensive nature of LiRA when adopting to unlearning; additional details can be found in (Hayes et al., 2024).

**JSD**  Jensen-Shannon divergence (JSD) is presented in Bad-T (Chundawat et al., 2023b). It measures the distance between the output distributions of the unlearned model and Retrain. JSD is measured as follows:

$$\text{JSD}_{\mathcal{D}}(\theta_{\mathbf{u}}, \theta_{\mathbf{r}}) = 0.5 \cdot KL(f(x; \theta_u) \,\|\, m) + 0.5 \cdot KL(f(x; \theta_r) \,\|\, m), \tag{6}$$

where $KL(\cdot)$ is Kullback-Leibler divergence, $x$ is data from $\mathcal{D}$, and m $= \frac{f(x;\theta_{\mathbf{u}})+f(x;\theta_{\mathbf{r}})}{2}$. Here, $f(x;\theta)$ represents the model's output probability distribution for input $x$ with parameters $\theta$. A smaller distance means better unlearning as the unlearned model better mimics Retrain.

**RTE**  Runtime efficiency (RTE) measures the time that an algorithm spends to complete the unlearning, where smaller RTE indicates more efficient unlearning (Fan et al., 2024; Jia et al., 2023; Foster et al., 2024). Since it measures the experiment wall-clock time, it has high variance depending on the experiment environment.

## C.2 Approximate MU Baselines.

We conduct our experiments on several widely used or recent approximate MU baselines: Finetuning (**FT**) (Golatkar et al., 2020a) finetunes Original $\theta_{\mathbf{o}}$ with retain set $\mathcal{D}_r$, inducing catastrophic forgetting (French, 1999; Kirkpatrick et al., 2017) of $\mathcal{D}_f$. Random labeling (**RL**) (Golatkar et al., 2020a) involves finetuning $\theta_{\mathbf{o}}$ with randomly labeled forget set $\mathcal{D}_f$. Gradient ascent (**GA**) (Thudi et al., 2022) trains $\theta_{\mathbf{o}}$ with reverse gradient steps using $\mathcal{D}_f$. **Bad-T** (Chundawat et al., 2023b) uses a teacher-student framework that utilizes distillation techniques, distinguishing between beneficial and detrimental influences through good and bad teachers to refine the learning process. Catastrophic forgetting-k (**CF-k**) and exact unlearning-k (**EU-k**) (Goel et al., 2022) involve either finetuning (CF-k) or retraining (EU-k) the last $k$ layers of the model using $\mathcal{D}_r$ while freezing the prior layers. **SCRUB** (Kurmanji et al., 2023) employs a technique of positive distillation from $\theta_{\mathbf{o}}$ using the $\mathcal{D}_r$, and negative distillation on the $\mathcal{D}_f$, which helps in selectively retaining beneficial knowledge while discarding the unwanted influences. $\ell_1$**-sparse** (Jia et al., 2023) enhances the model's ability to forget by strategically inducing weight sparsity in $\theta_{\mathbf{o}}$. **SALUN** (Fan et al., 2024) finetunes the salient

Table 4: Training configuration for Original and Retrain.

| Settings | CIFAR-10 / CIFAR-100 | | ImageNet-1K | |
|---|---|---|---|---|
| | Resnet-18 / Resnet-50 | ViT | ResNet-50 | ViT |
| Epochs | 300 | 3 | 90 | 30 |
| Batch Size | 128 | | 256 | 512 |
| LR | 0.1 | 0.00002 | 0.1 | 0.02 |
| Optimizer | SGD | | | |
| Momentum | 0.9 | | | |
| L2 regularization | 0.0005 | | 0 | |
| Scheduler | CosineAnnealing | | | |

weights of $\theta_o$ using a method that incorporates random labeling. **BoundaryShrink** (Chen et al., 2023) reassigns the $\mathcal{D}_f$ to their nearest but incorrect labels, splitting the decision space of the forgetting class. **BoundaryExpand** (Chen et al., 2023) maps $\mathcal{D}_f$ to an extra shadow class, bypassing the need to find nearest labels.

## C.3 DATASETS AND MODELS

We conduct image classification experiments utilizing well-established datasets and models. The datasets include CIFAR-10, CIFAR-100 (Krizhevsky, 2009), and ImageNet-1K (Deng et al., 2009); and the models are ResNet-18, ResNet-50 (He et al., 2016), and Vision Transformer (ViT) (Dosovitskiy et al., 2021). CIFAR-10 and CIFAR-100 each comprise 50,000 training images distributed across 10 and 100 classes, respectively, each with an original resolution of 32 x 32 pixels. In our experiments, we resize the images in ImageNet-1K, which consists of 1,281,167 training images across 1,000 classes, to 224 x 224 pixels. Similarly, for the ViT experiments, we resize CIFAR images to 224 x 224 pixels to accommodate the architecture's requirements. Throughout the training process, including pretraining and unlearning phases, we employ basic data augmentation techniques such as random cropping and random horizontal flipping.

## C.4 PRETRAINING SETTINGS

To perform unlearning, we require two models: **Original**, trained on the entire dataset $\mathcal{D}$, and **Retrain**, trained on the retain set $\mathcal{D}_r$. Original initializes the unlearning model. After unlearning, Retrain evaluates them. Table 4 summarizes the training configurations for each dataset and model combination. We train ResNet models from scratch and initialize ViT models with ImageNet-21K pretrained weights. For training on ImageNet-1K, we follow the configurations provided by Pytorch[1].

## C.5 UNLEARNING SETTINGS

We aim to follow the hyperparameters provided by the original papers. However, many hyperparameters are missing since most existing works do not experiment with large-scale datasets and models. Additionally, some values from the original papers result in poor performance, likely due to different experiment settings, as most previous work performed unlearning without any data augmentation, unlike our experiments. Therefore, we conduct thorough hyperparameter searches for each baseline. The detailed hyperparameters of each baseline, including our method COLA and COLA+, are shown in Table 9 and Table 10. We use the same optimizer and batch size from the original papers and focus on finding the best epoch number and learning rate in terms of unlearning accuracy (UA) and testing accuracy (TA). Note that we implement gradient ascent (GA) from SCRUB (Kurmanji et al., 2023) (referred to as 'NegGrad+') due to its strong performance.

---

[1]https://github.com/pytorch/examples/tree/main/imagenet

## C.6  COLA AND COLA+ PSEUDO CODE

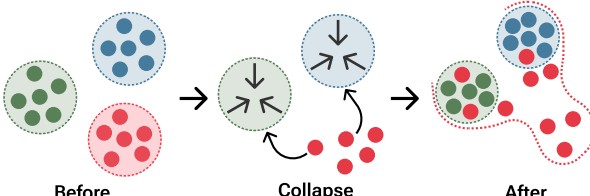

Figure 9: Illustration of the collapse phase of COLA. Features (post-encoder, pre-head) from forget set $\mathcal{D}_f$ are represented in red, while features from retain set $\mathcal{D}_r$ are represented in green and blue. The figure shows a class-wise forgetting task. Best viewed in color.

Algorithm 1 shows the pseudo code of our two-step framework COLA. Only using the retain set $\mathcal{D}_r$, in the *collapse phase* (see Figure 9), We first train the encoder of the model using supervised contrastive loss (Khosla et al., 2020) as follows:

$$\text{SupConLoss}(b, \theta_{\mathbf{enc}}) = \frac{1}{|b|} \sum_i \frac{1}{|P(i)|} \sum_{p \in P(i)} -\log \frac{\exp(\mathbf{z}_i \cdot \mathbf{z}_p / \tau)}{\sum_{a \in A(i)} \exp(\mathbf{z}_i \cdot \mathbf{z}_a / \tau)}, \tag{7}$$

where $P(i)$ is the set of indices of positive samples sharing the same label as sample $i$, $A(i)$ is the set of all indices excluding sample $i$, $\tau$ is a temperature, and $z_i = F(x_i; \theta_{\mathbf{enc}})$, the output feature of the model encoder. Then we train the whole network using cross-entropy loss in the *align phase*. COLA+ additionally utilizes forget set $\mathcal{D}_f$ in the collapse phase, where the label of forget samples is changed to the class label closest to the original label, determined by the logit output of the head of Original. Its pseudo code is presented in Algorithm 2.

---

**Algorithm 1** Pseudo Code of *COLA*
---

**Require:** learning rate $\eta$, number of epochs $E_1, E_2$, retain set $\mathcal{D}_r = \{(x_i, y_i) \mid (x_i, y_i) \in \mathcal{D}_r\}$, encoder $F(\cdot; \theta)$, and model weight $\theta = \{\theta_{\mathbf{enc}}, \theta_{\mathbf{head}}\}$

$\theta_{\mathbf{u,enc}} \leftarrow \theta_{\mathbf{o,enc}}$            ▷ Collapse phase
**for** $e \leftarrow 0$ **to** $E_1 - 1$ **do**
    **for** all batches $b$ of $\mathcal{D}_r$ **do**
        $L = \text{SupConLoss}(b, \theta_{\mathbf{u,enc}})$            ▷ Equation 7
        $\theta_{\mathbf{u,enc}} \leftarrow \theta_{\mathbf{u,enc}} - \eta \nabla_{\theta_{\mathbf{u,enc}}} L$
    **end for**
**end for**

$\theta_{\mathbf{u,head}} \leftarrow$ random initialization            ▷ Align phase
**for** $e \leftarrow 0$ **to** $E_2 - 1$ **do**
    **for** all batches $b$ of $\mathcal{D}_r$ **do**
        $\theta_{\mathbf{u}} \leftarrow \theta_{\mathbf{u}} - \eta \nabla_{\theta_{\mathbf{u}}} L_{CE}$
    **end for**
**end for**
**return** $\theta_{\mathbf{u}} = \{\theta_{\mathbf{u,enc}}, \theta_{\mathbf{u,head}}\}$

---

## C.7  IDI DETAILS

To derive IDI from features, it is necessary to train the critic functions $f_{\nu_\ell}$ and $g_{\eta_\ell}$, as referenced in Section 4. For the training of $g_{\eta_\ell}$, a learning rate of $5 \cdot 10^{-4}$ is applied in all architectures and datasets. Meanwhile, for $f_{\nu_\ell}$, the learning rates are set at $2 \cdot 10^{-5}$ for CIFAR10 ResNet-18, $2 \cdot 10^{-6}$ for ViT ImageNet-1K, and $1 \cdot 10^{-5}$ for the remaining architectures of the data set.

To get IDI, we analyzed the outputs from the layers of different models. Specifically, we evaluated the last two block outputs for ResNet18 and the final three for ResNet50. For Vision Transformer (ViT),

---

**Algorithm 2** Pseudo Code of *COLA+*

---

**Require:** learning rate $\eta$, number of epochs $E_1, E_2$, retain set $\mathcal{D}_r = \{(x_i, y_i) \mid (x_i, y_i) \in \mathcal{D}_r\}$, forget set $\mathcal{D}_f = \{(x'_i, y'_i) \mid (x'_i, y'_i) \in \mathcal{D}_f\}$, encoder $F(\cdot; \theta)$, head $G(\cdot; \theta)$, and model weight $\theta = \{\theta_{\mathbf{enc}}, \theta_{\mathbf{head}}\}$

$\theta_{\mathbf{u,enc}} \leftarrow \theta_{\mathbf{o,enc}}$                 ▷ Collapse phase
$\theta_{\mathbf{u,head}} \leftarrow \theta_{\mathbf{o,head}}$
**for** $e \leftarrow 0$ **to** $E_1 - 1$ **do**
  **for** $\{b_r, b_f\}$ in all batches of $\{\mathcal{D}_r, \mathcal{D}_f\}$ **do**
   **for** $x'_i \in b_f$ **do**
    $y'_i \leftarrow \arg\max_y \mathrm{softmax}(G(F(x'_i; \theta_{\mathbf{u,enc}}); \theta_{\mathbf{u,head}})) \cdot \mathbb{I}[y \neq y'_i]$   ▷ Pseudo-labeling
   **end for**
   $b \leftarrow b_r + b_f$
   $L = \mathrm{SupConLoss}(b, \theta_{\mathbf{u,enc}})$           ▷ Equation 7
   $\theta_{\mathbf{u,enc}} \leftarrow \theta_{\mathbf{u,enc}} - \eta \nabla_{\theta_{\mathbf{u,enc}}} L$
  **end for**
**end for**

                    ▷ Align phase
**for** $e \leftarrow 0$ **to** $E_2 - 1$ **do**
  **for** all batches $b$ of $\mathcal{D}_r$ **do**
   $\theta_{\mathbf{u}} \leftarrow \theta_{\mathbf{u}} - \eta \nabla_{\theta_{\mathbf{u}}} L_{CE}$
  **end for**
**end for**
**return** $\theta_{\mathbf{u}} = \{\theta_{\mathbf{u,enc}}, \theta_{\mathbf{u,head}}\}$

---

we examined the outputs of the final three transformer encoder blocks. Note that these selections of layers is based on the observation that the information differences of outputs from the initial layers of both original and retrained models are similar. For empirical justifications of these selections, please refer to Appendix E.2.

### C.8 SYSTEM SPECIFICATION

For fair comparison, all experiments are executed in Python 3.10, on an Ubuntu 18.04 machine with 72 CPU cores, 4 Nvidia RTX A6000 GPUs and 512GB memory.

## D ADDITIONAL UNLEARNING RESULTS

In this section, we provide the full experiment results on various machine unlearning settings, extending the results in Section 3 and Section 5.2.

### D.1 HEAD DISTILLATION (HD) RESULTS

By achieving strong black-box performance while modifying only the model head and retaining intermediate layer information (*i.e.*, preserving the information of the forget samples), we highlighted the limitations of black-box metrics in Section 3. To examine whether HD consistently exposes these limitations across diverse metrics (e.g., Accuracy, MIA, JSD, ZRF (Chundawat et al., 2023b), AUS (Cotogni et al., 2023), LiRA MIA (Carlini et al., 2022)) and scenarios (single-class, multi-class, and random data forgetting), we extend our analysis to include a broader range of evaluations. We compare HD with five other methods: FT, RL, GA, $\ell_1$-sparse, and SALUN.

**Single-Class Forgetting.** Figure 10 presents the results for single-class forgetting on CIFAR-10. Despite modifying only the last layer, HD achieves the best MIA performance among the five methods. Additionally, it delivers competitive results across accuracy, JSD, ZRF, and AUS metrics, completing the task in under ten seconds (RTE). Notably, HD also performs comparably on LiRA MIA (Kurmanji et al., 2023; Hayes et al., 2024), one of a recently proposed black-box MIA metrics. The strong

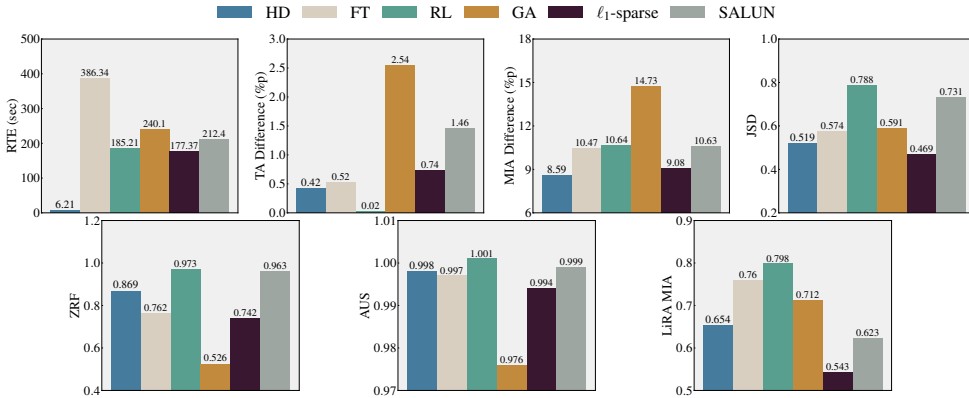

Figure 10: Performance of six methods (HD, FT, RL, GA, $\ell_1$-sparse, SALUN) on (CIFAR-10, ResNet-18), evaluated in efficiency (RTE), accuracy (TA), and efficacy (MIA, JSD, ZRF, AUS, LiRA MIA) in single-class forgetting scenarios. Lower differences from Retrain in TA, MIA, and JSD indicate closer similarity to Retrain, while higher values for ZRF and AUS represent better efficacy. Additionally, LiRA MIA values closer to 0.5 reflect higher efficacy.

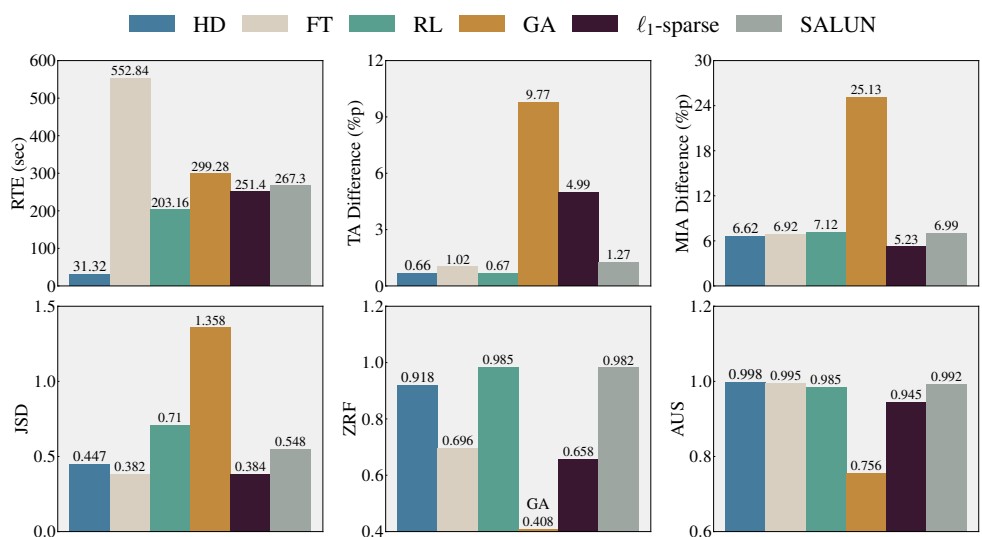

Figure 11: Performance of six methods (HD, FT, RL, GA, $\ell_1$-sparse, SALUN) on (CIFAR-100, ResNet-18), evaluated in efficiency (RTE), accuracy (TA), and efficacy (MIA, JSD, ZRF, AUS) in multi-class forgetting scenarios (5 classes). For TA, MIA, and JSD, lower differences from Retrain are preferred, indicating similarity to Retrain. For ZRF and AUS, higher values reflect better efficacy.

performance of HD across various black-box metrics underscores the need for robust white-box metrics to more effectively assess unlearning quality.

**Multi-Class Forgetting.** For the multi-class forgetting scenario, we extend the logit masking technique of HD, as described in Section 3, by incorporating additional masking for multiple classes. As shown in Figures 11 and 12, HD achieves comparable performance across TA, MIA, JSD, ZRF, and AUS, completing the tasks within the shortest time frame (31.32 seconds for forgetting 5 classes and 25.83 seconds for forgetting 20 classes). Similar to the single-class forgetting scenario, HD's comparable performance in this extended setup further highlights the limitations of black-box metrics.

**Random Data Forgetting.** In random data forgetting scenarios, HD cannot be directly applied, as all classes are included in the retain set. To address this, we use gradient descent on the retain set and

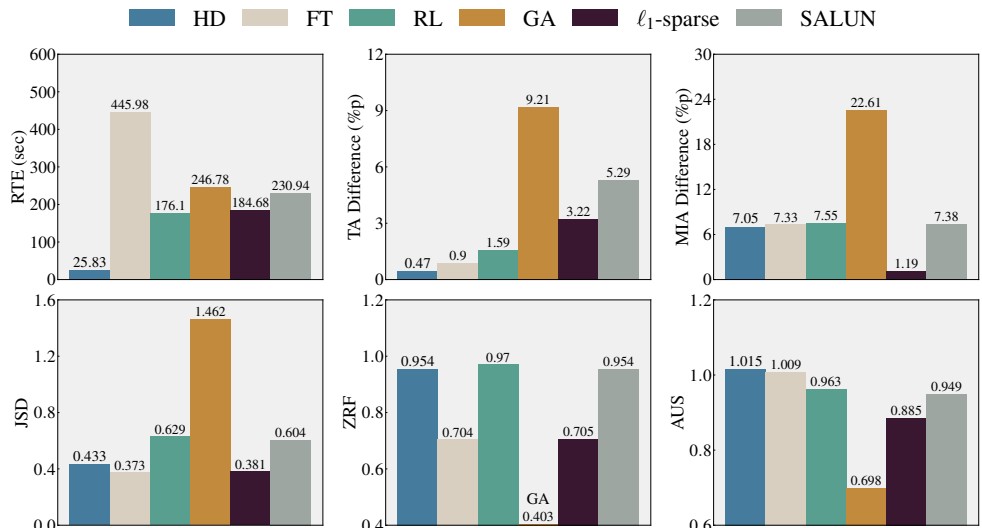

Figure 12: Performance of six methods (HD, FT, RL, GA, $\ell_1$-sparse, SALUN) on (CIFAR-100, ResNet-18), evaluated in efficiency (RTE), accuracy (TA), and efficacy (MIA, JSD, ZRF, AUS) in multi-class forgetting scenarios (20 classes). For TA, MIA, and JSD, lower differences from Retrain are preferred, indicating similarity to Retrain. For ZRF and AUS, higher values reflect better efficacy.

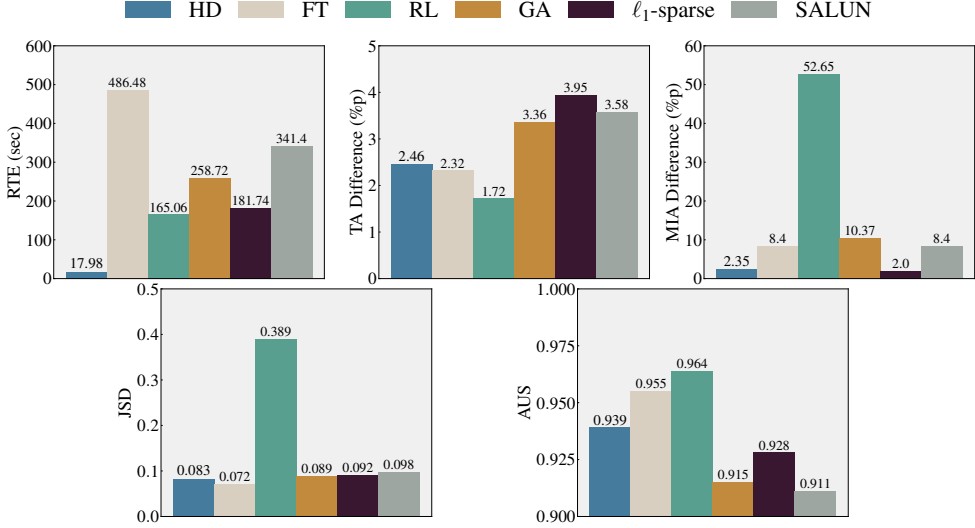

Figure 13: Performance of six methods (HD, FT, RL, GA, $\ell_1$-sparse, SALUN) on (CIFAR-10, ResNet-18), evaluated in efficiency (RTE), accuracy (TA), and efficacy (MIA, JSD, AUS) in random data forgetting scenarios (500 samples per class). For TA, MIA, and JSD, lower differences from Retrain are preferred, indicating similarity to Retrain. For AUS, higher values reflect better efficacy.

gradient ascent on the forget set while training only the model's head. While this approach differs from HD, which uses logit masking, we retain the name to emphasize its defining characteristic of modifying only the last layer. As shown in Figure 13, HD achieves strong performance on black-box metrics (accuracy, MIA, and JSD) within less than 20 seconds (RTE). This result demonstrates that HD can still deceive black-box metrics with comparable performance. Note that ZRF is omitted in this scenario, as it is not ideally suited for random data forgetting; for more details, refer to (Chundawat et al., 2023b).

## D.2 STANDARD DEVIATION

Due to the visual complexity of Figures 5 and 8, representing the standard deviation directly in these figures is challenging. Therefore, we include the standard deviation values separately in Tables 16 and 17 for Figure 5, and in Table 18 for Figure 8.

## D.3 SINGLE-CLASS FORGETTING RESULTS

**CIFAR-10 with Various Architectures**   Table 11 shows the full experiment results of the single-class forgetting experiments from Table 1 on CIFAR-10 using different models. This extended table includes Jensen-Shannon divergence (JSD) and the unlearning results with ResNet-50 and ViT. Although many baselines show promising results on output based evaluation metrics, they generally exhibit poor feature-level unlearning. In contrast, COLA not only outperforms existing baselines in IDI but also shows decent results in other metrics, demonstrating its effectiveness in removing the influence of the forget set within the encoder of the model.

**CIFAR-100 with Various Architectures**   As demonstrated in Table 12, we also compare COLA with other baselines on CIFAR-100. The results consistently highlight the difficulty of comparing and validating the efficacy of each unlearning method using existing output-based metrics. With the help of IDI, it is clear that COLA shows robustness in model unlearning on datasets with a large number of classes across various model architectures. Although SCRUB has achieved IDI near 0 for the CIFAR-10 ResNet-18 experiment, it shows significant variations in feature-level unlearning across different datasets and architectures.

## D.4 MULTI-CLASS FORGETTING RESULTS

**Multi-Class Forgetting on CIFAR-10 and CIFAR-100**   Table 13 presents the results of multi-class forgetting experiments on CIFAR-10 and CIFAR-100 using ResNet-18, which involves erasing the information of more than one class in the training set. We remove two classes from CIFAR-10 and five and twenty classes from CIFAR-100. Notably, many baselines exhibit higher IDI values as the number of forgetting class increases, demonstrating that the tendency to modify the head of the model strengthens with the difficulty of the unlearning tasks. In contrast, COLA shows remarkable effectiveness, achieving metric values closely aligned with Retrain. Specifically, COLA consistently achieves the lowest IDI values among the evaluated methods, indicating the necessity of the collapse phase for effective feature-level unlearning no matter the number of class to forget.

**Multi-Class Forgetting on ImageNet-1K**   We conduct 5-class unlearning on ImageNet-1K using ResNet-50 and ViT. Table 14 provides the complete results of Table 1 for ImageNet-1K, including all evaluation metrics and outcomes on the ViT architecture. However, it is important to note that IDI alone should not be used to assess unlearned models, as a low IDI might indicate a loss of overall information, including that from the retain set, which should be maintained at the same level as Original. This issue is evident in the RA, TA, and IDI of EU-10 and CF-10 in Table 14. In contrast, COLA consistently achieves IDI near 0 while maintaining accuracy measurements comparable to Retrain, demonstrating the scalability of our framework to the large-scale datasets.

## D.5 RANDOM DATA FORGETTING RESULTS

Table 15 presents the results of the random data forgetting task conducted on ResNet-18. For CIFAR-10 and CIFAR-100 datasets, we randomly selected 500 and 50 forget samples per class, respectively. COLA+, which incorporates pseudo-labeling, successfully eliminates the influence of the forgetting data while maintaining competitive performance.

# E ADDITIONAL DISCUSSIONS

## E.1 MUTUAL INFORMATION CURVES

Figure 18 illustrates the estimated mutual information $I(\mathbf{Z}_\ell; Y)$ of the features from the $\ell$-th layer $\mathbf{Z}_\ell$ and the binary label $Y$, computed by the InfoNCE loss across various architectures and datasets. We

Table 5: IDI values of methods on ResNet-18 with CIFAR-10 singleclass forgetting, computed using the last $n$ selected layers, where $n = 1$ considers only the final representation, and larger $n$ incrementally include earlier layers. $\star$ marks the IDI values reported in our work.

| Methods | Full Layers | $n = 4$ | $n = 3$ | $n = 2^{\star}$ | $n = 1$ |
|---|---|---|---|---|---|
| FT | $0.670_{\pm 0.011}$ | $0.670_{\pm 0.011}$ | $0.670_{\pm 0.013}$ | $0.671_{\pm 0.008}$ | $0.673_{\pm 0.012}$ |
| RL | $0.833_{\pm 0.005}$ | $0.833_{\pm 0.004}$ | $0.830_{\pm 0.004}$ | $0.830_{\pm 0.005}$ | $0.837_{\pm 0.002}$ |
| GA | $0.338_{\pm 0.006}$ | $0.336_{\pm 0.007}$ | $0.334_{\pm 0.007}$ | $0.334_{\pm 0.014}$ | $0.333_{\pm 0.008}$ |
| Bad-T | $1.020_{\pm 0.023}$ | $1.016_{\pm 0.023}$ | $1.012_{\pm 0.023}$ | $1.014_{\pm 0.004}$ | $1.016_{\pm 0.022}$ |
| EU-5 | $0.531_{\pm 0.004}$ | $0.531_{\pm 0.005}$ | $0.530_{\pm 0.006}$ | $0.528_{\pm 0.005}$ | $0.524_{\pm 0.007}$ |
| CF-5 | $0.674_{\pm 0.023}$ | $0.675_{\pm 0.023}$ | $0.673_{\pm 0.024}$ | $0.675_{\pm 0.027}$ | $0.679_{\pm 0.026}$ |
| EU-10 | $-0.352_{\pm 0.007}$ | $-0.347_{\pm 0.007}$ | $-0.344_{\pm 0.006}$ | $-0.349_{\pm 0.019}$ | $-0.310_{\pm 0.018}$ |
| CF-10 | $-0.058_{\pm 0.010}$ | $-0.060_{\pm 0.010}$ | $-0.061_{\pm 0.010}$ | $-0.060_{\pm 0.017}$ | $-0.056_{\pm 0.008}$ |
| SCRUB | $-0.055_{\pm 0.028}$ | $-0.053_{\pm 0.029}$ | $-0.051_{\pm 0.027}$ | $-0.056_{\pm 0.008}$ | $-0.048_{\pm 0.026}$ |
| SALUN | $0.941_{\pm 0.029}$ | $0.937_{\pm 0.029}$ | $0.935_{\pm 0.029}$ | $0.936_{\pm 0.012}$ | $0.935_{\pm 0.027}$ |
| l1-sparse | $0.292_{\pm 0.011}$ | $0.293_{\pm 0.012}$ | $0.294_{\pm 0.011}$ | $0.293_{\pm 0.012}$ | $0.297_{\pm 0.013}$ |
| COLA | $0.010_{\pm 0.009}$ | $0.012_{\pm 0.009}$ | $0.012_{\pm 0.008}$ | $0.010_{\pm 0.006}$ | $0.015_{\pm 0.013}$ |

Table 6: IDI values of methods on ResNet-50 with CIFAR-10 single class forgetting, computed using the last $n$ selected layers, where $n = 1$ considers only the final representation, and larger $n$ incrementally include earlier layers. $\star$ marks the IDI values reported in our work.

| Methods | Full Layers | $n = 5$ | $n = 4$ | $n = 3^{\star}$ | $n = 2$ | $n = 1$ |
|---|---|---|---|---|---|---|
| FT | $0.617_{\pm 0.006}$ | $0.618_{\pm 0.009}$ | $0.618_{\pm 0.013}$ | $0.607_{\pm 0.009}$ | $0.610_{\pm 0.013}$ | $0.563_{\pm 0.014}$ |
| RL | $0.808_{\pm 0.012}$ | $0.808_{\pm 0.012}$ | $0.811_{\pm 0.006}$ | $0.804_{\pm 0.006}$ | $0.814_{\pm 0.003}$ | $0.797_{\pm 0.000}$ |
| GA | $0.334_{\pm 0.018}$ | $0.338_{\pm 0.018}$ | $0.337_{\pm 0.018}$ | $0.334_{\pm 0.023}$ | $0.339_{\pm 0.017}$ | $0.269_{\pm 0.015}$ |
| Bad-T | $1.156_{\pm 0.016}$ | $1.151_{\pm 0.020}$ | $1.152_{\pm 0.021}$ | $1.153_{\pm 0.026}$ | $1.157_{\pm 0.018}$ | $1.163_{\pm 0.024}$ |
| EU-5 | $1.044_{\pm 0.009}$ | $1.043_{\pm 0.008}$ | $1.050_{\pm 0.005}$ | $1.047_{\pm 0.005}$ | $1.061_{\pm 0.002}$ | $1.080_{\pm 0.002}$ |
| CF-5 | $0.904_{\pm 0.005}$ | $0.906_{\pm 0.005}$ | $0.910_{\pm 0.006}$ | $0.906_{\pm 0.002}$ | $0.916_{\pm 0.001}$ | $0.914_{\pm 0.002}$ |
| EU-10 | $0.760_{\pm 0.014}$ | $0.766_{\pm 0.011}$ | $0.766_{\pm 0.011}$ | $0.757_{\pm 0.011}$ | $0.756_{\pm 0.010}$ | $0.715_{\pm 0.010}$ |
| CF-10 | $0.592_{\pm 0.015}$ | $0.594_{\pm 0.017}$ | $0.590_{\pm 0.018}$ | $0.579_{\pm 0.009}$ | $0.582_{\pm 0.018}$ | $0.516_{\pm 0.024}$ |
| SCRUB | $0.067_{\pm 0.005}$ | $0.073_{\pm 0.007}$ | $0.071_{\pm 0.007}$ | $0.067_{\pm 0.020}$ | $0.076_{\pm 0.008}$ | $0.011_{\pm 0.005}$ |
| SALUN | $0.831_{\pm 0.014}$ | $0.833_{\pm 0.011}$ | $0.832_{\pm 0.019}$ | $0.832_{\pm 0.027}$ | $0.842_{\pm 0.009}$ | $0.771_{\pm 0.005}$ |
| l1-sparse | $0.185_{\pm 0.005}$ | $0.183_{\pm 0.007}$ | $0.181_{\pm 0.007}$ | $0.184_{\pm 0.023}$ | $0.185_{\pm 0.016}$ | $0.191_{\pm 0.007}$ |
| COLA | $0.019_{\pm 0.006}$ | $0.023_{\pm 0.009}$ | $0.021_{\pm 0.009}$ | $0.019_{\pm 0.025}$ | $0.022_{\pm 0.009}$ | $0.007_{\pm 0.011}$ |

compute mutual information (MI) for all layers from the ResNet encoder and last five layers from the ViT encoder based single-class forgetting retain and forget sets. The upper bound of MI is given by the entropy $H(Y) \geq I(\mathbf{Z}_{\ell}; Y) = H(Y) - H(Y \mid \mathbf{Z}_{\ell})$. The estimated MI values fall within the range of the upper and lower bounds (0), validating the use of InfoNCE for MI estimation. Notably, all MI curves consistently show a larger difference between Original and Retrain in the later layers of the encoder across various datasets and architectures, while differences are minimal in the earlier layers. These observations underscore the validity of computing the information difference (ID) for the last few layers to quantify unlearning. Furthermore, the difference between Original and Retrain becomes more significant with increasing numbers of forget classes, as shown in Figure 19.

## E.2 EFFECT OF NUMBER OF LAYERS FOR IDI

Conceptually, estimating mutual information for IDI involves all intermediate layers, as introduced in Section 4.3. However, in practice, earlier layers exhibit similar mutual information levels across models, as shown in Figure 5, Figure 18, and Figure 19. Consequently, estimating mutual information from only a few later layers is sufficient for evaluation. This observation aligns with findings in Yosinski et al. (2014); Zeiler and Fergus (2014), which indicate that earlier layers primarily capture general features, while later layers focus on distinctive features, resulting in greater variability in mutual information. To validate this approach, we measure IDI using different numbers of accumulated layers from the back, as presented in Table 5 and Table 6. These experiments use the same settings discussed in Figure 5. Our results demonstrate minimal differences in IDI as $n$ increases, indicating a negligible contribution of earlier layers to IDI. Specifically, when comparing the two columns ("Full layers" and "$n$ with $\star$"), the discrepancy between the ideal IDI and our practical approach is minimal, empirically supporting the validity of focusing on the last selected layers. This property is particularly beneficial for reducing computational costs, as mutual information

Table 7: IDI for four different reference models (Retrain, COLA, EU-10, and FT$^\star$). FT$^\star$ is finetuned with a learning rate of 5e-5, while FT is finetuned with a learning rate of 1e-5. Since FT typically does not remove all residual information while maintaining test accuracy, using a higher learning rate for FT$^\star$ can be justified if you want to use it as the reference model. The 'Order' has been arranged in ascending sequence according to the IDI values.

| | CIFAR-10 | | | | | CIFAR-100 | | | |
|---|---|---|---|---|---|---|---|---|---|
| Methods | Order | $\theta_s$ = Retrain | $\theta_s$ = COLA | $\theta_s$ = EU-10 | $\theta_s$ = FT$^\star$ | Order | $\theta_s$ = Retrain | $\theta_s$ = COLA | $\theta_s$ = EU-10 | $\theta_s$ = FT$^\star$ |
| FT | 8 | $0.671_{\pm0.008}$ | $0.668_{\pm0.008}$ | $0.756_{\pm0.006}$ | $0.662_{\pm0.008}$ | 11 | $0.610_{\pm0.022}$ | $0.624_{\pm0.021}$ | $0.680_{\pm0.018}$ | $0.481_{\pm0.029}$ |
| RL | 10 | $0.830_{\pm0.005}$ | $0.828_{\pm0.005}$ | $0.874_{\pm0.004}$ | $0.825_{\pm0.005}$ | 9 | $0.467_{\pm0.010}$ | $0.486_{\pm0.010}$ | $0.563_{\pm0.008}$ | $0.291_{\pm0.013}$ |
| GA | 6 | $0.334_{\pm0.014}$ | $0.328_{\pm0.014}$ | $0.506_{\pm0.010}$ | $0.315_{\pm0.014}$ | 8 | $0.392_{\pm0.021}$ | $0.414_{\pm0.020}$ | $0.502_{\pm0.017}$ | $0.191_{\pm0.028}$ |
| Bad-T | 12 | $1.014_{\pm0.004}$ | $1.014_{\pm0.004}$ | $1.010_{\pm0.003}$ | $1.014_{\pm0.004}$ | 12 | $1.079_{\pm0.024}$ | $1.076_{\pm0.023}$ | $1.065_{\pm0.020}$ | $1.105_{\pm0.032}$ |
| EU-5 | 7 | $0.528_{\pm0.005}$ | $0.523_{\pm0.005}$ | $0.650_{\pm0.004}$ | $0.515_{\pm0.005}$ | 3 | $0.064_{\pm0.037}$ | $0.098_{\pm0.036}$ | $0.233_{\pm0.030}$ | $-0.245_{\pm0.049}$ |
| CF-5 | 9 | $0.675_{\pm0.027}$ | $0.672_{\pm0.027}$ | $0.759_{\pm0.020}$ | $0.666_{\pm0.028}$ | 7 | $0.388_{\pm0.010}$ | $0.410_{\pm0.010}$ | $0.499_{\pm0.008}$ | $0.186_{\pm0.013}$ |
| EU-10 | 1 | $-0.349_{\pm0.019}$ | $-0.362_{\pm0.019}$ | $0.0_{\pm0.014}$ | $-0.387_{\pm0.020}$ | 1 | $-0.221_{\pm0.009}$ | $-0.177_{\pm0.009}$ | $0.0_{\pm0.007}$ | $-0.624_{\pm0.012}$ |
| CF-10 | 2 | $-0.060_{\pm0.017}$ | $-0.070_{\pm0.017}$ | $0.214_{\pm0.013}$ | $-0.090_{\pm0.017}$ | 4 | $0.175_{\pm0.040}$ | $0.205_{\pm0.039}$ | $0.324_{\pm0.033}$ | $-0.097_{\pm0.053}$ |
| SCRUB | 3 | $-0.056_{\pm0.008}$ | $-0.066_{\pm0.008}$ | $0.217_{\pm0.006}$ | $-0.086_{\pm0.008}$ | 6 | $0.339_{\pm0.069}$ | $0.363_{\pm0.067}$ | $0.458_{\pm0.057}$ | $0.121_{\pm0.092}$ |
| SALUN | 11 | $0.936_{\pm0.012}$ | $0.935_{\pm0.012}$ | $0.953_{\pm0.009}$ | $0.934_{\pm0.012}$ | 10 | $0.529_{\pm0.022}$ | $0.546_{\pm0.022}$ | $0.614_{\pm0.018}$ | $0.373_{\pm0.029}$ |
| $\ell_1$-sparse | 5 | $0.293_{\pm0.012}$ | $0.286_{\pm0.012}$ | $0.476_{\pm0.012}$ | $0.273_{\pm0.012}$ | 5 | $0.334_{\pm0.026}$ | $0.358_{\pm0.025}$ | $0.454_{\pm0.021}$ | $0.114_{\pm0.035}$ |
| COLA | 4 | $0.010_{\pm0.006}$ | $0.0_{\pm0.006}$ | $0.266_{\pm0.004}$ | $-0.018_{\pm0.006}$ | 2 | $-0.038_{\pm0.006}$ | $0.0_{\pm0.006}$ | $0.150_{\pm0.005}$ | $-0.381_{\pm0.008}$ |

| IDI: **1.000** | IDI: **0.906±0.002** | IDI: **0.607±0.009** | IDI: **0.184±0.023** | IDI: **0.019±0.025** | IDI: **0.000** |
|---|---|---|---|---|---|
| (a) Original | (b) CF-5 | (c) FT | (d) $\ell_1$-sparse | (e) COLA | (f) Retrain |

Figure 14: t-SNE visualizations of encoder outputs for Original, Retrain, and unlearned models from four MU methods (SALUN, $\ell_1$-sparse, SCRUB, EU-10) on single-class forgetting with (CIFAR-10, ResNet-50). In each t-SNE plot, features of the forgetting class are represented in purple.

computations for later layers require less time due to the shallower $g$ networks involved (refer to Section 4.1). Practitioners can determine the appropriate $n$ by observing the information gap per layer between Original and Retrain for a given unlearning setup.

### E.3 IDI without Retrain Model

In real-world applications, using the Retrain is often infeasible. In such cases, any reasonable model can be used as the **standard model**, denoted as $\theta_s$. Although the absence of a Retrain inevitably affects how the IDI value is interpreted (*i.e.*, an IDI value of zero indicates that unlearning has been properly achieved, equivalent to the Retrain), it still provides useful insights into the degree of unlearning achieved relative to the chosen reference. To accommodate this, we introduce an extended version of the ID metric. Differences are highlighted with $(\cdot)$:

$$\mathbf{ID}(\theta_u, \theta_s) = \sum_{\ell=1}^{L} \left( I(\mathbf{Z}_\ell^{(u)}; Y_C) - I(\mathbf{Z}_\ell^{(s)}; Y_C) \right). \tag{8}$$

The main difference from the original ID is that $\theta_s$ can be set as any model including $\theta_r$ (Retrain), while the previous version fixed $\theta_s = \theta_r$. This extension also leads to the modified IDI metric:

$$\mathbf{IDI}(\theta_u, \theta_s) = \frac{\mathbf{ID}(\theta_u, \theta_s)}{\mathbf{ID}(\theta_o, \theta_s)} = \frac{\sum_{\ell=1}^{L} \left( I(\mathbf{Z}_\ell^{(u)}; Y) - I(\mathbf{Z}_\ell^{(s)}; Y) \right)}{\sum_{\ell=1}^{L} \left( I(\mathbf{Z}_\ell^{(o)}; Y) - I(\mathbf{Z}_\ell^{(s)}; Y) \right)}. \tag{9}$$

We test IDI using different reference models, as demonstrated in Table 7. Intuitively, since only the standard model changes in Equation (9), the order of the IDI values remains consistent.

### E.4 IDI and t-SNE Relationship

In Section 3.2, we identified significant residual information in unlearned models through their tightly clustered t-SNE plots (see Figure 3) and their ability to easily recover forgotten information (see Figure 4). Black-box assessments failed to detect these residuals, as shown by the success of HD (see Figure 2), which only altered the last layer. In contrast, IDI effectively captures these hidden residuals, showing a strong correlation with t-SNE plots (see Figure 14), and aligning with accuracy

recovered across unlearned models (see Figure 4). By complementing existing metrics, IDI offers a comprehensive evaluation of approximate MU methods, addressing crucial aspects to ensure strong unlearning beyond superficial modifications.

Figure 15 presents the full t-SNE plots illustrating the intermediate features and corresponding IDI measurements of MU baselines on the single-class unlearning on CIFAR-10 with ResNet-18. A high IDI corresponds to better clustering and similarity among features of the forgetting class, as seen in (l) SALUN and (f) Bad-T, which show inadequate unlearning performance. These examinations show the high relationship between IDI and the residual information of forget set. Additionally, the IDI metric reveals instances of over-unlearning, where the forgetting class becomes excessively dispersed, as demonstrated in (i) EU-10. Among the evaluated methods, (n) COLA has the closest IDI to Retrain, suggesting its high efficacy in achieving the desired removal of the forget set influence in the intermediate layers of the model. This trend is also visible in ResNet-50 (see Figure 16) and ViT (see Figure 17).

Furthermore, we confirm that IDI for the random data forgetting correctly captures the encoder's information, similar to IDI for the class-wise forgetting. In Figure 21, the t-SNE plots of forget sample features for two baselines with the same unlearning accuracy (UA) – Bad-T and $\ell_1$-sparse – and their IDI values in the random data forgetting task is visualized. Comparing them, IDI successfully reflects the residual information in the features, as the features of Bad-T form more compact clusters than those of $\ell_1$-sparse, indicating more influence of the forget set remains in Bad-T. IDI the for random data forgetting captures the hidden information that cannot be noticed from existing metrics, which may suggest that both methods unlearn similarly due to their same forget accuracy.

### E.5 MUTUAL INFORMATION AND ACCURACY

We extend the experiment to measure the accuracy of the intermediate features of the model's encoder. Similar to measuring MI using the InfoNCE loss, we freeze the layers up to the $\ell$-th layer of the encoder and train the remaining encoder layers and an additional head using cross-entropy loss. The additional head perform binary classification to determine whether the input belongs to the retain or forget set.

Figure 20 shows the train accuracy curves on the CIFAR-10 single-class forgetting dataset with ResNet-18. For Original encoder, the trained model readily classifies the retain and forget sets. However, for Retrain encoder, the model fails to classifies all samples at the last two layers, with the accuracy dropping more in the later layer. These curves correspond to the those from Figure 18, indicating that the estimated MI accurately reflects the model's knowledge of the retain and forget sets. In addition, the small accuracy gap between Original and Retrain provides the necessity of MI for accurate residual information quantification.

### E.6 COMPUTATIONAL COMPLEXITY OF IDI

Table 8 presents the runtime of mutual information (MI) computation for intermediate features from each block, using the MI estimation method proposed in Section 4.1, in the CIFAR-100 single-class forgetting setup with ResNet-18, ResNet-50, and ViT.

Although MI estimation across all layers can be time-consuming, our selected layers for IDI computation (*i.e.*, features from the last two blocks for ResNet-18 and the last three blocks for ResNet-50 and ViT, as detailed in Appendix C.7, and empirically justified in Appendix E.2) significantly reduce runtime without harming metric performance. Specifically, the runtime decreases by factors of 2.63, 2.18, and 4.30 for ResNet-18, ResNet-50, and ViT, respectively, when the estimation is sequentially processed for each block. Furthermore, using only 10% of the retain set improves runtime by an additional 4 to 5 times without affecting the general trend, as shown in Figure 8. Since training the latter layers requires fewer FLOPs compared to earlier layers, the computational complexity of IDI is further reduced. These techniques can effectively alleviate potential computational challenges when applying our metric in practice.

Table 8: Runtime (in minutes) for mutual information estimation at the final layer of each block. A 'block' refers to a group of residual layers in ResNet (commonly referred to as stages, with four blocks in ResNet-18) or a transformer block in ViT. Results are presented for evaluations conducted using 10% of the retain set and the full dataset in the CIFAR-10 single-class forgetting scenario.

| | ResNet-18 | | | | | ResNet-50 | | | | | |
|---|---|---|---|---|---|---|---|---|---|---|---|
| Ratios | Block1 | Block2 | Block3 | Block4 | Block5 | Ratios | Block1 | Block2 | Block3 | Block4 | Block5 | Block6 |
| 10% | $1.61_{\pm0.05}$ | $1.55_{\pm0.21}$ | $1.63_{\pm0.09}$ | $1.49_{\pm0.11}$ | $1.42_{\pm0.13}$ | 10% | $4.17_{\pm0.13}$ | $3.85_{\pm0.04}$ | $3.43_{\pm0.01}$ | $3.42_{\pm0.15}$ | $3.35_{\pm0.02}$ | $3.24_{\pm0.15}$ |
| Full | $6.64_{\pm0.08}$ | $6.51_{\pm0.17}$ | $6.32_{\pm0.12}$ | $6.04_{\pm0.10}$ | $5.90_{\pm0.15}$ | Full | $19.91_{\pm0.18}$ | $17.70_{\pm0.01}$ | $15.75_{\pm0.02}$ | $15.32_{\pm0.11}$ | $14.98_{\pm0.07}$ | $14.83_{\pm0.04}$ |

| | ViT | | | | | | | | | | | |
|---|---|---|---|---|---|---|---|---|---|---|---|---|
| Ratios | Block1 | Block2 | Block3 | Block4 | Block5 | Block6 | Block7 | Block8 | Block9 | Block10 | Block11 | Block12 |
| 10% | 32.46 | 32.12 | 31.43 | 30.99 | 30.64 | 30.21 | 29.50 | 29.01 | 28.75 | 28.43 | 27.95 | 27.14 |
| Full | 167.00 | 162.81 | 160.64 | 159.56 | 157.65 | 155.35 | 151.82 | 147.78 | 146.05 | 144.25 | 142.81 | 139.61 |

## F  BROADER IMPACT

Our work on improving machine unlearning focuses on foundational research aimed at enhancing privacy and data removal. However, there is a potential risk that our methodology could be misused to evade data retention policies or obscure accountability. Despite this possibility, it is unlikely that our work will introduce new harmful practices beyond what existing unlearning methods already permit, as we are not introducing new capabilities. Therefore, while there might be concerns related to privacy, security, and fairness, our work does not pose a greater risk compared to other foundational research in machine unlearning.

## G  LIMITATIONS

Our methodology accomplishes its main objective, but there are a few limitations we point out. Although our IDI successfully investigates hidden information in intermediate features, its computation requires multiple training runs, which can be computationally intensive. For instance, The computation of IDI for ResNet-50 on the CIFAR-100 dataset takes approximately 40-50 minutes. However, one can mitigate this by computing mutual information for only the last few layers, as the early stages of the encoder are largely similar for both the Retrain and Original models. Thus, this approach requires fine-tuning only the later layers, reducing the overall computational burden. Additionally, by adjusting the forget-to-retain ratio, it is possible to improve efficiency and possibly decrease the processing time to merely 3-4 minutes.

Table 9: Hyperparameters of baselines for *class-wise forgetting*. Retain Batch Size is the batch size of retain set $\mathcal{D}_r$ and Forget Batch Size is the batch size of forget set $\mathcal{D}_f$. Baselines without Forget Batch Size imply that they do not use forget set $\mathcal{D}_f$. Bad-T uses the entire dataset $\mathcal{D}$, so there is no separation of retain and forget of Batch Size. SCRUB has separate epochs for retain set and forget set, which is visualized as Retain Epochs (Forget Epochs). For COLA, A + B Epochs indicates collapse epochs A and align epochs B.

| Class-wise Forgetting | | | |
|---|---|---|---|
| Settings | CIFAR-10
ResNet-18 / ResNet-50 / ViT | CIFAR-100
ResNet-18 / ResNet-50 / ViT | ImageNet-1K
ResNet-50 / ViT |
| FT | 25 Epochs, Adam
LR $10^{-5}$/$10^{-5}$/$10^{-4}$
Retain Batch Size 64 | | 3/4 Epochs, Adam
LR $10^{-5}$
Retain Batch Size 128 |
| RL | 7 / 7 / 10 Epochs, SGD
LR $10^{-5}$ / $2 \cdot 10^{-5}$ / $10^{-3}$
Retain Batch Size 64
Forget Batch Size 16 | 7 / 7 / 10 Epochs, SGD
LR $2 \cdot 10^{-5}$ / $10^{-4}$ / $10^{-4}$
Retain Batch Size 64
Forget Batch Size 16 | 3 Epochs, SGD
LR $10^{-3}$ / $10^{-4}$
Retain Batch Size 128
Forget Batch Size 16 |
| GA | 10 Epochs, SGD
LR $2 \cdot 10^{-3}$ / $2 \cdot 10^{-3}$ / $5 \cdot 10^{-3}$
Retain Batch Size 64
Forget Batch Size 16 | 10 Epochs, SGD
LR $9 \cdot 10^{-4}$ / $9 \cdot 10^{-4}$ / $5 \cdot 10^{-3}$
Retain Batch Size 64
Forget Batch Size 16 | 3 Epochs, SGD
LR $2 \cdot 10^{-3}$ / $10^{-3}$
Retain Batch Size 128
Forget Batch Size 16 |
| Bad-T | 10 Epochs, Adam
LR $10^{-5}$
Batch Size 256 | | 3 Epochs, Adam
LR $10^{-5}$
Batch Size 256 |
| Boundary Expand /
Boundary Shrink | 10 Epochs, SGD
LR $10^{-5}$
Forget Batch Size 64 | | |
| EU-5 /
EU-10 | 14 Epochs, SGD
LR $10^{-2}$
Retain Batch Size 64 | | 2 Epochs, SGD
LR $5 \cdot 10^{-3}$
Retain Batch Size 128 |
| CF-5 /
CF-10 | 14 / 14 / 18 Epochs, SGD
LR $10^{-2}$ / $10^{-2}$ / $3 \cdot 10^{-2}$
Retain Batch Size 64 | | 5 Epochs, SGD
LR $5 \cdot 10^{-3}$
Retain Batch Size 128 |
| SCRUB | 3(2) Epochs, SGD
LR $5 \cdot 10^{-4}$ / $5 \cdot 10^{-4}$ / $10^{-4}$
Retain Batch Size 64
Forget Batch Size 256 / 256 / 64 | 3(2) Epochs, SGD
LR $5 \cdot 10^{-4}$
Retain Batch Size 128
Forget Batch Size 8 | 2(2) Epochs, SGD
LR $5 \cdot 10^{-4}$ / $10^{-4}$
Retain Batch Size 128
Forget Batch Size 256 |
| SALUN | 10 Epochs, SGD
LR $5 \cdot 10^{-4}$ / $10^{-3}$ / $10^{-3}$
Retain Batch Size 64
Forget Batch Size 16 | 15 Epochs, SGD
LR $10^{-3}$
Retain Batch Size 64
Forget Batch Size 16 | 5/2 Epochs, SGD
LR $10^{-3}$
Retain Batch Size 128
Forget Batch Size 16 |
| $\ell_1$-sparse | 10 Epochs, SGD
LR $2 \cdot 10^{-4}$ / $2 \cdot 10^{-4}$ / $9 \cdot 10^{-4}$
Retain Batch Size 64 | 10 Epochs, SGD
LR $2 \cdot 10^{-4}$ / $2 \cdot 10^{-4}$ / $5 \cdot 10^{-4}$
Retain Batch Size 64 | 5 Epochs, SGD
LR $9 \cdot 10^{-4}$
Retain Batch Size 128 |
| COLA | 10+10 Epochs, Adam
Contrast LR $2 \cdot 10^{-4}$ / $2 \cdot 10^{-4}$ / $1.5 \cdot 10^{-4}$
Finetune LR $5 \cdot 10^{-6}$ / $10^{-5}$ / $5 \cdot 10^{-5}$
Retain Batch Size 64 | 10+10 Epochs, Adam
Contrast LR $5 \cdot 10^{-4}$ / $5 \cdot 10^{-4}$ / $5 \cdot 10^{-4}$
Finetune LR $5 \cdot 10^{-6}$ / $10^{-5}$ / $5 \cdot 10^{-5}$
Retain Batch Size 256 | 1+2 Epochs, Adam
Contrast LR $2 \cdot 10^{-5}$ / $5 \cdot 10^{-5}$
Finetune LR $1 \cdot 10^{-5}$ / $5 \cdot 10^{-5}$
Retain Batch Size 256 |

Table 10: Hyperparameters of baselines for *random data forgetting*. Retain Batch Size is the batch size of retain set $\mathcal{D}_r$ and Forget Batch Size is the batch size of forget set $\mathcal{D}_f$. Baselines without Forget Batch Size imply that they do not use forget set $\mathcal{D}_f$. Bad-T uses the entire dataset $\mathcal{D}$, so there is no separation of retain and forget of Batch Size. SCRUB uses separate epochs for retain set and forget set, which is visualized as Retain Epochs (Forget Epochs). For COLA+, A + B Epochs indicates collapse epochs A and align epochs B.

| | **Random Data Forgetting** | |
|---|---|---|
| Settings | CIFAR-10 | CIFAR-100 |
| | ResNet-18 | |
| FT | 25 Epochs, Adam | |
| | LR $10^{-4}$ | LR $2 \cdot 10^{-4}$ |
| | Retain Batch Size 64 | |
| RL | 7 Epochs, SGD | |
| | LR $10^{-3}$ | LR $5 \cdot 10^{-4}$ |
| | Retain Batch Size 64 | |
| | Forget Batch Size 16 | |
| GA | 10 Epochs, SGD | 10 Epochs, SGD |
| | LR $2.5 \cdot 10^{-3}$ | LR $1 \cdot 10^{-3}$ |
| | Retain Batch Size 64 | |
| | Forget Batch Size 16 | |
| Bad-T | 10 Epochs, Adam | |
| | LR $1 \cdot 10^{-5}$ | |
| | Batch Size 256 | |
| EU-5 / EU-10 | 14 Epochs, SGD | |
| | LR $10^{-1}$ | LR $5 \cdot 10^{-2}$ |
| | Retain Batch Size 64 | |
| CF-5 / CF-10 | 14 Epochs, SGD | |
| | LR $10^{-1}$ | LR $5 \cdot 10^{-2}$ |
| | Retain Batch Size 64 | |
| SCRUB | 5(5) Epochs, SGD | |
| | LR $2.5 \cdot 10^{-5}$ | LR $5.4 \cdot 10^{-4}$ |
| | Retain Batch Size 16 | |
| | Forget Batch Size 64 | |
| SALUN | 10 Epochs, SGD | 15 Epochs, SGD |
| | LR $8.3 \cdot 10^{-4}$ | LR $5 \cdot 10^{-4}$ |
| | Retain Batch Size 64 | |
| | Forget Batch Size 16 | |
| $\ell_1$-sparse | 10 Epochs, SGD | |
| | LR $4 \cdot 10^{-4}$ | LR $3 \cdot 10^{-4}$ |
| | Retain Batch Size 64 | Retain Batch Size 64 |
| COLA+ | 10+10 Epochs, Adam | 10+10 Epochs, Adam |
| | Contrast LR $2 \cdot 10^{-4}$ | Contrast LR $2.5 \cdot 10^{-4}$ |
| | Finetune LR $1 \cdot 10^{-4}$ | Finetune LR $2 \cdot 10^{-5}$ |
| | Retain Batch Size 32 | Retain Batch Size 64 |
| | Forget Batch Size 64 | Forget Batch Size 192 |

Table 11: Single-class forgetting result on CIFAR-10 dataset across different model architectures. A better performance of an MU method corresponds to a smaller performance gap with Retrain (except RTE), with the top method in **bold** and the second best underlined. The $\star$ symbol indicated in RTE of Original and Retrain means that models are pretrained on ImageNet-21K and then finetuned on CIFAR-10, with the reported time reflecting only the finetuning process. In contrast, Original and Retrain without $\star$ are trained from scratch on CIFAR-10.

| | | | **CIFAR-10 - ResNet-18** | | | | |
|---|---|---|---|---|---|---|---|
| Methods | UA | RA | TA | MIA | JSD | **IDI** | RTE (min) |
| Original | 0.0 | 100.0 | 95.46 | 91.50 | 3.21 | 1.000 | 170.32 |
| Retrain | 100.0 | 100.0 | 95.64 | 10.64 | 0.0 | 0.0 | 154.56 |
| FT | $\mathbf{100.0}_{\pm 0.0}$ | $\mathbf{100.0}_{\pm 0.0}$ | $95.12_{\pm 0.09}$ | $0.17_{\pm 0.05}$ | $0.57_{\pm 0.03}$ | $0.671_{\pm 0.008}$ | $6.44_{\pm 0.07}$ |
| RL | $99.93_{\pm 0.01}$ | $\mathbf{100.0}_{\pm 0.0}$ | $\mathbf{95.66}_{\pm 0.09}$ | $0.0_{\pm 0.0}$ | $0.79_{\pm 0.01}$ | $0.830_{\pm 0.005}$ | $3.09_{\pm 0.03}$ |
| GA | $\mathbf{100.0}_{\pm 0.0}$ | $99.06_{\pm 0.25}$ | $93.10_{\pm 0.50}$ | $25.37_{\pm 3.24}$ | $0.59_{\pm 0.05}$ | $0.334_{\pm 0.014}$ | $4.00_{\pm 0.08}$ |
| Bad-T | $99.90_{\pm 0.14}$ | $\underline{99.99}_{\pm 0.0}$ | $94.99_{\pm 0.12}$ | $68.17_{\pm 42.80}$ | $3.69_{\pm 0.85}$ | $1.014_{\pm 0.004}$ | $4.64_{\pm 0.05}$ |
| BoundaryExpand | $71.39_{\pm 0.31}$ | $99.20_{\pm 0.04}$ | $92.53_{\pm 0.02}$ | $7.69_{\pm 0.33}$ | $1.16_{\pm 0.0}$ | $0.892_{\pm 0.001}$ | $\mathbf{0.19}_{\pm 0.01}$ |
| BoundaryShrink | $85.16_{\pm 0.42}$ | $99.60_{\pm 0.17}$ | $93.48_{\pm 0.40}$ | $0.25_{\pm 0.43}$ | $0.75_{\pm 0.01}$ | $0.887_{\pm 0.009}$ | $\underline{0.59}_{\pm 0.02}$ |
| EU-5 | $\mathbf{100.0}_{\pm 0.0}$ | $\mathbf{100.0}_{\pm 0.0}$ | $95.25_{\pm 0.02}$ | $0.06_{\pm 0.03}$ | $0.53_{\pm 0.02}$ | $0.528_{\pm 0.005}$ | $1.54_{\pm 0.00}$ |
| CF-5 | $98.13_{\pm 1.39}$ | $\mathbf{100.0}_{\pm 0.0}$ | $\underline{95.54}_{\pm 0.09}$ | $0.0_{\pm 0.0}$ | $0.56_{\pm 0.04}$ | $0.675_{\pm 0.027}$ | $1.57_{\pm 0.03}$ |
| EU-10 | $\mathbf{100.0}_{\pm 0.0}$ | $99.50_{\pm 0.02}$ | $93.61_{\pm 0.08}$ | $15.24_{\pm 1.08}$ | $\mathbf{0.40}_{\pm 0.01}$ | $-0.349_{\pm 0.019}$ | $2.42_{\pm 0.11}$ |
| CF-10 | $\mathbf{100.0}_{\pm 0.0}$ | $99.98_{\pm 0.0}$ | $94.95_{\pm 0.05}$ | $\mathbf{11.61}_{\pm 0.91}$ | $\underline{0.41}_{\pm 0.01}$ | $-0.060_{\pm 0.017}$ | $2.31_{\pm 0.03}$ |
| SCRUB | $\mathbf{100.0}_{\pm 0.0}$ | $\mathbf{100.0}_{\pm 0.0}$ | $95.37_{\pm 0.04}$ | $19.73_{\pm 1.92}$ | $0.47_{\pm 0.01}$ | $\underline{-0.056}_{\pm 0.008}$ | $3.49_{\pm 0.02}$ |
| SALUN | $\underline{99.99}_{\pm 0.01}$ | $\mathbf{100.0}_{\pm 0.0}$ | $95.42_{\pm 0.12}$ | $0.01_{\pm 0.01}$ | $0.73_{\pm 0.04}$ | $0.936_{\pm 0.012}$ | $3.54_{\pm 0.11}$ |
| $\ell_1$-sparse | $\mathbf{100.0}_{\pm 0.0}$ | $99.93_{\pm 0.02}$ | $94.90_{\pm 0.10}$ | $1.56_{\pm 0.09}$ | $0.47_{\pm 0.03}$ | $0.293_{\pm 0.012}$ | $2.96_{\pm 0.03}$ |
| **COLA** | $\mathbf{100.0}_{\pm 0.0}$ | $\mathbf{100.0}_{\pm 0.00}$ | $95.36_{\pm 0.06}$ | $\underline{12.64}_{\pm 0.92}$ | $0.44_{\pm 0.04}$ | $\mathbf{0.010}_{\pm 0.006}$ | $4.91_{\pm 0.04}$ |
| | | | **CIFAR-10 - ResNet-50** | | | | |
| Methods | UA | RA | TA | MIA | JSD | **IDI** | RTE (min) |
| Original | 0.0 | 100.0 | 95.42 | 95.58 | 4.11 | 1.000 | 341.86 |
| Retrain | 100.0 | 100.0 | 95.49 | 14.92 | 0.0 | 0.0 | 312.24 |
| FT | $\mathbf{100.0}_{\pm 0.0}$ | $\underline{99.99}_{\pm 0.0}$ | $95.28_{\pm 0.11}$ | $2.17_{\pm 1.28}$ | $0.73_{\pm 0.02}$ | $0.607_{\pm 0.009}$ | $14.50_{\pm 0.34}$ |
| RL | $\mathbf{100.0}_{\pm 0.0}$ | $\mathbf{100.0}_{\pm 0.0}$ | $95.56_{\pm 0.03}$ | $0.0_{\pm 0.0}$ | $0.99_{\pm 0.02}$ | $0.804_{\pm 0.006}$ | $6.26_{\pm 0.04}$ |
| GA | $\mathbf{100.0}_{\pm 0.0}$ | $98.06_{\pm 0.34}$ | $92.07_{\pm 0.63}$ | $20.56_{\pm 3.87}$ | $0.66_{\pm 0.06}$ | $0.334_{\pm 0.023}$ | $8.69_{\pm 0.03}$ |
| Bad-T | $\mathbf{100.0}_{\pm 0.0}$ | $99.94_{\pm 0.04}$ | $94.74_{\pm 0.24}$ | $49.95_{\pm 40.74}$ | $3.02_{\pm 0.64}$ | $1.153_{\pm 0.026}$ | $10.19_{\pm 0.32}$ |
| EU-5 | $\mathbf{100.0}_{\pm 0.0}$ | $\mathbf{100.0}_{\pm 0.0}$ | $95.59_{\pm 0.08}$ | $0.0_{\pm 0.0}$ | $0.78_{\pm 0.08}$ | $1.047_{\pm 0.005}$ | $\underline{4.86}_{\pm 0.43}$ |
| CF-5 | $\underline{17.84}_{\pm 0.93}$ | $\mathbf{100.0}_{\pm 0.0}$ | $95.64_{\pm 0.11}$ | $0.0_{\pm 0.0}$ | $1.43_{\pm 0.04}$ | $0.906_{\pm 0.002}$ | $\mathbf{4.84}_{\pm 0.10}$ |
| EU-10 | $\mathbf{100.0}_{\pm 0.0}$ | $\mathbf{100.0}_{\pm 0.0}$ | $\underline{95.51}_{\pm 0.12}$ | $0.17_{\pm 0.05}$ | $0.65_{\pm 0.02}$ | $0.757_{\pm 0.011}$ | $6.92_{\pm 0.02}$ |
| CF-10 | $\mathbf{100.0}_{\pm 0.0}$ | $\mathbf{100.0}_{\pm 0.0}$ | $\mathbf{95.49}_{\pm 0.13}$ | $0.07_{\pm 0.03}$ | $0.67_{\pm 0.08}$ | $0.579_{\pm 0.009}$ | $7.09_{\pm 0.02}$ |
| SCRUB | $\mathbf{100.0}_{\pm 0.0}$ | $\mathbf{100.0}_{\pm 0.0}$ | $95.23_{\pm 0.20}$ | $\underline{18.19}_{\pm 0.10}$ | $0.59_{\pm 0.01}$ | $\underline{0.067}_{\pm 0.020}$ | $8.69_{\pm 0.03}$ |
| SALUN | $\mathbf{100.0}_{\pm 0.0}$ | $99.67_{\pm 0.17}$ | $93.90_{\pm 0.48}$ | $1.58_{\pm 0.98}$ | $0.67_{\pm 0.03}$ | $0.832_{\pm 0.027}$ | $11.00_{\pm 0.06}$ |
| $\ell_1$-sparse | $\mathbf{100.0}_{\pm 0.0}$ | $99.88_{\pm 0.06}$ | $94.49_{\pm 0.29}$ | $4.06_{\pm 0.91}$ | $\mathbf{0.47}_{\pm 0.01}$ | $0.184_{\pm 0.023}$ | $12.33_{\pm 0.04}$ |
| **COLA** | $\mathbf{100.0}_{\pm 0.0}$ | $\underline{99.99}_{\pm 0.0}$ | $95.45_{\pm 0.05}$ | $\mathbf{13.69}_{\pm 0.84}$ | $\underline{0.52}_{\pm 0.02}$ | $\mathbf{0.019}_{\pm 0.025}$ | $11.98_{\pm 0.03}$ |
| | | | **CIFAR-10 - ViT** | | | | |
| Methods | UA | RA | TA | MIA | JSD | **IDI** | RTE (min) |
| Original | 0.36 | 99.55 | 98.40 | 89.12 | 3.96 | 1.000 | $100.68^\star$ |
| Retrain | 100.0 | 99.40 | 97.96 | 4.96 | 0.0 | 0.0 | $90.96^\star$ |
| FT | $98.10_{\pm 0.24}$ | $99.85_{\pm 0.06}$ | $\mathbf{97.58}_{\pm 0.36}$ | $21.14_{\pm 0.92}$ | $0.71_{\pm 0.13}$ | $-0.871_{\pm 0.141}$ | $130.13_{\pm 0.63}$ |
| RL | $97.88_{\pm 2.12}$ | $99.88_{\pm 0.01}$ | $99.01_{\pm 0.02}$ | $0.0_{\pm 0.0}$ | $0.74_{\pm 0.04}$ | $1.052_{\pm 0.011}$ | $65.45_{\pm 0.12}$ |
| GA | $\mathbf{100.0}_{\pm 0.0}$ | $99.80_{\pm 0.03}$ | $98.49_{\pm 0.12}$ | $\mathbf{4.82}_{\pm 0.98}$ | $0.39_{\pm 0.05}$ | $0.498_{\pm 0.025}$ | $68.32_{\pm 0.80}$ |
| Bad-T | $\mathbf{100.0}_{\pm 0.0}$ | $\mathbf{99.55}_{\pm 0.03}$ | $\underline{98.40}_{\pm 0.20}$ | $0.0_{\pm 0.0}$ | $0.84_{\pm 0.06}$ | $0.997_{\pm 0.016}$ | $100.90_{\pm 1.02}$ |
| EU-5 | $\mathbf{100.0}_{\pm 0.0}$ | $99.76_{\pm 0.01}$ | $98.80_{\pm 0.01}$ | $0.30_{\pm 0.01}$ | $0.28_{\pm 0.03}$ | $0.901_{\pm 0.006}$ | $\underline{29.89}_{\pm 0.09}$ |
| CF-5 | $\mathbf{100.0}_{\pm 0.0}$ | $99.76_{\pm 0.0}$ | $98.86_{\pm 0.02}$ | $0.35_{\pm 0.03}$ | $0.26_{\pm 0.01}$ | $0.941_{\pm 0.001}$ | $34.12_{\pm 0.09}$ |
| EU-10 | $\mathbf{100.0}_{\pm 0.0}$ | $99.72_{\pm 0.02}$ | $98.63_{\pm 0.04}$ | $0.64_{\pm 0.02}$ | $\underline{0.23}_{\pm 0.03}$ | $\underline{0.268}_{\pm 0.016}$ | $32.74_{\pm 0.19}$ |
| CF-10 | $\mathbf{100.0}_{\pm 0.0}$ | $99.77_{\pm 0.01}$ | $98.75_{\pm 0.02}$ | $0.64_{\pm 0.04}$ | $\mathbf{0.21}_{\pm 0.02}$ | $0.377_{\pm 0.039}$ | $36.79_{\pm 0.15}$ |
| SCRUB | $\mathbf{100.0}_{\pm 0.0}$ | $\underline{99.66}_{\pm 0.0}$ | $98.57_{\pm 0.01}$ | $94.74_{\pm 0.26}$ | $3.87_{\pm 0.07}$ | $0.907_{\pm 0.027}$ | $\mathbf{22.99}_{\pm 0.24}$ |
| SALUN | $\mathbf{100.0}_{\pm 0.0}$ | $99.78_{\pm 0.02}$ | $98.89_{\pm 0.02}$ | $0.01_{\pm 0.01}$ | $0.39_{\pm 0.05}$ | $1.066_{\pm 0.041}$ | $61.37_{\pm 0.10}$ |
| $\ell_1$-sparse | $\mathbf{100.0}_{\pm 0.0}$ | $97.48_{\pm 0.27}$ | $95.78_{\pm 0.16}$ | $\underline{3.89}_{\pm 0.79}$ | $0.41_{\pm 0.03}$ | $-0.573_{\pm 0.290}$ | $51.44_{\pm 0.04}$ |
| **COLA** | $\underline{99.44}_{\pm 0.02}$ | $\mathbf{100.0}_{\pm 0.0}$ | $98.82_{\pm 0.06}$ | $11.90_{\pm 1.36}$ | $0.63_{\pm 0.11}$ | $\mathbf{-0.067}_{\pm 0.010}$ | $116.01_{\pm 0.96}$ |

Table 12: Single-class forgetting result on CIFAR-100 dataset across different model architectures. A better performance of an MU method corresponds to a smaller performance gap with Retrain (except RTE), with the top method in **bold** and the second best underlined. The $\star$ symbol indicated in RTE of Original and Retrain means that models are pretrained on ImageNet-21K and then finetuned on CIFAR-100, with the reported time reflecting only the finetuning process. In contrast, Original and Retrain without are $\star$ trained from scratch on CIFAR-100.

| | | | **CIFAR-100 - ResNet-18** | | | | |
|---|---|---|---|---|---|---|---|
| Methods | UA | RA | TA | MIA | JSD | **IDI** | RTE (min) |
| Original | 0.0 | 99.98 | 78.18 | 92.80 | 2.91 | 1.000 | 175.08 |
| Retrain | 100.0 | 99.96 | 79.48 | 2.00 | 0.0 | 0.0 | 171.27 |
| FT | $\textbf{100.00}_{\pm 0.0}$ | $\textbf{99.97}_{\pm 0.0}$ | $77.49_{\pm 0.14}$ | $0.07_{\pm 0.09}$ | $0.37_{\pm 0.01}$ | $0.610_{\pm 0.022}$ | $9.50_{\pm 0.03}$ |
| RL | $93.80_{\pm 0.75}$ | $\underline{99.98}_{\pm 0.0}$ | $\underline{77.94}_{\pm 0.10}$ | $0.0_{\pm 0.0}$ | $0.52_{\pm 0.01}$ | $0.467_{\pm 0.010}$ | $3.52_{\pm 0.0}$ |
| GA | $\underline{99.93}_{\pm 0.09}$ | $96.87_{\pm 0.52}$ | $69.87_{\pm 0.78}$ | $21.40_{\pm 2.04}$ | $1.18_{\pm 0.02}$ | $0.392_{\pm 0.021}$ | $5.32_{\pm 0.01}$ |
| Bad-T | $\textbf{100.0}_{\pm 0.0}$ | $\underline{99.98}_{\pm 0.0}$ | $77.66_{\pm 0.26}$ | $40.87_{\pm 36.87}$ | $2.53_{\pm 0.44}$ | $1.079_{\pm 0.024}$ | $5.78_{\pm 0.02}$ |
| BoundaryExpand | $98.93_{\pm 0.12}$ | $98.30_{\pm 0.10}$ | $69.47_{\pm 0.16}$ | $\textbf{1.60}_{\pm 0.0}$ | $0.69_{\pm 0.0}$ | $0.757_{\pm 0.008}$ | $\textbf{0.11}_{\pm 0.01}$ |
| BoundaryShrink | $99.13_{\pm 0.42}$ | $98.67_{\pm 0.12}$ | $69.73_{\pm 0.52}$ | $\underline{1.13}_{\pm 0.42}$ | $0.68_{\pm 0.02}$ | $0.752_{\pm 0.018}$ | $\underline{0.59}_{\pm 0.04}$ |
| EU-5 | $\textbf{100.0}_{\pm 0.0}$ | $99.78_{\pm 0.01}$ | $75.01_{\pm 0.04}$ | $9.33_{\pm 0.75}$ | $0.66_{\pm 0.01}$ | $\underline{0.064}_{\pm 0.037}$ | $2.14_{\pm 0.0}$ |
| CF-5 | $\textbf{100.0}_{\pm 0.0}$ | $\textbf{99.97}_{\pm 0.0}$ | $77.30_{\pm 0.28}$ | $\underline{2.87}_{\pm 0.66}$ | $0.40_{\pm 0.03}$ | $0.388_{\pm 0.010}$ | $2.14_{\pm 0.01}$ |
| EU-10 | $\textbf{100.0}_{\pm 0.0}$ | $91.94_{\pm 0.08}$ | $72.84_{\pm 0.04}$ | $12.67_{\pm 0.47}$ | $0.53_{\pm 0.02}$ | $-0.221_{\pm 0.009}$ | $4.39_{\pm 0.02}$ |
| CF-10 | $\textbf{100.0}_{\pm 0.0}$ | $99.89_{\pm 0.02}$ | $76.49_{\pm 0.02}$ | $7.07_{\pm 0.84}$ | $0.49_{\pm 0.01}$ | $0.175_{\pm 0.040}$ | $4.29_{\pm 0.04}$ |
| SCRUB | $\textbf{100.0}_{\pm 0.0}$ | $\underline{99.98}_{\pm 0.0}$ | $\textbf{78.17}_{\pm 0.04}$ | $0.07_{\pm 0.09}$ | $\underline{0.31}_{\pm 0.01}$ | $0.339_{\pm 0.069}$ | $2.27_{\pm 0.02}$ |
| SALUN | $95.73_{\pm 0.85}$ | $99.22_{\pm 0.13}$ | $74.20_{\pm 0.52}$ | $0.09_{\pm 0.02}$ | $0.65_{\pm 0.01}$ | $0.529_{\pm 0.022}$ | $4.63_{\pm 0.06}$ |
| $\ell_1$-sparse | $96.93_{\pm 0.19}$ | $98.90_{\pm 0.12}$ | $74.69_{\pm 0.06}$ | $6.60_{\pm 0.43}$ | $0.34_{\pm 0.01}$ | $0.334_{\pm 0.026}$ | $4.55_{\pm 0.01}$ |
| **COLA** | $\textbf{100.0}_{\pm 0.0}$ | $99.80_{\pm 0.00}$ | $76.48_{\pm 0.11}$ | $9.60_{\pm 1.31}$ | $\textbf{0.26}_{\pm 0.01}$ | $\textbf{-0.037}_{\pm 0.006}$ | $7.51_{\pm 0.02}$ |

| | | | **CIFAR-100 - ResNet-50** | | | | |
|---|---|---|---|---|---|---|---|
| Methods | UA | RA | TA | MIA | JSD | **IDI** | RTE (min) |
| Original | 0.0 | 99.98 | 79.84 | 91.60 | 3.43 | 1.000 | 345.54 |
| Retrain | 100.0 | 99.97 | 79.42 | 3.40 | 0.0 | 0.0 | 338.58 |
| FT | $99.33_{\pm 0.09}$ | $99.93_{\pm 0.03}$ | $77.71_{\pm 0.18}$ | $0.40_{\pm 0.16}$ | $0.57_{\pm 0.02}$ | $0.618_{\pm 0.018}$ | $16.34_{\pm 0.47}$ |
| RL | $\textbf{100.0}_{\pm 0.0}$ | $99.95_{\pm 0.02}$ | $\underline{79.56}_{\pm 0.04}$ | $0.0_{\pm 0.0}$ | $0.80_{\pm 0.0}$ | $0.649_{\pm 0.013}$ | $8.38_{\pm 0.14}$ |
| GA | $99.60_{\pm 0.43}$ | $98.00_{\pm 0.72}$ | $72.73_{\pm 1.16}$ | $13.33_{\pm 4.43}$ | $0.99_{\pm 0.04}$ | $0.526_{\pm 0.009}$ | $9.50_{\pm 0.54}$ |
| Bad-T | $\textbf{100.0}_{\pm 0.0}$ | $99.90_{\pm 0.10}$ | $77.53_{\pm 1.21}$ | $94.80_{\pm 2.75}$ | $3.98_{\pm 0.25}$ | $0.990_{\pm 0.033}$ | $12.69_{\pm 1.54}$ |
| EU-5 | $\textbf{100.0}_{\pm 0.0}$ | $\textbf{99.97}_{\pm 0.01}$ | $78.31_{\pm 0.21}$ | $\underline{1.20}_{\pm 0.99}$ | $0.61_{\pm 0.04}$ | $0.520_{\pm 0.023}$ | $\underline{6.81}_{\pm 0.01}$ |
| CF-5 | $\textbf{100.0}_{\pm 0.0}$ | $\textbf{99.97}_{\pm 0.01}$ | $78.98_{\pm 0.16}$ | $0.27_{\pm 0.09}$ | $0.50_{\pm 0.02}$ | $0.575_{\pm 0.016}$ | $6.82_{\pm 0.01}$ |
| EU-10 | $\textbf{100.0}_{\pm 0.0}$ | $98.52_{\pm 0.14}$ | $75.66_{\pm 0.03}$ | $15.00_{\pm 1.45}$ | $0.69_{\pm 0.01}$ | $\underline{0.050}_{\pm 0.004}$ | $7.81_{\pm 0.01}$ |
| CF-10 | $\textbf{100.0}_{\pm 0.0}$ | $99.95_{\pm 0.01}$ | $78.47_{\pm 0.10}$ | $5.87_{\pm 0.09}$ | $0.50_{\pm 0.02}$ | $0.302_{\pm 0.035}$ | $7.82_{\pm 0.02}$ |
| SCRUB | $\textbf{100.0}_{\pm 0.0}$ | $\textbf{99.97}_{\pm 0.0}$ | $79.61_{\pm 0.09}$ | $0.20_{\pm 0.16}$ | $\underline{0.43}_{\pm 0.02}$ | $0.620_{\pm 0.034}$ | $\textbf{4.59}_{\pm 0.13}$ |
| SALUN | $\underline{99.73}_{\pm 0.38}$ | $\underline{99.98}_{\pm 0.0}$ | $\textbf{79.51}_{\pm 0.15}$ | $0.0_{\pm 0.0}$ | $0.80_{\pm 0.01}$ | $0.679_{\pm 0.010}$ | $12.83_{\pm 0.87}$ |
| $\ell_1$-sparse | $96.20_{\pm 0.16}$ | $99.42_{\pm 0.06}$ | $76.16_{\pm 0.31}$ | $\textbf{2.60}_{\pm 0.33}$ | $\underline{0.43}_{\pm 0.01}$ | $0.325_{\pm 0.018}$ | $15.78_{\pm 0.05}$ |
| **COLA** | $\textbf{100.0}_{\pm 0.0}$ | $99.90_{\pm 0.01}$ | $78.59_{\pm 0.28}$ | $10.27_{\pm 0.90}$ | $\textbf{0.42}_{\pm 0.02}$ | $\textbf{0.016}_{\pm 0.031}$ | $16.25_{\pm 0.10}$ |

| | | | **CIFAR-100 - ViT** | | | | |
|---|---|---|---|---|---|---|---|
| Methods | UA | RA | TA | MIA | JSD | **IDI** | RTE (min) |
| Original | 7.00 | 95.85 | 90.78 | 69.20 | 2.71 | 1.000 | $102.45^{\star}$ |
| Retrain | 100.0 | 95.79 | 90.58 | 10.00 | 0.0 | 0.0 | $94.29^{\star}$ |
| FT | $\textbf{100.0}_{\pm 0.0}$ | $99.79_{\pm 0.04}$ | $88.69_{\pm 0.11}$ | $14.80_{\pm 2.40}$ | $0.57_{\pm 0.03}$ | $-0.934_{\pm 0.011}$ | $140.61_{\pm 0.25}$ |
| RL | $99.19_{\pm 0.23}$ | $97.11_{\pm 0.02}$ | $92.28_{\pm 0.06}$ | $0.31_{\pm 0.01}$ | $0.82_{\pm 0.01}$ | $1.091_{\pm 0.031}$ | $73.12_{\pm 0.18}$ |
| GA | $\textbf{100.0}_{\pm 0.0}$ | $98.19_{\pm 0.20}$ | $\textbf{90.59}_{\pm 0.21}$ | $17.60_{\pm 4.78}$ | $0.31_{\pm 0.04}$ | $0.587_{\pm 0.011}$ | $75.22_{\pm 0.61}$ |
| Bad-T | $95.80_{\pm 0.08}$ | $\textbf{95.88}_{\pm 0.12}$ | $90.15_{\pm 0.02}$ | $0.0_{\pm 0.0}$ | $1.11_{\pm 0.14}$ | $1.213_{\pm 0.002}$ | $96.43_{\pm 0.01}$ |
| EU-5 | $\textbf{100.0}_{\pm 0.0}$ | $97.59_{\pm 0.04}$ | $92.04_{\pm 0.02}$ | $\underline{7.10}_{\pm 0.70}$ | $\underline{0.27}_{\pm 0.01}$ | $1.143_{\pm 0.008}$ | $\underline{32.17}_{\pm 0.02}$ |
| CF-5 | $\textbf{100.0}_{\pm 0.0}$ | $97.81_{\pm 0.01}$ | $91.98_{\pm 0.05}$ | $6.93_{\pm 0.32}$ | $\underline{0.27}_{\pm 0.01}$ | $1.087_{\pm 0.050}$ | $36.73_{\pm 0.03}$ |
| EU-10 | $\textbf{100.0}_{\pm 0.0}$ | $97.87_{\pm 0.01}$ | $91.45_{\pm 0.07}$ | $13.30_{\pm 1.97}$ | $0.36_{\pm 0.02}$ | $0.849_{\pm 0.012}$ | $34.23_{\pm 0.02}$ |
| CF-10 | $\textbf{100.0}_{\pm 0.0}$ | $97.87_{\pm 0.01}$ | $91.61_{\pm 0.05}$ | $15.80_{\pm 0.80}$ | $0.32_{\pm 0.02}$ | $0.734_{\pm 0.011}$ | $39.12_{\pm 0.0}$ |
| SCRUB | $\textbf{100.0}_{\pm 0.00}$ | $96.95_{\pm 0.03}$ | $92.12_{\pm 0.06}$ | $17.00_{\pm 1.21}$ | $\underline{0.27}_{\pm 0.02}$ | $\underline{0.037}_{\pm 0.036}$ | $\textbf{17.84}_{\pm 0.13}$ |
| SALUN | $\underline{99.73}_{\pm 0.31}$ | $98.32_{\pm 0.04}$ | $92.23_{\pm 0.05}$ | $0.47_{\pm 0.06}$ | $0.78_{\pm 0.02}$ | $1.123_{\pm 0.043}$ | $203.12_{\pm 0.51}$ |
| $\ell_1$-sparse | $\textbf{100.0}_{\pm 0.0}$ | $96.37_{\pm 0.06}$ | $\underline{90.92}_{\pm 0.07}$ | $3.80_{\pm 1.62}$ | $\textbf{0.23}_{\pm 0.01}$ | $1.144_{\pm 0.002}$ | $56.93_{\pm 0.32}$ |
| **COLA** | $\textbf{100.0}_{\pm 0.0}$ | $99.76_{\pm 0.02}$ | $90.23_{\pm 0.04}$ | $\textbf{12.00}_{\pm 2.20}$ | $0.54_{\pm 0.01}$ | $\textbf{-0.022}_{\pm 0.016}$ | $112.58_{\pm 0.82}$ |

Table 13: Multi-class forgetting on CIFAR-10 and CIFAR-100 datasets on ResNet-18 model. A better performance of an MU method corresponds to a smaller performance gap with Retrain (except RTE), with the top method in **bold** and the second best underlined.

| | | | **CIFAR-10 - 2-class forgetting** | | | | |
|---|---|---|---|---|---|---|---|
| Methods | UA | RA | TA | MIA | JSD | **IDI** | RTE (min) |
| Original | 0.0 | 100.0 | 95.76 | 91.10 | 3.55 | 1.000 | 170.32 |
| Retrain | 100.0 | 100.0 | 96.38 | 29.58 | 0.0 | 0.0 | 135.23 |
| FT | $\underline{99.98}_{\pm 0.01}$ | $\mathbf{100.0}_{\pm 0.0}$ | $\underline{96.36}_{\pm 0.09}$ | $0.96_{\pm 0.53}$ | $0.58_{\pm 0.08}$ | $0.750_{\pm 0.009}$ | $5.92_{\pm 0.09}$ |
| RL | $99.70_{\pm 0.02}$ | $\mathbf{100.0}_{\pm 0.0}$ | $\mathbf{96.39}_{\pm 0.01}$ | $0.0_{\pm 0.0}$ | $1.07_{\pm 0.01}$ | $0.863_{\pm 0.001}$ | $2.79_{\pm 0.02}$ |
| GA | $99.07_{\pm 0.38}$ | $99.43_{\pm 0.13}$ | $94.83_{\pm 0.22}$ | $\underline{26.71}_{\pm 3.68}$ | $0.42_{\pm 0.02}$ | $0.612_{\pm 0.001}$ | $3.72_{\pm 0.13}$ |
| Bad-T | $99.96_{\pm 0.05}$ | $\mathbf{100.0}_{\pm 0.0}$ | $95.33_{\pm 0.09}$ | $67.47_{\pm 34.59}$ | $3.98_{\pm 1.08}$ | $1.010_{\pm 0.005}$ | $4.40_{\pm 0.20}$ |
| EU-5 | $\mathbf{100.0}_{\pm 0.0}$ | $\mathbf{100.0}_{\pm 0.0}$ | $96.48_{\pm 0.06}$ | $0.06_{\pm 0.03}$ | $0.57_{\pm 0.05}$ | $0.624_{\pm 0.001}$ | $\mathbf{1.39}_{\pm 0.02}$ |
| CF-5 | $80.06_{\pm 8.26}$ | $\mathbf{100.0}_{\pm 0.0}$ | $96.70_{\pm 0.04}$ | $0.0_{\pm 0.0}$ | $0.80_{\pm 0.02}$ | $0.781_{\pm 0.006}$ | $\underline{1.41}_{\pm 0.05}$ |
| EU-10 | $\mathbf{100.0}_{\pm 0.0}$ | $99.67_{\pm 0.02}$ | $94.94_{\pm 0.17}$ | $25.92_{\pm 0.79}$ | $\underline{0.35}_{\pm 0.01}$ | $\mathbf{-0.011}_{\pm 0.011}$ | $2.20_{\pm 0.17}$ |
| CF-10 | $\mathbf{100.0}_{\pm 0.0}$ | $99.67_{\pm 0.02}$ | $94.94_{\pm 0.17}$ | $21.20_{\pm 1.43}$ | $\underline{0.35}_{\pm 0.01}$ | $\underline{0.221}_{\pm 0.007}$ | $2.19_{\pm 0.14}$ |
| SCRUB | $\underline{99.98}_{\pm 0.0}$ | $\underline{99.99}_{\pm 0.0}$ | $96.31_{\pm 0.08}$ | $46.74_{\pm 5.31}$ | $1.47_{\pm 0.10}$ | $0.374_{\pm 0.005}$ | $3.27_{\pm 0.01}$ |
| SALUN | $95.86_{\pm 4.18}$ | $\underline{99.99}_{\pm 0.01}$ | $96.27_{\pm 0.11}$ | $0.04_{\pm 0.01}$ | $0.89_{\pm 0.05}$ | $0.951_{\pm 0.019}$ | $3.17_{\pm 0.02}$ |
| $\ell_1$-sparse | $99.91_{\pm 0.05}$ | $99.98_{\pm 0.0}$ | $96.47_{\pm 0.09}$ | $1.57_{\pm 0.11}$ | $0.50_{\pm 0.02}$ | $0.560_{\pm 0.004}$ | $2.62_{\pm 0.06}$ |
| **COLA** | $\mathbf{100.0}_{\pm 0.0}$ | $99.92_{\pm 0.0}$ | $96.41_{\pm 0.15}$ | $\mathbf{31.40}_{\pm 2.98}$ | $\mathbf{0.26}_{\pm 0.01}$ | $\mathbf{0.011}_{\pm 0.029}$ | $4.59_{\pm 0.02}$ |

| | | | **CIFAR-100 - 5-class forgetting** | | | | |
|---|---|---|---|---|---|---|---|
| Methods | UA | RA | TA | MIA | JSD | **IDI** | RTE (min) |
| Original | 0.0 | 99.98 | 77.95 | 95.00 | 3.18 | 1.000 | 175.08 |
| Retrain | 100.0 | 99.98 | 78.45 | 7.12 | 0.0 | 0.0 | 165.92 |
| FT | $\mathbf{100.00}_{\pm 0.0}$ | $99.93_{\pm 0.06}$ | $77.43_{\pm 0.20}$ | $0.20_{\pm 0.06}$ | $0.38_{\pm 0.01}$ | $0.596_{\pm 0.009}$ | $9.21_{\pm 0.06}$ |
| RL | $98.61_{\pm 0.22}$ | $\mathbf{99.98}_{\pm 0.0}$ | $\mathbf{77.78}_{\pm 0.19}$ | $0.0_{\pm 0.0}$ | $0.71_{\pm 0.01}$ | $0.613_{\pm 0.008}$ | $3.39_{\pm 0.09}$ |
| GA | $79.99_{\pm 4.75}$ | $95.18_{\pm 0.40}$ | $68.68_{\pm 0.52}$ | $32.25_{\pm 2.02}$ | $1.36_{\pm 0.06}$ | $0.236_{\pm 0.010}$ | $4.99_{\pm 0.04}$ |
| Bad-T | $\mathbf{100.0}_{\pm 0.0}$ | $\mathbf{99.98}_{\pm 0.0}$ | $75.93_{\pm 0.57}$ | $44.60_{\pm 31.96}$ | $2.86_{\pm 0.25}$ | $1.021_{\pm 0.031}$ | $5.51_{\pm 0.11}$ |
| EU-5 | $\mathbf{100.0}_{\pm 0.0}$ | $99.75_{\pm 0.02}$ | $75.14_{\pm 0.12}$ | $12.40_{\pm 0.26}$ | $0.54_{\pm 0.01}$ | $\underline{0.054}_{\pm 0.010}$ | $\mathbf{2.01}_{\pm 0.0}$ |
| CF-5 | $\mathbf{100.0}_{\pm 0.0}$ | $\underline{99.97}_{\pm 0.0}$ | $77.36_{\pm 0.06}$ | $\underline{3.37}_{\pm 0.52}$ | $\underline{0.36}_{\pm 0.02}$ | $0.319_{\pm 0.011}$ | $\underline{2.10}_{\pm 0.0}$ |
| EU-10 | $\mathbf{100.0}_{\pm 0.0}$ | $91.76_{\pm 0.12}$ | $73.24_{\pm 0.11}$ | $21.96_{\pm 0.49}$ | $0.48_{\pm 0.01}$ | $-0.155_{\pm 0.008}$ | $4.25_{\pm 0.0}$ |
| CF-10 | $\mathbf{100.0}_{\pm 0.0}$ | $99.88_{\pm 0.01}$ | $76.59_{\pm 0.24}$ | $\mathbf{10.69}_{\pm 1.29}$ | $0.40_{\pm 0.01}$ | $0.087_{\pm 0.019}$ | $4.29_{\pm 0.01}$ |
| SCRUB | $\mathbf{100.0}_{\pm 0.0}$ | $\underline{99.97}_{\pm 0.0}$ | $\underline{77.64}_{\pm 0.11}$ | $0.95_{\pm 0.35}$ | $0.56_{\pm 0.03}$ | $0.289_{\pm 0.015}$ | $2.27_{\pm 0.03}$ |
| SALUN | $\mathbf{100.0}_{\pm 0.0}$ | $99.96_{\pm 0.01}$ | $77.18_{\pm 0.14}$ | $0.13_{\pm 0.09}$ | $0.55_{\pm 0.01}$ | $0.597_{\pm 0.029}$ | $4.46_{\pm 0.04}$ |
| $\ell_1$-sparse | $\underline{98.63}_{\pm 0.37}$ | $97.50_{\pm 0.14}$ | $73.46_{\pm 0.25}$ | $12.35_{\pm 0.82}$ | $0.38_{\pm 0.01}$ | $0.196_{\pm 0.011}$ | $4.19_{\pm 0.01}$ |
| **COLA** | $\mathbf{100.0}_{\pm 0.0}$ | $99.82_{\pm 0.0}$ | $77.47_{\pm 0.26}$ | $11.16_{\pm 0.54}$ | $\mathbf{0.29}_{\pm 0.01}$ | $\mathbf{0.044}_{\pm 0.010}$ | $7.31_{\pm 0.02}$ |

| | | | **CIFAR-100 - 20-class forgetting** | | | | |
|---|---|---|---|---|---|---|---|
| Methods | UA | RA | TA | MIA | JSD | **IDI** | RTE (min) |
| Original | 0.0 | 99.97 | 78.03 | 95.04 | 3.15 | 1.000 | 175.08 |
| Retrain | 100.0 | 99.98 | 80.01 | 7.55 | 0.0 | 0.0 | 139.93 |
| FT | $\underline{99.81}_{\pm 0.04}$ | $\underline{99.97}_{\pm 0.00}$ | $\mathbf{79.11}_{\pm 0.35}$ | $0.22_{\pm 0.05}$ | $0.37_{\pm 0.01}$ | $\underline{0.474}_{\pm 0.007}$ | $7.43_{\pm 0.07}$ |
| RL | $95.77_{\pm 0.09}$ | $\mathbf{99.98}_{\pm 0.01}$ | $78.42_{\pm 0.05}$ | $0.0_{\pm 0.0}$ | $0.63_{\pm 0.01}$ | $1.207_{\pm 0.001}$ | $2.94_{\pm 0.01}$ |
| GA | $67.06_{\pm 2.58}$ | $96.65_{\pm 0.47}$ | $70.80_{\pm 0.65}$ | $30.16_{\pm 1.42}$ | $1.46_{\pm 0.11}$ | $1.027_{\pm 0.006}$ | $4.11_{\pm 0.02}$ |
| Bad-T | $95.54_{\pm 0.61}$ | $\mathbf{99.98}_{\pm 0.01}$ | $69.71_{\pm 0.32}$ | $32.07_{\pm 35.23}$ | $2.83_{\pm 0.26}$ | $1.211_{\pm 0.011}$ | $5.17_{\pm 0.11}$ |
| EU-5 | $\mathbf{100.0}_{\pm 0.0}$ | $99.82_{\pm 0.02}$ | $76.89_{\pm 0.03}$ | $14.50_{\pm 0.54}$ | $0.52_{\pm 0.01}$ | $0.807_{\pm 0.003}$ | $\underline{1.83}_{\pm 0.04}$ |
| CF-5 | $\mathbf{100.0}_{\pm 0.0}$ | $99.96_{\pm 0.02}$ | $\underline{78.82}_{\pm 0.06}$ | $2.68_{\pm 0.21}$ | $\underline{0.33}_{\pm 0.01}$ | $1.060_{\pm 0.008}$ | $\mathbf{1.80}_{\pm 0.03}$ |
| EU-10 | $\mathbf{100.0}_{\pm 0.0}$ | $93.25_{\pm 0.32}$ | $74.79_{\pm 0.39}$ | $25.63_{\pm 0.38}$ | $0.47_{\pm 0.01}$ | $0.617_{\pm 0.005}$ | $3.61_{\pm 0.51}$ |
| CF-10 | $\mathbf{100.0}_{\pm 0.0}$ | $99.91_{\pm 0.01}$ | $78.39_{\pm 0.24}$ | $13.57_{\pm 0.32}$ | $0.39_{\pm 0.01}$ | $0.889_{\pm 0.005}$ | $3.68_{\pm 0.16}$ |
| SCRUB | $95.03_{\pm 0.75}$ | $99.90_{\pm 0.00}$ | $77.61_{\pm 0.07}$ | $0.93_{\pm 0.13}$ | $0.38_{\pm 0.01}$ | $0.997_{\pm 0.007}$ | $2.14_{\pm 0.02}$ |
| SALUN | $90.69_{\pm 0.76}$ | $98.97_{\pm 0.14}$ | $74.72_{\pm 0.54}$ | $0.17_{\pm 0.03}$ | $0.60_{\pm 0.01}$ | $1.113_{\pm 0.008}$ | $3.85_{\pm 0.0}$ |
| $\ell_1$-sparse | $83.49_{\pm 0.46}$ | $99.52_{\pm 0.03}$ | $76.79_{\pm 0.20}$ | $\mathbf{6.36}_{\pm 0.59}$ | $0.38_{\pm 0.01}$ | $1.035_{\pm 0.007}$ | $3.08_{\pm 0.07}$ |
| **COLA** | $\mathbf{100.0}_{\pm 0.0}$ | $99.92_{\pm 0.0}$ | $78.59_{\pm 0.32}$ | $\underline{11.52}_{\pm 0.39}$ | $\mathbf{0.24}_{\pm 0.01}$ | $\mathbf{0.007}_{\pm 0.010}$ | $6.97_{\pm 0.01}$ |

Table 14: 5-class forgetting results on ImageNet-1K dataset across different model architectures. A better performance of an MU method corresponds to a smaller performance gap with Retrain (except RTE), with the top method in **bold** and the second best underlined. The $\star$ symbol indicated in RTE of Original and Retrain means that models are pretrained on ImageNet-21K and then finetuned on ImageNet-1K, with the reported time reflecting only the finetuning process. In contrast, Original and Retrain without $\star$ are trained from scratch on ImageNet-1K.

| ImageNet-1K - ResNet-50 | | | | | | | |
|---|---|---|---|---|---|---|---|
| Methods | UA | RA | TA | MIA | JSD | **IDI** | RTE (min) |
| Original | 11.72 | 87.45 | 76.11 | 61.69 | 3.73 | 1.000 | 2680.15 |
| Retrain | 100.0 | 88.80 | 75.88 | 9.41 | 0.0 | 0.0 | 2661.90 |
| FT | $\mathbf{100.0}_{\pm 0.0}$ | $\mathbf{88.52}_{\pm 0.0}$ | $\underline{76.16}_{\pm 0.01}$ | $8.24_{\pm 1.23}$ | $\underline{0.24}_{\pm 0.01}$ | $0.102_{\pm 0.026}$ | $140.04_{\pm 1.42}$ |
| RL | $\underline{99.96}_{\pm 0.03}$ | $86.46_{\pm 0.07}$ | $75.23_{\pm 0.01}$ | $0.23_{\pm 0.01}$ | $1.57_{\pm 0.03}$ | $1.002_{\pm 0.007}$ | $200.73_{\pm 1.87}$ |
| GA | $\mathbf{100.0}_{\pm 0.0}$ | $80.77_{\pm 0.22}$ | $71.49_{\pm 0.10}$ | $4.20_{\pm 0.46}$ | $0.42_{\pm 0.03}$ | $0.328_{\pm 0.023}$ | $212.14_{\pm 2.61}$ |
| Bad-T | $98.01_{\pm 0.02}$ | $84.03_{\pm 0.03}$ | $73.42_{\pm 0.03}$ | $69.13_{\pm 12.57}$ | $3.51_{\pm 0.41}$ | $1.152_{\pm 0.072}$ | $211.52_{\pm 0.96}$ |
| BoundaryExpand | $77.22_{\pm 0.11}$ | $82.79_{\pm 0.08}$ | $71.78_{\pm 0.09}$ | $1.43_{\pm 0.51}$ | $1.34_{\pm 0.0}$ | $0.628_{\pm 0.005}$ | $\underline{5.14}_{\pm 0.02}$ |
| BoundaryShrink | $91.20_{\pm 0.02}$ | $81.41_{\pm 0.17}$ | $70.55_{\pm 0.08}$ | $1.45_{\pm 0.34}$ | $1.13_{\pm 0.01}$ | $0.543_{\pm 0.011}$ | $\mathbf{4.81}_{\pm 0.03}$ |
| EU-5 | $\mathbf{100.0}_{\pm 0.0}$ | $79.62_{\pm 0.0}$ | $71.22_{\pm 0.13}$ | $13.33_{\pm 1.53}$ | $0.26_{\pm 0.01}$ | $0.183_{\pm 0.028}$ | $193.38_{\pm 0.78}$ |
| CF-5 | $\mathbf{100.0}_{\pm 0.0}$ | $84.31_{\pm 0.08}$ | $74.16_{\pm 0.06}$ | $10.21_{\pm 5.33}$ | $\mathbf{0.23}_{\pm 0.01}$ | $0.701_{\pm 0.014}$ | $81.53_{\pm 0.56}$ |
| EU-10 | $\mathbf{100.0}_{\pm 0.0}$ | $71.84_{\pm 0.03}$ | $65.78_{\pm 0.02}$ | $16.65_{\pm 1.91}$ | $0.35_{\pm 0.04}$ | $\underline{-0.051}_{\pm 0.021}$ | $193.79_{\pm 0.47}$ |
| CF-10 | $\mathbf{100.0}_{\pm 0.0}$ | $80.87_{\pm 0.04}$ | $72.34_{\pm 0.08}$ | $13.99_{\pm 5.41}$ | $0.25_{\pm 0.01}$ | $0.608_{\pm 0.012}$ | $82.29_{\pm 0.34}$ |
| SCRUB | $99.28_{\pm 0.07}$ | $\underline{88.39}_{\pm 0.04}$ | $76.51_{\pm 0.03}$ | $7.42_{\pm 0.51}$ | $0.25_{\pm 0.01}$ | $0.517_{\pm 0.011}$ | $426.04_{\pm 2.98}$ |
| SALUN | $89.67_{\pm 0.27}$ | $86.25_{\pm 0.15}$ | $75.54_{\pm 0.10}$ | $0.50_{\pm 0.09}$ | $0.88_{\pm 0.01}$ | $0.343_{\pm 0.017}$ | $793.82_{\pm 3.32}$ |
| $\ell_1$-sparse | $97.57_{\pm 0.61}$ | $85.33_{\pm 0.07}$ | $74.77_{\pm 0.03}$ | $\underline{8.84}_{\pm 1.39}$ | $0.32_{\pm 0.02}$ | $0.239_{\pm 0.031}$ | $226.74_{\pm 1.35}$ |
| **COLA** | $\mathbf{100.0}_{\pm 0.0}$ | $87.93_{\pm 0.05}$ | $\mathbf{76.15}_{\pm 0.04}$ | $\mathbf{9.95}_{\pm 1.21}$ | $\underline{0.24}_{\pm 0.01}$ | $\mathbf{0.040}_{\pm 0.042}$ | $171.44_{\pm 0.75}$ |

| ImageNet-1K - ViT | | | | | | | |
|---|---|---|---|---|---|---|---|
| Methods | UA | RA | TA | MIA | JSD | **IDI** | RTE (min) |
| Original | 2.48 | 98.18 | 80.59 | 71.00 | 4.45 | 1.000 | $1943.69^{\star}$ |
| Retrain | 100.0 | 98.33 | 80.42 | 8.09 | 0.0 | 0.0 | $1920.77^{\star}$ |
| FT | $96.39_{\pm 0.01}$ | $98.85_{\pm 0.03}$ | $80.93_{\pm 0.06}$ | $3.88_{\pm 0.33}$ | $0.65_{\pm 0.02}$ | $0.937_{\pm 0.009}$ | $281.73_{\pm 2.30}$ |
| RL | $98.33_{\pm 0.02}$ | $98.99_{\pm 0.07}$ | $81.65_{\pm 0.07}$ | $0.0_{\pm 0.0}$ | $2.13_{\pm 0.15}$ | $1.152_{\pm 0.033}$ | $150.32_{\pm 4.31}$ |
| GA | $\mathbf{100.0}_{\pm 0.01}$ | $97.04_{\pm 0.01}$ | $\underline{80.17}_{\pm 0.04}$ | $8.26_{\pm 2.14}$ | $0.52_{\pm 0.23}$ | $0.674_{\pm 0.021}$ | $193.73_{\pm 2.23}$ |
| Bad-T | $98.21_{\pm 0.03}$ | $\underline{97.85}_{\pm 0.07}$ | $\mathbf{80.58}_{\pm 0.03}$ | $0.0_{\pm 0.0}$ | $2.62_{\pm 0.06}$ | $1.312_{\pm 0.015}$ | $721.15_{\pm 5.23}$ |
| EU-5 | $\mathbf{100.0}_{\pm 0.0}$ | $93.82_{\pm 0.02}$ | $80.00_{\pm 0.01}$ | $4.74_{\pm 1.33}$ | $0.63_{\pm 0.02}$ | $0.519_{\pm 0.008}$ | $300.55_{\pm 0.76}$ |
| CF-5 | $98.75_{\pm 0.0}$ | $96.57_{\pm 0.01}$ | $80.09_{\pm 0.04}$ | $4.49_{\pm 0.34}$ | $0.64_{\pm 0.01}$ | $0.731_{\pm 0.024}$ | $\mathbf{122.39}_{\pm 0.53}$ |
| EU-10 | $\mathbf{100.0}_{\pm 0.0}$ | $87.33_{\pm 0.10}$ | $76.26_{\pm 0.13}$ | $\mathbf{8.09}_{\pm 0.20}$ | $\mathbf{0.36}_{\pm 0.02}$ | $-2.662_{\pm 0.231}$ | $345.37_{\pm 0.70}$ |
| CF-10 | $\underline{99.95}_{\pm 0.01}$ | $93.86_{\pm 0.02}$ | $78.69_{\pm 0.01}$ | $7.68_{\pm 1.11}$ | $0.72_{\pm 0.03}$ | $\underline{0.009}_{\pm 0.021}$ | $\underline{140.11}_{\pm 0.49}$ |
| SCRUB | $\mathbf{100.0}_{\pm 0.00}$ | $98.84_{\pm 0.02}$ | $81.62_{\pm 0.01}$ | $3.19_{\pm 0.91}$ | $1.062_{\pm 0.03}$ | $-0.846_{\pm 0.032}$ | $404.02_{\pm 2.96}$ |
| SALUN | $94.64_{\pm 0.76}$ | $\mathbf{98.13}_{\pm 0.21}$ | $80.74_{\pm 0.05}$ | $0.13_{\pm 0.01}$ | $1.83_{\pm 0.09}$ | $0.980_{\pm 0.065}$ | $321.13_{\pm 2.75}$ |
| $\ell_1$-sparse | $93.55_{\pm 0.62}$ | $94.69_{\pm 0.37}$ | $78.84_{\pm 0.10}$ | $2.98_{\pm 0.33}$ | $\underline{0.49}_{\pm 0.01}$ | $0.831_{\pm 0.022}$ | $717.42_{\pm 3.21}$ |
| **COLA** | $\mathbf{100.0}_{\pm 0.0}$ | $96.42_{\pm 0.03}$ | $79.28_{\pm 0.21}$ | $\underline{8.02}_{\pm 1.36}$ | $0.59_{\pm 0.02}$ | $\mathbf{0.006}_{\pm 0.007}$ | $501.12_{\pm 2.17}$ |

Table 15: Random data forgetting on CIFAR-10 and CIFAR-100 datasets on ResNet-18 model. A better performance of an MU method corresponds to a smaller performance gap with Retrain (except RTE), with the top method in **bold** and the second best underlined.

| | | | | **CIFAR-10 - ResNet-18** | | | |
|---|---|---|---|---|---|---|---|
| Methods | UA | RA | TA | MIA | JSD | **IDI** | RTE (min) |
| Original | 0.0 | 100.0 | 95.54 | 92.90 | 0.09 | 1.000 | 170.32 |
| Retrain | 3.94 | 100.0 | 95.26 | 75.12 | 0.0 | 0.0 | 152.87 |
| FT | $5.03_{\pm0.40}$ | $98.95_{\pm0.21}$ | $92.94_{\pm0.26}$ | $83.52_{\pm0.58}$ | $0.07_{\pm0.11}$ | $\underline{-0.069}_{\pm0.013}$ | $8.11_{\pm0.03}$ |
| RL | $4.77_{\pm0.27}$ | $\mathbf{99.92}_{\pm0.0}$ | $\mathbf{93.54}_{\pm0.04}$ | $22.47_{\pm1.19}$ | $0.38_{\pm0.02}$ | $0.084_{\pm0.030}$ | $2.75_{\pm0.01}$ |
| GA | $2.86_{\pm0.76}$ | $98.37_{\pm0.71}$ | $91.90_{\pm0.70}$ | $85.49_{\pm2.17}$ | $0.09_{\pm0.01}$ | $0.924_{\pm0.028}$ | $4.31_{\pm0.03}$ |
| Bad-T | $5.47_{\pm1.05}$ | $\underline{99.87}_{\pm0.05}$ | $91.51_{\pm0.61}$ | $39.53_{\pm3.43}$ | $0.27_{\pm0.03}$ | $0.939_{\pm0.053}$ | $4.78_{\pm0.09}$ |
| EU-10 | $3.16_{\pm0.19}$ | $98.68_{\pm0.08}$ | $93.07_{\pm0.12}$ | $\underline{83.40}_{\pm0.21}$ | $\underline{0.06}_{\pm0.01}$ | $-0.110_{\pm0.013}$ | $\underline{2.13}_{\pm0.05}$ |
| CF-10 | $2.71_{\pm0.24}$ | $99.11_{\pm0.06}$ | $\underline{93.47}_{\pm0.15}$ | $84.33_{\pm0.05}$ | $\mathbf{0.05}_{\pm0.01}$ | $0.219_{\pm0.029}$ | $\mathbf{2.10}_{\pm0.06}$ |
| SCRUB | $\underline{4.31}_{\pm1.50}$ | $96.21_{\pm1.70}$ | $88.83_{\pm1.86}$ | $37.88_{\pm7.65}$ | $0.56_{\pm0.09}$ | $0.322_{\pm0.016}$ | $3.37_{\pm0.05}$ |
| SALUN | $2.74_{\pm0.30}$ | $97.77_{\pm0.04}$ | $91.68_{\pm0.44}$ | $83.52_{\pm2.20}$ | $0.10_{\pm0.03}$ | $0.861_{\pm0.012}$ | $5.69_{\pm0.04}$ |
| $\ell_1$-sparse | $5.47_{\pm0.22}$ | $96.66_{\pm0.07}$ | $91.31_{\pm0.25}$ | $\mathbf{77.12}_{\pm0.21}$ | $0.09_{\pm0.01}$ | $-0.157_{\pm0.026}$ | $3.03_{\pm0.04}$ |
| **COLA+** | $\mathbf{3.90}_{\pm0.08}$ | $99.24_{\pm0.17}$ | $93.23_{\pm0.09}$ | $83.48_{\pm0.10}$ | $\underline{0.06}_{\pm0.01}$ | $\mathbf{0.024}_{\pm0.010}$ | $7.80_{\pm0.02}$ |
| | | | | **CIFAR-100 - ResNet-18** | | | |
| Methods | UA | RA | TA | MIA | JSD | **IDI** | RTE (min) |
| Original | 0.0 | 99.98 | 78.09 | 95.82 | 0.56 | 1.000 | 175.08 |
| Retrain | 23.10 | 99.98 | 77.78 | 39.72 | 0.0 | 0.0 | 170.31 |
| FT | $17.44_{\pm1.12}$ | $98.46_{\pm0.24}$ | $70.99_{\pm0.45}$ | $67.35_{\pm0.53}$ | $0.46_{\pm0.02}$ | $0.311_{\pm0.034}$ | $8.40_{\pm0.13}$ |
| RL | $24.67_{\pm0.42}$ | $\underline{99.66}_{\pm0.0}$ | $73.10_{\pm0.49}$ | $2.13_{\pm0.17}$ | $0.84_{\pm0.02}$ | $-0.246_{\pm0.056}$ | $2.95_{\pm0.03}$ |
| GA | $11.73_{\pm1.43}$ | $95.21_{\pm0.78}$ | $68.38_{\pm1.03}$ | $74.97_{\pm1.10}$ | $0.65_{\pm0.01}$ | $0.704_{\pm0.039}$ | $4.66_{\pm0.03}$ |
| Bad-T | $64.35_{\pm7.44}$ | $99.07_{\pm0.56}$ | $53.05_{\pm2.53}$ | $11.85_{\pm5.93}$ | $1.51_{\pm0.16}$ | $1.003_{\pm0.006}$ | $5.04_{\pm0.05}$ |
| EU-10 | $24.15_{\pm0.09}$ | $90.15_{\pm0.08}$ | $72.25_{\pm0.36}$ | $\mathbf{59.47}_{\pm0.39}$ | $0.27_{\pm0.01}$ | $0.404_{\pm0.085}$ | $\underline{2.31}_{\pm0.02}$ |
| CF-10 | $20.40_{\pm0.20}$ | $95.06_{\pm0.24}$ | $\mathbf{74.44}_{\pm0.23}$ | $62.18_{\pm0.27}$ | $\underline{0.25}_{\pm0.01}$ | $0.464_{\pm0.061}$ | $\mathbf{2.30}_{\pm0.02}$ |
| SCRUB | $3.47_{\pm2.85}$ | $97.77_{\pm2.31}$ | $71.89_{\pm2.87}$ | $71.49_{\pm4.15}$ | $0.37_{\pm0.02}$ | $0.528_{\pm0.013}$ | $3.59_{\pm0.05}$ |
| SALUN | $32.77_{\pm1.20}$ | $\mathbf{99.87}_{\pm0.02}$ | $71.97_{\pm0.37}$ | $3.32_{\pm0.28}$ | $0.81_{\pm0.02}$ | $\underline{-0.226}_{\pm0.078}$ | $5.99_{\pm0.09}$ |
| $\ell_1$-sparse | $\mathbf{22.83}_{\pm0.15}$ | $88.94_{\pm0.41}$ | $69.54_{\pm0.73}$ | $62.36_{\pm0.37}$ | $0.26_{\pm0.01}$ | $0.634_{\pm0.072}$ | $3.37_{\pm0.03}$ |
| **COLA+** | $\underline{23.50}_{\pm0.16}$ | $93.78_{\pm0.07}$ | $\underline{73.15}_{\pm0.59}$ | $\underline{59.58}_{\pm0.24}$ | $\mathbf{0.24}_{\pm0.01}$ | $\mathbf{0.078}_{\pm0.013}$ | $10.2_{\pm0.16}$ |

Table 16: Standard Deviation of Figure 5 - (CIFAR-10, ResNet-18)

| Method | Block 1 | Block 2 | Block 3 | Block 4 | Block 5 | IDI |
|---|---|---|---|---|---|---|
| Original | 0.002 | 0.002 | 0.003 | 0.006 | 0.006 | 0.005 |
| Retrain | 0.001 | 0.003 | 0.003 | 0.006 | 0.007 | 0.007 |
| FT | 0.001 | 0.002 | 0.004 | 0.010 | 0.008 | 0.007 |
| RL | 0.002 | 0.004 | 0.005 | 0.003 | 0.005 | 0.004 |
| GA | 0.001 | 0.001 | 0.004 | 0.006 | 0.011 | 0.013 |
| l1-sparse | 0.001 | 0.000 | 0.002 | 0.007 | 0.007 | 0.011 |
| SCRUB | 0.001 | 0.005 | 0.003 | 0.004 | 0.005 | 0.007 |
| SALUN | 0.002 | 0.001 | 0.003 | 0.005 | 0.012 | 0.011 |

Table 17: Standard Deviation of Figure 5 - (CIFAR-10, ResNet-50)

| Method | Block 1 | Block 2 | Block 3 | Block 4 | Block 5 | Block 6 | IDI |
|---|---|---|---|---|---|---|---|
| Original | 0.003 | 0.002 | 0.009 | 0.007 | 0.013 | 0.008 | 0.011 |
| Retrain | 0.001 | 0.003 | 0.001 | 0.008 | 0.005 | 0.007 | 0.009 |
| FT | 0.003 | 0.005 | 0.008 | 0.015 | 0.011 | 0.012 | 0.019 |
| RL | 0.005 | 0.005 | 0.007 | 0.003 | 0.008 | 0.004 | 0.009 |
| GA | 0.002 | 0.001 | 0.003 | 0.011 | 0.010 | 0.013 | 0.018 |
| l1-sparse | 0.002 | 0.004 | 0.002 | 0.004 | 0.021 | 0.015 | 0.023 |
| SCRUB | 0.000 | 0.004 | 0.004 | 0.023 | 0.028 | 0.031 | 0.060 |
| SALUN | 0.001 | 0.000 | 0.006 | 0.005 | 0.020 | 0.011 | 0.019 |

Table 18: Standard Deviation of Figure 8

| Method | Ratio 1:5 | Ratio 1:20 | Ratio 1:99 |
|---|---|---|---|
| FT | 0.013 | 0.008 | 0.019 |
| RL | 0.016 | 0.007 | 0.009 |
| GA | 0.020 | 0.008 | 0.018 |
| CF-10 | 0.007 | 0.016 | 0.035 |
| SALUN | 0.023 | 0.025 | 0.019 |
| SCRUB | 0.006 | 0.013 | 0.023 |

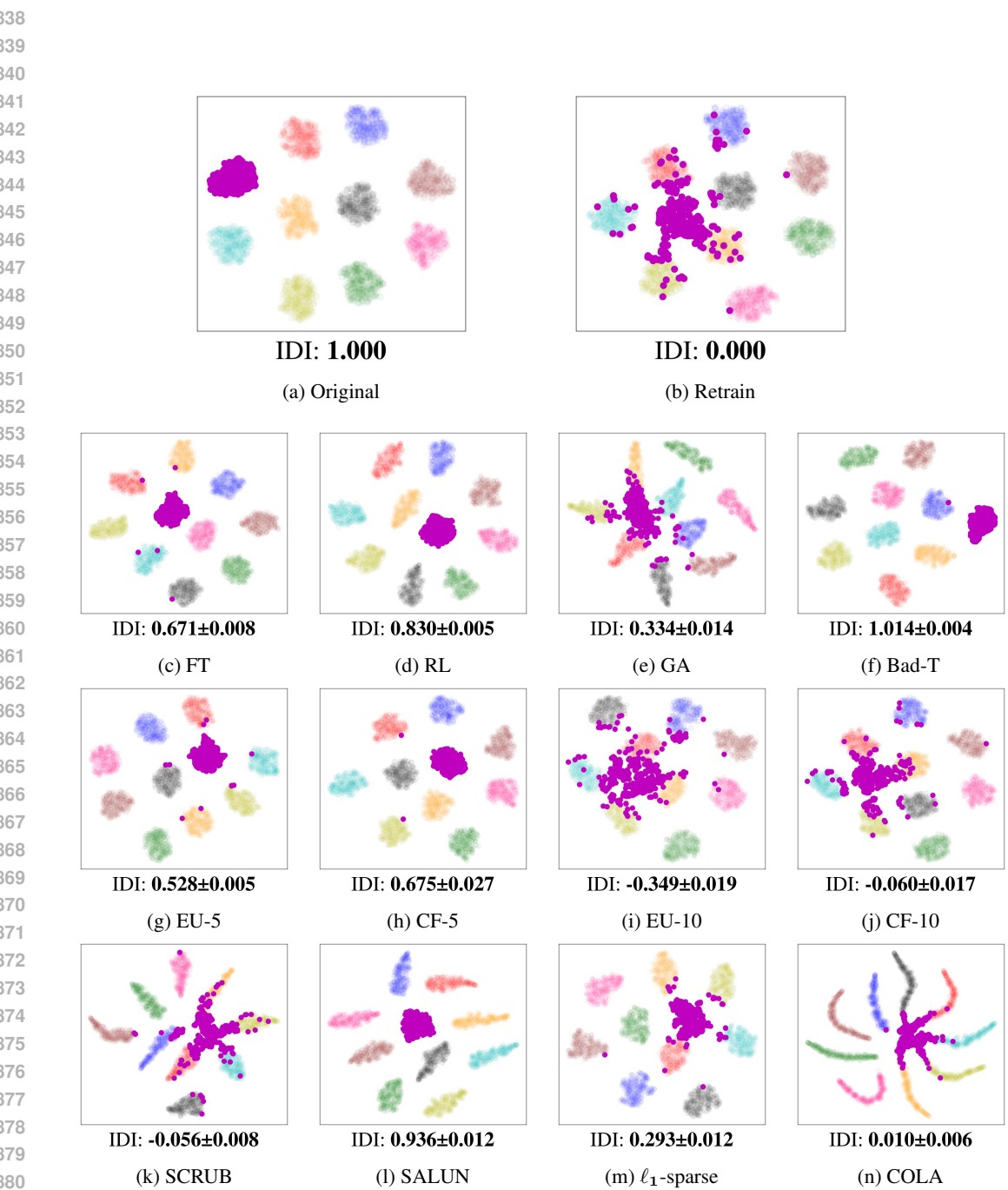

Figure 15: t-SNE visualizations of features of Original, Retrain, and unlearned models (FT, RL, GA, Bad-T, EU-5, CF-5, EU-10, CF-10, SCRUB, SALUN, $\ell_1$-sparse, and COLA) on CIFAR-10 with ResNet-18. The forgetting class is represented in purple, while rest of the points represents the remaining class.

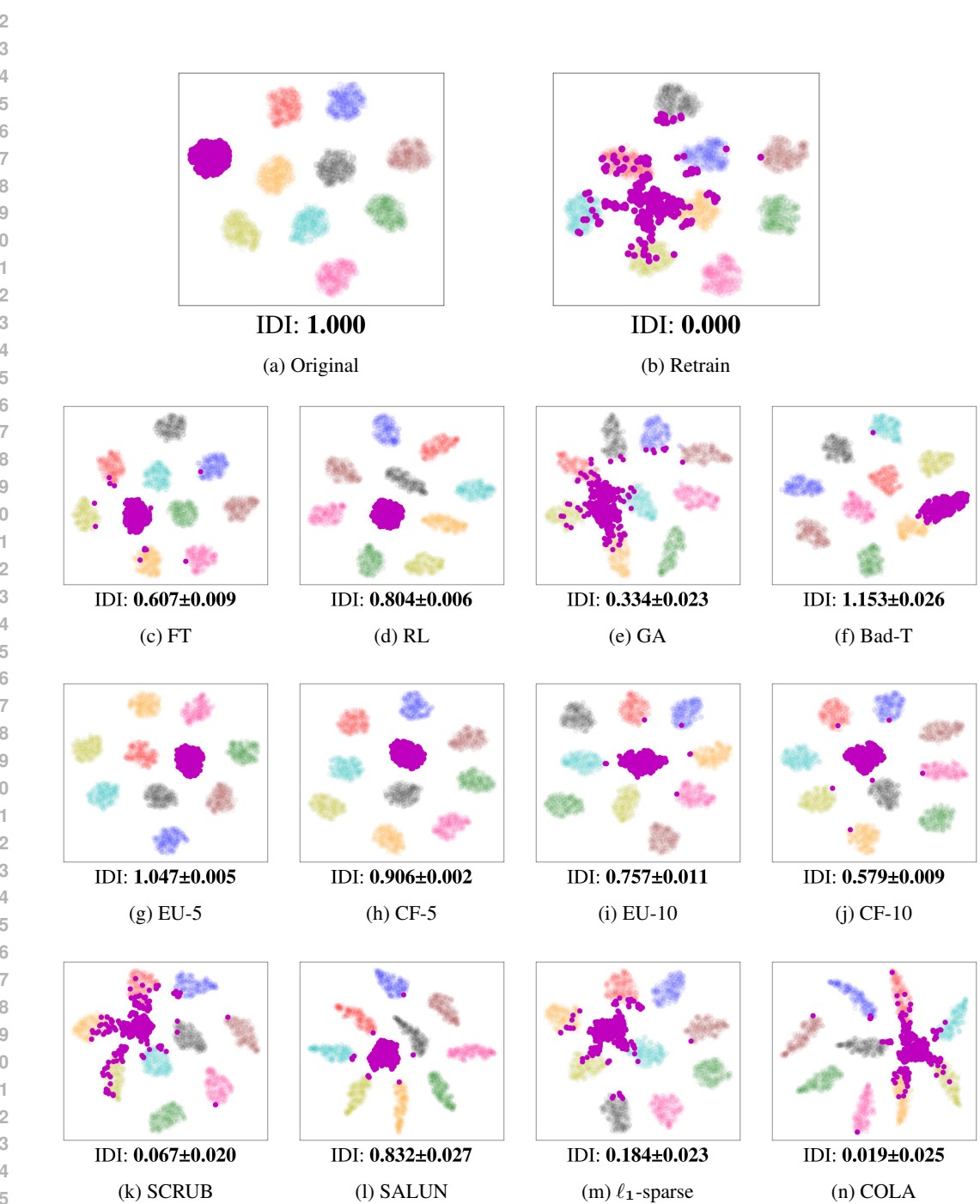

Figure 16: t-SNE visualizations of feature of Original, Retrain, and unlearned models (FT, RL, GA, Bad-T, EU-5, CF-5, EU-10, CF-10, SCRUB, SALUN, $\ell_1$-sparse, and COLA) on CIFAR-10 with ResNet-50. The forgetting class is represented in purple, while rest of the points represents the remaining class.

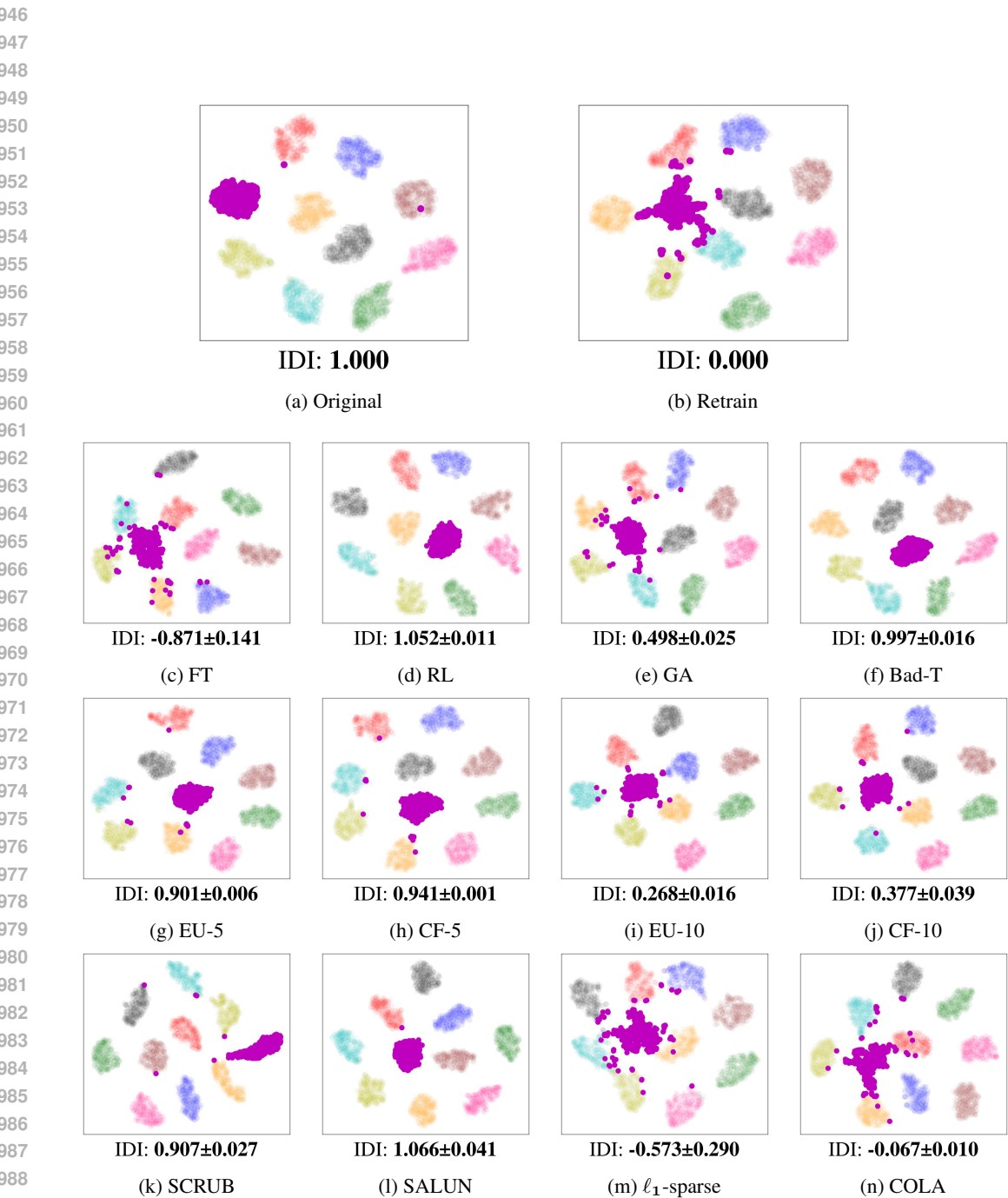

Figure 17: t-SNE visualizations of features of Original, Retrain, and unlearned models (FT, RL, GA, Bad-T, EU-5, CF-5, EU-10, CF-10, SCRUB, SALUN, $\ell_1$-sparse, and COLA) on CIFAR-10 with ViT. The forgetting class is represented in purple, while rest of the points represents the remaining class.

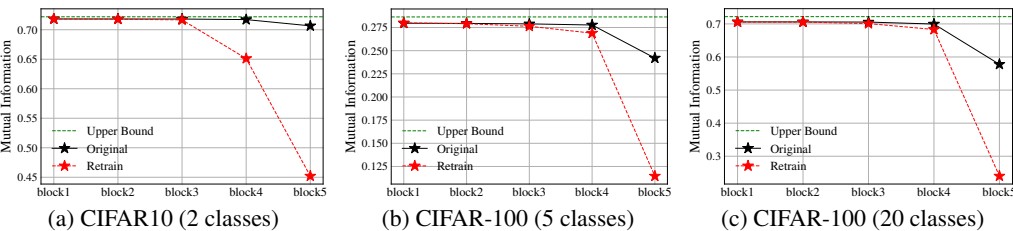

Figure 18: Mutual information curves across various datasets and model architectures. It illustrates the estimated mutual information $I(\mathbf{Z}_\ell; Y)$ of the features from the $\ell$-th layer $\mathbf{Z}_\ell$ and the binary label $Y$, computed by the InfoNCE loss. 'block(-k)' means the k block front from the last layer.

Figure 19: Mutual information curves for multiple class unlearning in ResNet-18 architecture. It illustrates the estimated mutual information $I(\mathbf{Z}_\ell; Y)$ of the features from the $\ell$-th layer $\mathbf{Z}_\ell$ and the binary label $Y$, computed by the InfoNCE loss.

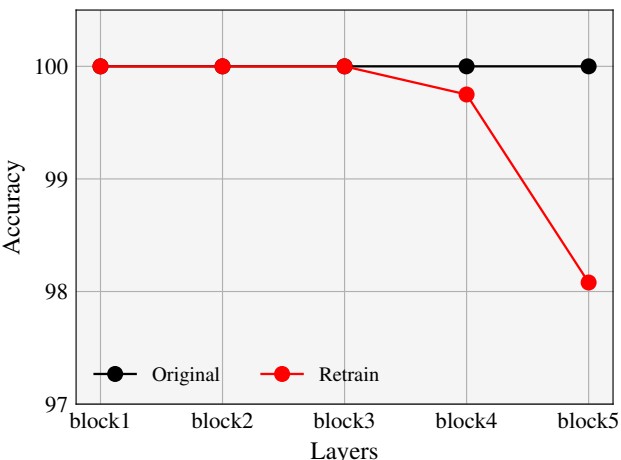

Figure 20: Binary train accuracy on CIFAR-10 in single-class forgetting with retain and forget sets. Interestingly, it shows similar results with mutual information plots shown Figure 18.

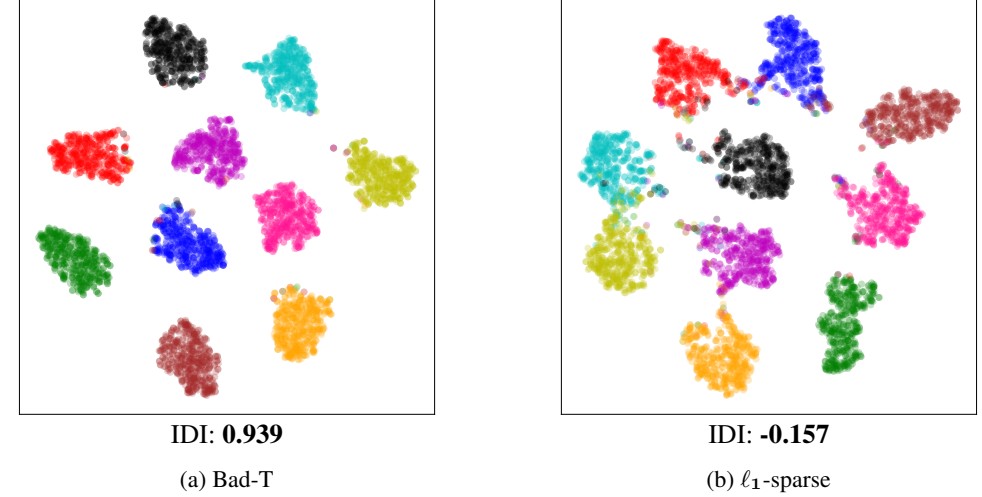

IDI: **0.939**                                          IDI: **-0.157**

(a) Bad-T                                          (b) $\ell_1$-sparse

Figure 21: t-SNE visualizations of features of forget samples of Bad-T and $\ell_1$-sparse in a random data forgetting task on (CIFAR-10, ResNet-18). The clusters of $\ell_1$-sparse are more disperse than those of Bad-T.

