# OpenReview forum: "AN INFORMATION THEORETIC EVALUATION METRIC FOR STRONG UNLEARNING"
_ICLR.cc/2025/Conference — Submitted to ICLR 2025_

### Official Review · Reviewer_33Yi · 2024-11-03

**Soundness:** 3
**Presentation:** 3
**Contribution:** 2
**Rating:** 5
**Confidence:** 3

**Summary:**

This paper investigate the shortcomings of current white-box unlearning evaluation method. The author use an example in class-wise unlearn to illustrate the residual information in the model. Then, they analyze the InfoNCE loss and propose the new white-box evaluation metrics called IDI. To better evaluate the model, the author proposes COLA. The extensive experiment results show the new evaluation metrics can evaluate the residual information in the model better.

**Strengths:**

- The topic itself is really interesting and necessary for MU.
- The experiments and plots are detailed and promising.
- The baseline method selection is relatively complete and the description of the method is complete.

**Weaknesses:**

- The references need revision. For example, Model sparsity can simplify machine unlearning is the paper from NeurIPS 2023 and the author list is incorrect.
- What MIA is used? May I ask the reason for choosing this MIA over the others, like the MIA in [2]? I guess the MIA used here is based on [1].
- In addition to the proposed evaluation metrics, it would be more promising if the author can report
- The existing metrics like UA, TA, which is not perfect. There are still some metrics, like ZRF [3], AUS [4], etc. Since the author wants to introduce a new and better evaluation metrics, a complete overview for the existing metrics should be necessary.
- The introduction of the proposed evaluation metrics is a little bit hard to follow because of the notation mass. For example, there are several weights in this paper, like $\theta$, $\nu$, $\eta$. Maybe a notation table can make everything clear.
- The performance in Table 1 is useful for the reader to connect your proposed method with the existing metrics. However, I wonder why there is no such a table for random forgetting.
- More baselines like Boundary Unlearning [5] should be added.


[1] Song, Liwei, and Prateek Mittal. "Systematic evaluation of privacy risks of machine learning models." 30th USENIX Security Symposium (USENIX Security 21). 2021.

[2] Carlini, Nicholas, et al. "Membership inference attacks from first principles." 2022 IEEE Symposium on Security and Privacy (SP). IEEE, 2022.

[3] Poppi, Samuele, et al. "Multi-Class Unlearning for Image Classification via Weight Filtering." IEEE Intelligent Systems (2024).

[4] Cotogni, Marco, et al. "Duck: Distance-based unlearning via centroid kinematics." arXiv preprint arXiv:2312.02052 (2023).

[5] Chen, Min, et al. "Boundary unlearning: Rapid forgetting of deep networks via shifting the decision boundary." Proceedings of the IEEE/CVF Conference on Computer Vision and Pattern Recognition. 2023.

**Questions:**

- What is your point of using HD to support your argument the black-box is insufficient? Why do you think the change in most of the parameters is a good sign of unlearning? And why do treat HD as a good method to illustrate your opinion. From my perspective, there are some probabilities that the information is controlled by the key parameters, where is those in last layer in your example.
- From the practical perspective, I wonder whether the meaning of this work will be decreased as long as the evaluation is mostly based on the output.
- What is the meaning of $g_{n_l}(0)$ and $g_{n_l}(1)$? What is the output? Could you please give an example?
- In Table 1, there are both positive and negative IDIs. Is there any difference between the positive and negative value?
- It seems that the proposed method needs some training. Could you please present the testing time of the proposed method?
- In the sentence of Line 358,

---

> ### Author Response · Authors · 2024-11-20
> **comment (1/2)**
>
> > W1. Reference Revision
>
> We sincerely apologize for the oversight and fixed it. We also have carefully reviewed all references throughout the manuscript, and made the necessary corrections (in **blue**). Thank you!
>
> >W2. The choice of MIA over other MIAs (C-MIA, LiRA MIA).
>
> We primarily use the Entropy-based MIA (E-MIA) [1] as our black-box MIA variant due to its widespread adoption [2,3,4] and its greater variability across algorithms, as noted in **App. C.1**. In contrast, the Confidence-based MIA (C-MIA) [5] shows limited variability across algorithms, as reported in Table A2 of [6].
> **Additionally, we include results using LiRA MIA [7], specifically for ResNet-18 on CIFAR-10 in single-class forgetting**
>
> |Method|Retrain|HD|FT|RL|GA|$l$1-sparse|SALUN|COLA|
> |-|-|-|-|-|-|-|-|-|
> |E-MIA|10.64±0.0|2.05±0.11|0.17±0.05|0.0±0.0|25.37±3.24|1.56±0.09|0.01±0.01|12.64±0.92|
> |Lira MIA|0.5|0.654|0.76|0.798|0.712|0.543|0.623|0.547|
>
> We emphasize that LiRA MIA, while comprehensive, is computationally prohibitive for large-scale experiments involving models like ViT or datasets like ImageNet, as it requires training over 10,000 models for unlearning evaluation.
> Moreover, these results confirm that LiRA MIA shares the limitations of other black-box metrics, which fail to capture residual information. HD’s decent performance on these metrics, despite modifying only the last layer, underscores the need for white-box metrics like IDI to offer a more reliable and comprehensive evaluation of unlearning.
>
> [1] R. Shokri et al., Membership inference attacks against machine learning models, ISSP, 2017\
> [2] Kurmanji et al., Towards unbounded machine unlearning, NeurIPS, 2023\
> [3] Chundawat et al., Can bad teaching induce forgetting? unlearning in deep networks using an incompetent teacher, AAAI, 2023\
> [4] J. Foster et al., Fast machine unlearning without retraining through selective synaptic dampening, AAAI, 2024\
> [5] Song et al., Systematic evaluation of privacy risks of machine learning models, USENIX Security, 2021\
> [6] C. Fan et al., Salun: Empowering machine unlearning via gradient-based weight saliency in both image classification and generation, ICLR, 2024\
> [7] Carlini et al., Membership inference attacks from first principles, S&P, 2022
>
> > W3,4. Additional evaluation metrics like ZRF, AUS are needed
>
> **We evaluated unlearning algorithms using ZRF and AUS to demonstrate their black-box inherent limitations as below. Note that higher values of ZRF and AUS indicate better performance.**
>
> - CIFAR-10 - ResNet-18 - 1 class
> |Method|Original|Retrain|HD|FT|RL|GA|$l$1-sparse|SALUN|
> |-|-|-|-|-|-|-|-|-|
> |ZRF|0.478±0.005|0.587±0.002|0.869±0.15|0.762±0.015|0.973±0.002|0.526±0.031|0.742±0.005|0.963±0.008|
> |AUS|0.500±0.0|1.002±0.0|0.998±0.006|0.997±0.008|1.001±0.004|0.976±0.032|0.994±0.009|0.999±0.010|
> - CIFAR-100 - ResNet18 - 5 classes
> |Method|Original|Retrain|HD|FT|RL|GA|$l$1-sparse|SALUN|
> |-|-|-|-|-|-|-|-|-|
> |ZRF|0.343±0.001|0.628±0.001|0.918±0.001|0.696±0.001|0.985±0.001|0.408±0.002|0.658±0.011|0.982±0.002|
> |AUS|0.5±0.0|1.005±0.0|0.998±0.002|0.995±0.180|0.985±0.036|0.756±0.251|0.945±0.024|0.992±0.012|
> - CIFAR-100 - ResNet18 - 20 classes
> |Method|Original|Retrain|HD|FT|RL|GA|$l$1-sparse|SALUN|
> |-|-|-|-|-|-|-|-|-|
> |ZRF|0.343±0.001|0.65±0.0|0.954±0.001|0.70±0.004|0.970±0.0|0.403±0.007|0.705±0.006|0.954±0.002|
> |AUS|0.5±0.0|1.020±0.0|1.015±0.002|1.009±0.034|0.963±0.010|0.698±0.156|0.885±0.076|0.949±0.142|
> - CIFAR-10 - ResNet18 - 500 samples
> |Method|Original|Retrain|HD|FT|RL|GA|$l$1-sparse|SALUN|
> |-|-|-|-|-|-|-|-|-|
> |ZRF|0.478±0.0|0.492±0.0|0.515±0.001|0.482±0.001|0.668±0.005|0.483±0.010|0.521±0.001|0.494±0.0|
> |AUS|0.957±0.0|0.989±0.0|0.939±0.221|0.955±0.083|0.964±0.016|0.916±0.072|0.928±0.022|0.911±0.053|
>
> HD's performance, achieving comparable scores on ZRF and AUS while modifying only the last layer, highlights the need for robust white-box metrics like IDI. Note that we follow the common practice in the unlearning literature [1, 2], referred to as the “full-stack” evaluation criteria [3]. By the reviewer’s suggestion, we have expanded our discussion in the Preliminary section (**Sec. 2.2**). Additionally, we have updated the manuscript to include a comparison of IDI with white-box MIAs in **Sec. 5.3**.
>
> [1] Kurmanji,et al., Towards unbounded machine unlearning, NeurIPS, 2023\
> [2] Chundawat et al., Can bad teaching induce forgetting? unlearning in deep networks using an incompetent teacher, AAAI, 2023\
> [3] C. Fan et al., Salun: Empowering machine unlearning via gradient-based weight saliency in both image classification and generation, ICLR, 2024
>
> > W5. Notations in the paper
>
> Thank you for your suggestion. We have simplified the notation by adopting batch notation instead of individual input-label pairs and clarified the notations for critic functions in **App. B**. Please refer to the updated **Sec. 4.1 and App. B** in the revised manuscript.

---

> ### Author Response · Authors · 2024-11-20
> **comment (2/2)**
>
> > W6. Random data forgetting result in the main text
>
> Thank you for your consideration. We have moved random forgetting results from the appendix to main text in the revised manuscript. Part of random forgetting results are shown below. Please refer to **Tab. 2 and Sec. 5.2** for the details.
> - CIFAR-10 - ResNet-18 - 500 samples per class
> |Methods|UA|TA|MIA|IDI|RTE(min)|
> |-|-|-|-|-|-|
> |Retrain|3.94|95.26|75.12|0.000|152.87
> |HD|3.64±1.66|92.80±1.18|77.47±4.09|1.000|0.30±0.05|
> |FT|5.03±0.40|92.94±.26|83.52±0.58|-0.069±0.013|8.11±0.03|
> |RL|4.77±0.27|93.54±0.04|22.47±1.19|0.084±0.030|2.75±0.01|
> |SCRUB|4.31±1.50|88.83±1.86|37.88±7.65|0.322±0.016|3.37±0.05|
> |$l$1-sparse|5.47±0.22|91.31±0.25|77.12±0.21|-0.157±0.026|3.03±0.04|
> |COLA+|3.90±0.08|93.23±0.09|83.48±0.10|0.024±0.010|7.80±0.02|
>
> > W7. More comparison baselines
>
> As suggested, we have included Boundary Unlearning in our main experimental table **(Tab.1)** and in the appendix **(Tab. 11, 12, and 14)**. We are also considering further revisions to additional tables. Thank you!
>
> > Q1. Point of using HD for insufficiency of black-box metrics
>
> HD retains the same residual information about forget set across layers as Original, which differs significantly from Retrain, as shown in **Fig. 5**. Despite this, black-box metrics often classify HD as a properly unlearned model although altering only the last layer as HD is insufficient to make a model indistinguishable (strong unlearning) from Retrain. Thereby, HD serves as a clear example, reinforcing the need for white-box metrics like IDI, which analyze residual information in middle layers for more reliable unlearning assessments. Additionally, we have extended HD’s performance analysis to multi-class and random data forgetting scenarios, detailed in **App. D.1** and included in **Tab. 1 and Tab. 2**.
>
> > Q2. Practicality of this work due to output based evaluations
>
> Our work points out the limitations of output based evaluations (black-box) and we believe that this can encourage the community to transition toward internal evaluations using white-box metrics like our proposed IDI.
>
> > Q3. Meaning of $g$ in MI estimation
>
> **The function $g_\eta$ is a mapping function from label $Y$ to the label representation vector of dimension $d$, matching the output size of $f_\nu$. Thus, $g_\eta (0)$ and $g_\eta (1)$ represent the vector embeddings for labels $Y=0$ and $Y=1$, respectively.** For example, if the output of $f_\nu$ is a 512-dimensional vector, then the output of $g_\eta$ will also be a 512-dimensional vector corresponding to the input label. To apply the InfoNCE loss, we align the dimension of label representations with the feature vectors, enabling the loss to effectively measure the mutual information between features and labels.
>
> > Q4. Meaning of negative IDI values
>
> **A negative numerator in IDI, $\mathbf{ID}(\theta_u)$, means that the residual information level in the unlearned model ($\theta_u$) is lower than in Retrain, a phenomenon we describe as “over-unlearning” (as noted in **line 399**).** This property makes IDI highly interpretable, as a negative value indicates a reduction in residual information compared to Retrain, quantifying the extent of unlearning achieved.
>
> > Q5. Testing time of the proposed method, IDI
>
> Here is the table presenting the time cost (in minutes) for computing mutual information for each layer in CIFAR-10 single class forgetting. Please note that we compute IDI using only the last $n$ selected layers (i.e., the last layers of the last two blocks for ResNet-18 and the last three blocks for ResNet-50, details added in **App. E.2** and **App. E.6**), as earlier layers exhibit nearly identical information between Original and Retrain, as shown in  **Fig. 5**. We added the clarification of this approach in **line 392** of our revised manuscript.
> - ResNet-18 and ResNet-50
> |Retain set ratio|Block 1|Block 2|Block 3|Block 4|Block 5|Retain set ratio|Block 1|Block 2|Block 3|Block 4|Block 5|Block 6|
> |-|-|-|-|-|-|-|-|-|-|-|-|-|
> |100%|6.64±0.05|6.51±0.17|6.32±0.12|6.04±0.10|5.90±0.15|100%|19.91±0.18|17.70±0.01|15.75±0.02|15.32±0.11|14.98±0.07|14.83±0.04|
> |10%|1.61±0.05|1.55±0.21|1.63±0.09|1.49±0.11|1.42±0.13|10%|4.17±0.13|3.85±0.04|3.43±0.01|3.42±0.15|3.35±0.02|3.24±0.15|
> - ViT
> |Retain set ratio|Block 1|Block 2|Block 3|Block 4|Block 5|Block 6|Block 7|Block 8|Block 9|Block 10|Block 11|Block 12|
> |-|-|-|-|-|-|-|-|-|-|-|-|-|
> |100%|167.00|162.81|160.64|159.56|157.65|155.35|151.82|147.78|146.05|144.25|142.81|139.61|
> |10%|32.46|32.12|31.43|30.99|30.64|30.21|29.50|29.01|28.75|28.43|27.95|27.14|
>
> **This process takes approximately 11 minutes for ResNet-18 and 45 minutes for ResNet-50 when computed sequentially**, while IDI remains identical to that with full layers as shown in Tab.5 in the appendix. Additionally, as demonstrated in **Fig. 8**, our method remains effective even when using only 10% of the dataset for metric computation, reducing the required time by a factor of 4.

---

> ### Author Response · Authors · 2024-12-02
> **Gentle Reminder for Reviewer 33Yi**
>
> We sincerely appreciate the effort you have dedicated to reviewing our manuscript. As the discussion period is nearing its conclusion, we kindly remind the reviewer that we have invested significant effort in addressing your questions. We would be grateful if you could share any additional questions or comments regarding our responses.

---

> ### Comment · Reviewer_33Yi · 2024-12-03
> **Response from the Reviewer**
>
> Hi authors,
>
> Thanks for your detailed reply. I really appreciate it. However, I am still concern about the meaning of this work. My concerns are as follows.
>
>
> 1. Compared with the black-box method, we compare the difference between the model output. In this paper, you study the output of each layer. What is the essential difference between these two strategies?
>
> 2. The source of your idea is from head distillation. However, I don't think head distillation can capture the model difference. Your paper is about the difference for each layer. However, your inspiration is from the difference in last layer. I think there is some mismatching.
>
> 3. I am still confused about the structure of critics. Are they MLP or just linear layers?
>
> In all, I appreciate the authors' hard working a lot and the completeness of the work. However, the source of my concern is whether the author uses a correct way to study the model difference compared with the existing way. I hope the author can try to address my concern and I will read your paper at least two times again although I have read it a lot of times. I have decreased my confidence score since maybe I am wrong. I did some research in this area. However, an evaluation metrics should be careful and serious. I am looking forward to the reply from the author and I am sure I will consider your paper seriously, discuss with AC and other reviewers before I submit my final score.
>
> Best,
> Reviewer 33Yi

---

> ### Author Response · Authors · 2024-12-03
> **Additional response by the authors to the reviewer 33Yi's response**
>
> > Q2. Clarification of head distillation (HD)
>
> We would like to clarify that the purpose of HD is to highlight the limitations of black-box evaluation schemes in unlearning. As the reviewer rightly pointed out, HD is not a valid unlearning algorithm, as it modifies only the last layer while retaining all internal information. However, as shown in Sec. 3 and App. D.1, HD achieves results comparable to state-of-the-art methods under current black-box metrics, underscoring their incompleteness and unreliability. To truly achieve the objectives of unlearning, we argue that white-box evaluation schemes are essential, as they comprehensively assess residual information across the model rather than relying solely on outputs.
>
> > Q1. The essential difference between the black-box metrics and our metric
>
> The fundamental difference between black-box metrics and our approach lies in the ability to evaluate residual influence of forgotten samples not only at the output level but also across intermediate layers of the model. Black-box evaluations in Machine Unlearning rely on output differences, such as logits, and often equate favorable results with successful unlearning. However, as evidenced by our HD (Head Distillation) experiments, these metrics have critical limitations. HD modifies only the last layer while retaining residual information in intermediate layers, yet still achieves favorable results under black-box metrics. This demonstrates that such metrics are insufficient to ensure complete unlearning.
> Residual information in intermediate layers is particularly problematic, as it renders the unlearned model distinguishable from a retrained model and undermines privacy objectives by enabling the reconstruction of forgotten data with minimal effort (Sec. 3.2, Fig. 4). This clearly highlights the need for robust white-box metrics capable of assessing unlearning across all layers of the model, including intermediate ones.
> Our proposed metric, IDI, directly addresses this gap by leveraging an information-theoretic foundation, providing stability, robustness, and interpretability. Unlike existing white-box metrics, such as $\ell_2$ distance or white-box MIA [1], which often lack consistency and reliability (Sec. 5.3), IDI offers a practical and comprehensive framework for unlearning evaluation. To the best of our knowledge, IDI is the first white-box metric specifically designed to rigorously assess unlearning across the model, including intermediate layers.
>
> > Q3. The structure of critic functions
>
> When measuring mutual information between $Y_{\ell}$ and $Z$, the critic function $f_{\ell}$ maps the $\ell$-th intermediate feature $Y_{\ell}$, while $g$ maps the forget label $Z$. For example, at layer $\ell=2$, $f_{\ell}$ corresponds to the model architecture starting from layer 3 onward, and $g$, being simpler in structure, is a single linear layer since it only processes a binary label. These critic functions act as auxiliary networks essential for mutual information computation. They are optimized to maximize the InfoNCE loss, which provides a lower bound on mutual information. Further details can be found in Sec. 4 and App. B.
>
> We sincerely thank the reviewer for the thoughtful question and engagement with our work, and we hope this clarification fully addresses your concern.

---

### Official Review · Reviewer_Nhgb · 2024-11-03

**Soundness:** 2
**Presentation:** 3
**Contribution:** 2
**Rating:** 6
**Confidence:** 4

**Summary:**

This paper first introduces a naive machine unlearning method that modifies only the last layer of the model, achieving strong performance based on existing black-box evaluation metrics. Building on this, the author introduces a new white-box metric based on the mutual information between hidden features and labels to be forgotten. Furthermore, the paper proposes a new unlearning method, Collapse and Align (COLA), which effectively removes residual information at the feature level.

**Strengths:**

1. The topic of designing evaluation metrics for machine unlearning is both significant and challenging. This paper introduces a new metric based on the mutual information between the features and labels to be forgotten, effectively capturing the residual information in the unlearned model.

2. A new unlearning method inspired by the proposed evaluation metric is introduced, which is greatly appreciated.

3. The presentation is clear and easy to follow.

4. Several experiments are conducted to demonstrate the efficacy of the proposed evaluation method.

**Weaknesses:**

1. If I understand correctly, for each unlearned model with $L$ layers, mutual information must be estimated $3L$ times, each requiring data sampling and the training of two networks. Coupled with the time needed for retraining, the overall complexity is substantial.

2. Given the complexity, the necessity of such calculations needs further scrutiny. A simpler white-box baseline, using an MLP to train a mapping from concatenated hidden features to forget_labels and obtaining a similar metric based on equation 2, might reduce complexity. It would be interesting to see if your proposed method outperforms this baseline.

3. One benefit claimed by the author is that the proposed metric exhibits a low standard deviation. However, this seems to hold true only when the number of forgettings is low, as demonstrated in Table 1.

4. The experiments comparing IDI with white-box MIA should be expanded and included in the main manuscript rather than the appendix to allow for a fair comparison.

5. Regarding Membership Inference Attacks (MIAs), only score-based MIA is employed, while Likelihood Ratio Attack (LiRA) [1] is not.

6. There lack of literature on auditing machine unlearning.


[1] Carlini, Nicholas, et al. "Membership inference attacks from first principles." 2022 IEEE Symposium on Security and Privacy (SP). IEEE, 2022.

**Questions:**

1. Could you explain why the sum of absolute values is not used when calculating Equations (1) and (2)?

2. Is there any possibility of encountering a near-zero denominator during the calculation in Equation (2)?

3. Could you provide a comparison of the effectiveness of computing mutual information for just the last few layers versus all layers?

---

> ### Author Response · Authors · 2024-11-20
> **comment (1/2)**
>
> > W1. Time for retraining and computational burden of IDI
>
> Since the goal of (strong) unlearning is to align the unlearned model's behavior with that of Retrain, evaluation typically requires Retrain as the gold standard. As a result, **existing evaluation metrics commonly depend on the presence of Retrain**, as outlined in Tab. 6 of [1].\
> **Moreover, IDI calculation does not impose significant complexity, as the mutual information is not estimated 3L times.** We compute MI on only the last $n$ selected layers (i.e., the last layers of later blocks in ResNet or ViT), where $n \ll L$, as earlier layers exhibit nearly identical information between Original and Retrain, as shown in **Fig. 5**. We added the clarification of this approach in **line 392 and App. E.2, App. E.6** of our revised manuscript. For instance, MI estimation only takes 45 minutes on ResNet-50 - CIFAR-100 whereas retraining takes 338 minutes -7.5 times faster.\
> Using subsets of data further reduces computation time as shown in **Fig. 8**: ResNet-50 requires only 10 GPU minutes with 10% of the data, and ViT takes 90 GPU minutes, roughly four times faster than using the full dataset. These optimizations demonstrate the practicality of IDI, even for large models like ViT.
>
> [1] Nguyen et al., A survey of machine unlearning, arXiv, 2022
>
> > W2. Suggested simpler baseline for MI estimation
>
> As the reviewer's suggestion, we empirically validated the proposed simpler baseline by concatenating the latent features from the final three residual blocks (blocks 3, 4, and 5) of ResNet-18 in the CIFAR-10 single-class forgetting setup. Specifically, we flattened and combined these features into a single vector, applied a 3-layer MLP (hidden dimensions: [25088-1024-512]) as the critic function $f$, and trained it using Equation 2 along with the additional function $g$ employed in our experiments.
> **However, this approach yielded nearly identical mutual information (MI) estimates for Original, Retrain, and other baseline algorithms, as demonstrated below.**
> ||Original|Retrain|FT|RL|GA|Bad-T|EU5|CF5|EU10|CF10|SCRUB|SALUN|$l$1 SPARSE|COLA|
> |-|-|-|-|-|-|-|-|-|-|-|-|-|-|-|
> |Concat|0.465±0.0|0.464±0.002|0.466±0.0|0.466±0.0|0.458±0.008|0.466±0.0|0.463±0.005|0.465±0.03|0.462±0.0|0.462±0.018|0.458±0.012|0.466±0.001|0.466±0.002|0.466±0.001|
> |IDI(Ours)|1.0|0.0|0.671±0.008|0.830±0.005|0.334±0.014|1.014±0.004|0.528±0.005|0.675±0.027|-0.349±0.019|-0.060±0.017|-0.056±0.008|0.936±0.012|0.293±0.012|0.010±0.006|
>
> We believe this occurs because features from earlier layers (e.g., block 3) exhibit negligible differences between Original and Retrain, which extends to the concatenated features. Consequently, the suggested baseline fails to effectively differentiate between Original and unlearned models, limiting its utility for measuring unlearning efficacy. **In contrast, our approach evaluates the MI differences for individual features and the label**, effectively avoiding this limitation, enabling a more precise evaluation.
> That said, we acknowledge that our current method for MI computation may not be optimal and could potentially be enhanced by incorporating alternative MI estimation techniques.
>
> > W3. Standard deviation of IDI is not low
>
> Thank you for the insightful question. IDI maintains a low standard deviation even when the number of forgettings is high since IDI captures overall differences in information, not individual data points. For instance, as shown in **Table 13 (bottom)**, when 20 classes are removed in CIFAR-100, the standard deviation remains modest, ranging from 0.003 to 0.011. This consistency reinforces IDI's robustness as an evaluation metric across various settings.
>
> > W4. Comparison of IDI with white-box MIA
>
> Thank you for the suggestion. We have included the comparison of IDI and white-box MIA in **Sec. 5.3, Line 512** of our revised manuscript. White-box MIA becomes unstable as the dataset scales, and inconsistent that MIA values of Retrain and randomly initialized model are similar.

---

> ### Author Response · Authors · 2024-11-20
> **comment (2/2)**
>
> > W5. For MIAs, only score-based MIA is employed, not LiRA MIA
>
> We use Entropy-based MIA (E-MIA) as the primary black-box MIA due to its widespread adoption in unlearning literature [1,2.3] and its greater variability across baseline algorithms, as detailed in **App. C.1**.  **However, we appreciate the reviewer's suggestion and have included additional results using LiRA MIA below, specifically for ResNet-18 on CIFAR-10, single-class forgetting.**
>
> |Method|Retrain|HD|FT|RL|GA|$l$1-sparse|SALUN|COLA|
> |-|-|-|-|-|-|-|-|-|
> |E-MIA|10.64±0.0|2.05±0.11|0.17±0.05|0.0±0.0|25.37±3.24|1.56±0.09|0.01±0.01|12.64±0.92|
> |Lira MIA|0.5|0.654|0.76|0.798|0.712|0.543|0.623|0.547|
>
> While LiRA is a comprehensive approach, its computational demands make it impractical for large-scale experiments, as it requires training over 10,000 additional models. Thus, we only include experiments where such constraints are manageable.
> Also, this result confirms that LiRA shares the same limitations as other black-box metrics, as HD achieves decent performance on these metrics while modifying only the last layer. This emphasizes the need for white-box metrics like IDI to provide a more reliable and comprehensive evaluation of unlearning quality.
>
> [1] Kurmanji et al., Towards unbounded machine unlearning, NeurIPS, 2023\
> [2] Chundawat et al., Can bad teaching induce forgetting? unlearning in deep networks using an incompetent teacher, AAAI, 2023\
> [3] Foster et al., Fast machine unlearning without retraining through selective synaptic dampening, AAAI, 2024
>
> > W6. Lack of literature on auditing machine unlearning
>
> We have added literatures on unlearning verification (auditing) and the difference between evaluation in **line 121** of the revised manuscript as suggested. Thank you!
>
> > Q1. Why is the sum of absolute values not used in Equations (1) and (2)?
>
> **It is to correctly preserve the amount of residual information change.** Retaining the numerator's sign allows IDI to capture phenomena like “over-unlearning” (discussed in **line 399** of the revised manuscript).
> An IDI of 0 indicates perfect alignment with Retrain, **while a negative IDI signals a reduction beyond Retrain.** For example, our method COLA achieves negative IDI in some setups by actively removing residual features. This design makes IDI an intuitive and effective tool for assessing unlearning efficacy, unlike other metrics that typically focus only on absolute differences.
>
> > Q2. Possibility of near-zero denominator in IDI
>
> **Yes, it is possible. It happens when the Original already closely resembles Retrain.** Our Information Difference ($\mathbf{ID}(\theta_o)$), the denominator of IDI, is designed to capture these subtle differences.\
> We empirically validated such extreme scenarios by selecting the most representative samples (the lowest-loss samples) from each class as a forget set in random data forgetting. Forget sets of 5, 10, 50 samples per class were used to simulate varying levels of omission. $\mathbf{ID}(\theta_o)$ was computed by substituting retrained models ($\theta_{5}, \theta_{10}, \theta_{50}$) into Equation (1).
>
> |Forget set|MI of $\theta_r$|MI of $\theta_o$|$\mathbf{ID}(\theta_o)$|
> |-|-|-|-|
> |similar 5|2.603|2.611|0.008|
> |similar 10|2.55|2.561|0.011|
> |similar 50|2.814|2.88|0.066|
> |random 500|1.965|2.297|0.332|
>
> As shown in the last column of the table, $\mathbf{ID}(\theta_o)$ values are near-zero for small forget sets, indicating minimal information loss compared to $\theta_{500}$. The increasing gap with larger forget sets aligns with information theory, showcasing our metric's robustness and interpretability. Notably, very small $\mathbf{ID}$, (e.g., for $\theta_{5}$), suggests negligible unlearning utility, indicating when unlearning may be unnecessary. These updates have been incorporated into **line 402** of the revised manuscript.
>
> > Q3. Computational time of IDI with last few layers
>
> ||ResNet-18||||ResNet-50||||
> |-|-|-|-|-|-|-|-|-|
> ||Time (min)|IDI (FT)|IDI (SCRUB)|IDI (COLA)|Time|IDI (FT)|IDI (SCRUB)|IDI (COLA)|
> |Full layers|31.41±0.59|0.670±0.011|-0.055±0.028|0.010±0.009|98.49±0.43|0.617±0.006|0.067±0.005|0.019±0.006|
> |Selected layers|11.94±0.25|0.671±0.008|-0.056±0.008|0.010±0.006|45.13±0.22|0.607±0.009|0.067±0.020|0.019±0.025|
>
> When computing mutual information using only the last few layers (two for ResNet-18 and three for ResNet-50), we observed that the process is 2.64 times faster for ResNet-18 and 2.18 times faster for ResNet-50 compared to analyzing all layers. **Importantly, the IDI values calculated using all layers and those calculated using only the last few layers remain nearly identical** (as shown in the above, which are IDI values on CIFAR-10 single class forgetting setup).This indicates that focusing on the last few layers with a large information gap as shown in **Fig. 5** captures the necessary information for evaluation without any significant loss in fidelity. **More results on other baselines are included in App. E.2.**

---

> > ### Comment · Reviewer_Nhgb · 2024-11-24
> > **Thank you for your responses.**
> >
> > __Thank you to the authors for providing further explanations and experiments, which address most of the concerns. I still have a few remaining points that I believe would be valuable to address:__
> >
> > > W2. Suggested simpler baseline for MI estimation
> >
> > Thank you for the additional experiments on the suggested simpler baseline. Since it is mentioned that "features from earlier layers (e.g., block 3) exhibit negligible differences," it would be interesting to evaluate the performance when training with only the last layer.
> >
> > >W5. For MIAs, only score-based MIA is employed, not LiRA MIA
> >
> > Thank you for including further experiments with LiRA MIA. It would be helpful to specify the experimental settings for LiRA, such as the data-splitting strategy and the number of surrogate models used.
> >
> > > Q1. Why is the sum of absolute values not used in Equations (1) and (2)?
> >
> > Please clarify the definition of over-unlearning. Additionally, it would be helpful to include a toy example that demonstrates the correlation between the over-unlearning effect and its corresponding IDI values.

---

> > > ### Author Response · Authors · 2024-11-24
> > > **Additional response by the authors to the reviewer Nhgb's response**
> > >
> > > > W2. It would be interesting to evaluate the performance when training with only the last layer.
> > >
> > > Thank you for your valuable suggestion. **We would like to clarify that your suggestion aligns with the case of the IDI value when using $n = 1$, as shown in the table below.** We appreciate your input and acknowledge its practical validity.
> > >
> > > ### ResNet-50 / IDI of different $n$ / CIFAR-10, single-class forgetting
> > > |Methods|Full Layers|$n=5$|$n=4$|$n=3^\star$|$n=2$|$\textbf{n=1}$|
> > > |-|-|-|-|-|-|-|
> > > |SCRUB|$0.067_{\pm0.005}$|$0.073_{\pm0.007}$|$0.071_{\pm0.007}$|$0.067_{\pm0.020}$|$0.076_{\pm0.008}$|$0.011_{\pm0.005}$|
> > > |SALUN|$0.831_{\pm0.014}$|$0.833_{\pm0.011}$|$0.832_{\pm0.019}$|$0.832_{\pm0.027}$|$0.842_{\pm0.009}$|$0.771_{\pm0.005}$|
> > > |$l$1-sparse|$0.185_{\pm0.005}$|$0.183_{\pm0.007}$|$0.181_{\pm0.007}$|$0.184_{\pm0.023}$|$0.185_{\pm0.016}$|$0.191_{\pm0.007}$|
> > > |COLA|$0.019_{\pm0.006}$|$0.023_{\pm0.009}$|$0.021_{\pm0.009}$|$0.019_{\pm0.025}$|$0.022_{\pm0.009}$|$0.007_{\pm0.011}$|
> > >
> > > The table displays IDI values calculated using different numbers of layers from the back of the model. The $\star$ column represents the values reported in our paper, while $n=1$ (in **bold**) corresponds to the reviewer's suggestion. We clarify that IDI calculation for $n=1$ employs MLP as the $f$ network.
> > >
> > > While focusing solely on the last intermediate layer is practically valid, we believe it does not fully capture the comprehensive evaluation of the encoder, as earlier layers also contribute significantly to the intermediate representation. Our choice of $n$ in the $\star$ column aims for an ideal case, closely approximating Full Layers, ensuring a more comprehensive assessment of intermediate representations.
> > >
> > > For further details on layer selection for all algorithms in ResNet-18 and ResNet-50, please refer to **App. E.2.** We appreciate the opportunity to clarify this point and thank you again for your valuable feedback.
> > >
> > > > W5. It would be helpful to specify the experimental settings for LiRA.
> > >
> > > Thank you for your question. For our experiments with LiRA MIA, we closely followed the methodology outlined in U-LiRA MIA [1]. Specifically, we trained 128 ResNet-18 models on random splits of half the CIFAR-10 training set. This ensures that, on average, each of the 50,000 samples in CIFAR-10 is included in the training set of 64 models and excluded from the training set of 64 models.
> > >
> > > For each of these 128 models, we apply the unlearning algorithm to 40 randomly selected forget sets (class 4, 200 samples) from the model’s training set, resulting in 128 × 40 = 5,120 models undergoing the unlearning procedure. To evaluate LiRA MIA, we took 2,560 shadow models. For each sample in class 4 of the CIFAR-10 training set, we identified 64 shadow models where the sample was included in the forget set, or excluded from training entirely. Similarly, we used 2,560 target models (distinct from the shadow models), testing on 200 samples from class 4 that are excluded from both the retain and forget sets (test sets) and forget sets.
> > >
> > > Further methodological details can be found in [1], as our setup is closely aligned with theirs. In response to the reviewer's suggestion, we have included additional details about LiRA MIA in **App. C.1**. We would like to highlight that the evaluation process for each method involves approximately 300-500 GPU hours. We hope this explanation clarifies our experimental setup, and we sincerely appreciate your thoughtful feedback.
> > >
> > > [1] Hayes et al. Inexact unlearning needs more careful evaluations to avoid a false sense of privacy. arXiv, 2024.
> > >
> > > > Q1. Clarify the definition of over-unlearning.
> > >
> > > The term "over-unlearning" refers to a scenario where the unlearning algorithm actively removes information about the forget set, resulting in an unlearned model that retains even less information than Retrain. This occurs when the unlearning process not only forgets the designated data but also eliminates related or shared information, leading to more information loss compared to retraining from scratch.
> > >
> > > To provide an intuitive example:
> > > - Consider a task in CIFAR-10 where we aim to unlearn the label "dog." In Retrain, there might still be some shared information about "dog" learned indirectly from "cat"-labeled images, as "dog" and "cat" share common features (e.g., fur texture, body shapes). However, an over-unlearning algorithm could actively destroy these shared features to the point where the unlearned model loses more information about "dog" than the retrained model would have, effectively "over-unlearning" the forget set.
> > >
> > > We can measure over-unlearning using IDI, where negative IDI values arise from a negative Information Difference (ID). This implies that the amount of information about the forget set is smaller in the unlearned model than in the Retrain.
> > > We hope this example clarifies the concept of over-unlearning and its relationship to IDI values. For further details, please refer to **Section 4.3, line 399.**

---

> > > > ### Comment · Reviewer_Nhgb · 2024-11-24
> > > > **Thank you for your responses.**
> > > >
> > > > Thank you to the authors for the clarifications. I have one additional concern:
> > > >
> > > > > Additionally, it would be helpful to include a toy example that demonstrates the correlation between the over-unlearning effect and its corresponding IDI values.
> > > >
> > > > Apologies for the ambiguity earlier. What I mean is, could you implement a numerical example that illustrates how an increase in the over-unlearning effect leads to a corresponding increase in the negative direction of the proposed IDI values?

---

> > > > > ### Author Response · Authors · 2024-11-25
> > > > > **Additional response by the authors to the reviewer Nhgb's response**
> > > > >
> > > > > > Implement an example which shows increase in over-unlearning effect leads to increase in (negative) IDI values
> > > > >
> > > > > We appreciate your suggestion and the opportunity to clarify the relationship between the over-unlearning effect and IDI values. Below, we provide the experimental results.
> > > > >
> > > > > **Consider the Setup:**
> > > > >
> > > > > * Suppose we aim to remove the “Dog” class in a single-class forgetting scenario. The corresponding Retrain (the gold standard model) is the model trained excluding only the “Dog” class. **To simulate “over-unlearning”, we create additional models that have fewer classes in the training set than Retrain by expanding the excluded classes, starting with (“Dog,” “Cat”), then (“Dog,” “Cat,” “Fox”), and so on.** These models retain significantly less information about “Dog” compared to Retrain, as they lack general information from other classes that aids in distinguishing “Dog” images. Consequently, these models are considered “over-unlearned” relative to Retrain. As the number of excluded classes increases, the over-unlearning effect becomes more pronounced, which is expected to be reflected in the IDI values. The most extreme case is the “random initialized” model, which has never been trained.
> > > > >
> > > > > Below are the IDI results for CIFAR-10 and CIFAR-100 using ResNet-18.
> > > > >
> > > > > ### ResNet 18, CIFAR-10
> > > > > |single class|2 classes|3 classes|5 classes|random|
> > > > > |-|-|-|-|-|
> > > > > 0.0|-0.482|-0.552|-1.042|-1.281|
> > > > >
> > > > > ### ResNet 18, CIFAR-100
> > > > > |single class|5 classes|20 classes|50 classes|random|
> > > > > |-|-|-|-|-|
> > > > > 0.0|-0.143|-0.309|-0.602|-2.955|
> > > > >
> > > > >
> > > > > The numbers in the columns represent the number of excluded classes. Moving from left to right, the over-unlearning effect intensifies, reflected by a gradual decrease in IDI values. The last column (randomly initialized model) exhibits the most negative IDI value. This demonstrates that IDI effectively captures the extent of over-unlearning in a clear and intuitive manner. If the reviewer finds the experimental results satisfactory, we will include them in the appendix for further reference. We sincerely appreciate your valuable feedback and hope this addresses your concern. Thank you for your thoughtful suggestion!

---

> > > > > > ### Comment · Reviewer_Nhgb · 2024-11-25
> > > > > > **Thank you for your responses**
> > > > > >
> > > > > > Thank you to the authors for providing additional clarifications, which have addressed my concerns. I have raised my score.

---

> > > > > > > ### Author Response · Authors · 2024-11-25
> > > > > > > **Thank you for thoughtful feedback and raising the score**
> > > > > > >
> > > > > > > Dear Reviewer Nhgb
> > > > > > >
> > > > > > > Thank you for your prompt reply and for acknowledging our efforts in addressing your queries. We are glad to know that our responses have met your expectations! If you have additional questions, feel free to reach out.
> > > > > > >
> > > > > > > Sincerely,\
> > > > > > > Authors

---

### Official Review · Reviewer_7jq1 · 2024-11-04

**Soundness:** 3
**Presentation:** 2
**Contribution:** 2
**Rating:** 6
**Confidence:** 4

**Summary:**

The paper address the issue that current black-box unlearning evaluation methods does not reflect the true unlearning quality -- many methods that pass such evaluation methods actually contains substantial information about the forget set, which make the unlearned model prone to relearning attacks. The paper proposes a new white-box evaluation metric based on mutual information. The estimated InfoNCE loss is used to calculate Information Difference Index (IDI) to ensure effective unlearning.

Based on this intuition the paper further proposes a new unlearning method by collapsing forgot set representation on the encoder layers then re-aligning the model. The paper demonstrated that such unlearning methods achieve low IDI compared to other unlearning methods.

**Strengths:**

Common unlearning metrics are known to be incomplete and not always reliable. This paper proposes a new evaluation methods for unlearning by estimation the mutual information using InfoNCE between intermediate layer features and output labels. This is a method that takes the internal behavior of the model into account, which measures unlearning at a deep level.

The paper also proposed a new unlearning methods from a similar motive. The CoLA method is able to achieve stronger unlearning compared to other unlearning methods.

**Weaknesses:**

One motivation for the new evaluation metric is from the concern that current metric only measures shallow unlearning which can be prone to relearning attacks, However, the paper does not present the strength of this new metric in a systematic manner. Figure 4 briefly mentioned retrain attacks but does not have the SCRUB method, it also comes before the new metric which makes it a bit confusing to read.

The evaluation method uses up to $\ell$-th layer and trains the rest to estimate the InfoNCE loss, which seems resource-intensive and scales poorly with the increase size of neural networks. It also requires a retrained model for IDI statistic, which seems infeasible for evaluating unlearning in practice.

**Questions:**

In Figure 5, why does the mutual information decrease with the layers? Won't the model get more information as the features are passed through the network?

---

> ### Author Response · Authors · 2024-11-20
>
> > W1. Figure 4 briefly mentioned relearn attacks but does not have the SCRUB method, it also comes before the new metric which makes it a bit confusing to read.
>
> We have included SCRUB into **Fig. 4** and updated the caption to clarify the purpose of our metric. IDI in Fig. 4 demonstrates its alignment with recovered accuracy. Its strength lies in its information-theoretic foundation to robustly measure unlearning in intermediate layers—an aspect often overlooked by black-box metrics. Empirically, IDI offers consistent insights across architectures, as demonstrated by t-SNE visualizations in **Fig. 15–17**. Additionally, IDI shows its robustness through the low standard deviation shown in **Tab. 1-3** and its practicality in large-scale models by utilizing only subsets to compute in **Fig. 8 and Tab. 8**.
>
> > W2-1. Computational burden of the evaluation method, IDI
>
> **Our evaluation method IDI is not resource-intensive in practice** as we only use the last $n$ selected layers (i.e., the last layers of later blocks of ViT) for computation, given that early layers exhibit negligible information differences between Original and Retrain, as shown in **Fig. 5**. This clarification has been added in **line 392 and App. E.2,  App. E.6** of the revised manuscript. Additionally, as shown in **Fig. 8**, IDI can be evaluated on dataset subsets, reducing computational time. For instance, using 10% of the data, IDI computation takes just 10 GPU minutes for ResNet-18 on CIFAR-100 and 90 GPU minutes for ViT—approximately four times faster than using the full dataset. The scalability of our metric to large models like ViT further highlights its practicality.\
> Moreover, developing a reliable evaluation metric for strong unlearning inherently involves complexity [1]. For instance, recent metrics like U-LiRA [2], which requires training an additional 10,240 models, and the NeurIPS 2023 Unlearning Challenge [3], prioritize reliability over efficiency.
>
> [1] Xu et al., Machine unlearning: A Survey, TETCI,  2024\
> [2] Hayes et al., Inexact unlearning needs more careful evaluations to avoid a false sense of privacy, arXiv, 2024\
> [3] Triantafillou et al., Are we making progress in unlearning? Findings from the first NeurIPS unlearning competition, arXiv, 2024
>
> > W2-2. The need of retrained model
>
> In real-world unlearning "verification" studies, yes, it often assumes the absence of Retrain [1]. However, we focus on "evaluation" studies, which assess whether unlearning algorithms effectively remove specific information from the model [2]. Since the goal of strong unlearning is to align the unlearned model's behavior with Retrain, evaluation typically relies on Retrain as the gold standard. **Consequently, existing evaluation metrics commonly depend on Retrain, as detailed in Tab. 6 of [2]**. We have clarified this distinction on **line 120** of the revised manuscript.
>
> **Nonetheless, IDI is adaptable to scenarios where Retrain is unavailable by leveraging an alternative reference model, such as a baseline unlearned model.** While this changes the interpretation of IDI, it continues to provide valuable insights into residual information within unlearned models. This flexibility is discussed in **Sec. 5.3** of our manuscript, emphasizing IDI's practicality in real-world applications.
>
> [1] Zhang et al.,Verification of machine unlearning is fragile, ICML, 2024\
> [2] Nguyen et al., A survey of machine unlearning, arXiv, 2022
>
> > Q1. The decrease of the mutual information with the layers
>
> It may seem counterintuitive that a model’s features lose information as they propagate through layers, but this aligns with the Information Bottleneck principle [1,2]. Intuitively, as input data $X$ passes through the layers of a neural network, it is compressed and transformed, reducing the original information about $X$, with the final layer retaining the least.
>
> The Data Processing Inequality (DPI) [1, 2] explains this phenomenon theoretically. In a neural network, the transformations across layers can be modeled as a Markov chain:\
> $Y\to X \to Z_1 \to \dots \to Z_l \to \hat{Y}$\
> where $X$ is the input, $Z_i$ is the intermediate representation at the \( i \)-th layer, and $Y$, $\hat{Y}$ denote the true label and model output, respectively.
> DPI ensures that mutual information decreases through successive layers:\
> $I(X; Y) \geq I(Z_1; Y) \geq I(Z_2; Y) \geq \dots \geq I(Z_l; Y).$\
> As shown in **Fig. 5**, this theoretical insight aligns with the observed reduction in mutual information across the network's layers.
>
> [1] Tishby et al., Deep learning and the information bottleneck principle, IEEE ITW, 2015\
> [2] Goldfeld et al., The information bottleneck problem and its applications in machine learning, IEEE JSAIT, 2020

---

> > ### Comment · Reviewer_7jq1 · 2024-11-27
> >
> > Thank you for your detailed explanations. These mostly clarified my confusions. I do agree that I focus more on the verification side of the MU, under which I think current black-box methods are not really reliant on retrained models. (In addition, in the whole-class removal setting as mentioned in paper, we can simply assume retrained acc to be 0). Personally I think about machine unlearning as more of a practical problem so I think a bit more about how can this be applied to verify MU.
> >
> > I think in general the paper is sound and offers good theoretical insights of how machine unlearning algorithms behave. But I do think that the white box evaluation scenario is not always applicable in real-world cases or with large generative models. Also, apart from removing the whole class, removing a few data entries should not collapsing the representation intuitively. I tend to keep my score.

---

> ### Author Response · Authors · 2024-11-28
> **Additional response by the authors to the reviewer 7jq1's response**
>
> Thank you for your detailed feedback and for clarifying your perspective. We are pleased that our explanations addressed many of your concerns and that our work aligns with your emphasis on verification aspects in machine unlearning (MU). We also appreciate your acknowledgment of the theoretical insights provided by the paper.
> We agree that black-box evaluation metrics have significant practical value, particularly in real-world applications where white-box access is often limited. However, as noted by Zhang et al.[1], black-box verification methods face challenges in reliably detecting residual information, which can undermine the foundational principle of unlearning--the "right to be forgotten." These challenges are further compounded in scenarios involving large-scale or proprietary models, where transparency becomes critical.
>
> Our work aims to contribute to this conversation by providing a flexible white-box evaluation framework. By focusing on interpretable metrics, such as mutual information differences and evaluations across intermediate layers, we hope to bridge the gap between theoretical rigor and practical utility. While we recognize that white-box methods may not be universally applicable, we believe they can complement existing black-box approaches by providing deeper insights into model behavior and unlearning effectiveness.
>
> We value your suggestion to consider the broader practical implications of MU methods and agree that scenarios involving incremental data removal warrant further investigation. Thank you again for your constructive feedback, which has been instrumental in refining our contributions. We hope this discussion inspires further advancements in machine unlearning evaluation.
>
> [1] Zhang, Binchi, et al. "Verification of machine unlearning is fragile.", ICML, 2024

---

### Official Review · Reviewer_iQyZ · 2024-11-04

**Soundness:** 3
**Presentation:** 3
**Contribution:** 3
**Rating:** 6
**Confidence:** 3

**Summary:**

This paper addresses a crucial and well-motivated issue in recent machine unlearning research: the inadequacy of common black-box metrics for assessing strong unlearning. To tackle this problem, the authors propose a new white-box metric called the Information Difference Index (IDI), which quantifies the residual information of the forget set in the intermediate layers of the network, providing a more comprehensive evaluation of unlearning efficacy.

**Strengths:**

S1. The assertion that "black-box metrics may not be sufficient for assessing strong unlearning" is well-motivated and compelling.

S2. The paper includes extensive empirical evaluations that support the claims made by the authors.

S3. The technical quality of the paper is good, and the ideas presented may be of significant interest to the machine learning community.

**Weaknesses:**

W1. This study seems to focus predominantly on the single-class forgetting scenario in the main text, relegating results related to other cases (e.g., random data forgetting, multi-class forgetting) to the appendices. Given that the paper aims to provide a general approach and is not limited to single-class forgetting, the authors are strongly encouraged to include essential findings from all three tasks in the main text. For instance, empirical evaluations demonstrating whether the proposed method consistently outperforms other baselines across all scenarios should be highlighted.

W2. The head distillation (HD) approach introduced in Section 3 seems to be applicable only to the single-class forgetting task. The authors should either demonstrate HD's applicability to multi-class and random data forgetting tasks, or explicitly state its limitations if it cannot be generalized.

If the generalization is possible, the authors should address whether the main observations regarding residual information and the limitations of black-box metrics are applicable in these other contexts. Specifically:
+ Would residual information still be retained within the intermediate layers of unlearned models?
+ Would black-box metrics continue to fail in capturing this residual information?

W3. I agree that a primary limitation of the proposed COLA approach is its computational burden due to the mutual information (MI) estimation process for each network layer, although this point is also noted by the authors in the paper.

**Questions:**

In addition to addressing W1 and W2, the authors are also expected to answer the following questions:

Q1. In Figure 7, COLA appears to obtain a negative IDI value, which is even lower than Retrain. Does this imply that COLA is "over-unlearned”, according to the statement in Line 410?

Q2. Intuitively, for the random data forgetting task, if the retained dataset contains many samples similar to a forgotten one, residual information will likely persist within the intermediate layers. Given that the value of the Information Difference Index (IDI) heavily relies on the amount of this residual information, the authors should provide empirical evidence or theoretical analysis to demonstrate the effectiveness (or potential limitations) of IDI in random data forgetting tasks, particularly when there is a high similarity between forgotten and retained samples. This would help clarify the robustness and generalizability of the proposed metric.

---

> ### Author Response · Authors · 2024-11-20
>
> > W1. Include essential findings from all three tasks in the main text
>
> Thank you for the thoughtful recommendation. We have included results from the three tasks. **Tab. 1** presents the experiment results for single-class and multi-class forgetting. In response to the reviewer’s suggestion, **we have included random data forgetting in Tab. 2, along with a detailed explanation in Sec. 5.2 of the revised manuscript.**
>
> > W2. HD's applicability to multi-class and random data forgetting tasks
>
> HD is well-suited for multi-class forgetting and can also be adapted for random data forgetting with appropriate modifications. In random data forgetting, where all classes are part of the retain set, HD’s logit masking technique cannot be directly applied. Instead, we perform gradient descent on the retain set and gradient ascent on the forget set, while **modifying only the last layer**. This aligns with HD’s emphasis on addressing the limitations of black-box metrics. Below, we present results for both scenarios.
>
> - Model: ResNet-18
> |Methods|CIFAR-10 5 classes||||CIFAR-100 20 classes|||| CIFAR-10 500 samples||||
> |-|-|-|-|-|-|-|-|-|-|-|-|-|
> ||UA|TA|MIA|RTE|UA|TA|MIA|RTE|UA|TA|MIA|RTE|
> |Retrain|100.0|78.45|7.12|165.92|100.0|80.01|7.55|139.93|3.94|95.26|75.12|152.87|
> |HD|100.0±0.0|77.79±0.03|0.50±0.03|0.52±0.09|100.0±0.0|79.54±0.01|0.50±0.01|0.43±0.02|3.64±2.64|92.80±2.18|77.47±8.09|0.30±0.01|
> |FT|100.0±0.0|77.43±0.20|0.20±0.06|9.21±0.02|99.81±0.04|79.11±0.35|0.22±0.05|7.43±0.07|5.03±0.40|92.94±0.26|83.52±0.58|8.11±0.03|
> |RL|98.61±0.22|77.78±0.19|0.0±0.0|3.38±0.09|95.77±0.09|78.42±0.05|0.0±0.0|2.94±0.01|4.77±0.27|93.54±0.04|22.47±1.19|2.75±0.01|
> |$l$1-sparse|98.63±0.37|73.46±0.25|12.35±0.82|4.19±0.01|83.49±0.46|76.79±0.20|6.36±0.59|3.08±0.07|5.47±0.22|91.31±0.25|77.12±0.21|3.03±0.04|
> |SALUN|100.0±0.0|77.18±0.14|0.13±0.09|4.45±0.41|90.69±0.76|74.72±0.54|0.17±0.03|3.85±0.0|2.74±0.30|91.68±0.44|83.52±2.20|5.69±0.04|
>
> HD delivered strong results across all scenarios with significantly faster speeds. This reinforces our argument that black-box metrics consistently fail to detect influences of the forget set remaining in the middle layers. **HD’s performance is added in Tab. 1 and 2, with more results included in App. D.1  of the revised manuscript.**
>
> > W3. Computational burden of COLA and IDI
>
> **COLA may have a higher runtime than some baselines, but its efficiency improves on larger datasets** as the feature collapsing accelerates with more classes. For example, in the ImageNet-1K - ResNet-50 experiment, COLA takes 172 minutes compared to SCRUB and SALUN, which require 426 and 794 minutes, respectively (Tab. 14, RTE column). While **COLA does not involve mutual information estimation, our IDI metric does**. We note that **IDI is not computationally heavy in practice.**  We compute IDI using only the last $n$ selected layers (i.e., later blocks of ViT), as earlier layers show negligible information differences between Original and Retrain (**Fig. 5**). This clarification is included in **line 392 and App.E.2, App. E.6** of the revised manuscript. Additionally, as shown in **Fig. 8**, IDI can be evaluated on data subsets, further reducing computation.
>
> > Q1. Over-unlearning in COLA
>
> Yes, this shows that COLA is “over-unlearned”. Its active feature elimination can sometimes reduce residual information to levels lower than those of Retrain while maintaining high accuracy, aligning with its intended goal. This highlights IDI's strength in capturing the extent of unlearning in intermediate layers—a key attribute for a robust unlearning metric.
>
> > Q2. Effectiveness of IDI/ID on highly similar forget/retain set in random forgetting
>
> When the forget set resembles the retain set, the residual information between Original and Retrain models becomes nearly identical. Our Information Difference ($\mathbf{ID}(\theta_o)$), the denominator of IDI, captures these subtle differences.
>
> To validate this, we ran experiments on (CIFAR-10, ResNet-18) using forget sets of 5, 10, and 50 lowest-loss samples per class (to simulate similar forget-retain set), calculating $\mathbf{ID}(\theta_o)$ for different retrained models ($\theta_{5}, \theta_{10}, \theta_{50}$).
> |Forget set|MI of $\theta_r$|MI of $\theta_o$|$\mathbf{ID}(\theta_o)$|
> |-|-|-|-|
> |similar 5|2.603|2.611|0.008|
> |similar 10|2.55|2.561|0.011|
> |similar 50|2.814|2.88|0.066|
> |random 500|1.965|2.297|0.332|
>
> As expected, $\mathbf{ID}(\theta_o)$ values are near-zero for small forget sets, indicating minimal information loss compared to $\theta_{500}$, the retrained model on 500 random forget samples per class. The increasing gap with larger forget sets aligns with information theory, demonstrating our metric's robustness and interpretability. Notably, very small $\mathbf{ID}$ (e.g., for $\theta_{5}$) suggests negligible unlearning utility, highlighting when unlearning may be unnecessary. These clarifications are included in **line 402** of the revised manuscript. Thank you for your feedback!

---

> ### Author Response · Authors · 2024-11-28
> **Gentle Reminder for Reviewer iQyZ**
>
> We sincerely appreciate the time and effort you have dedicated to reviewing our submission. We want to kindly remind you that we have carefully addressed your concerns through our responses and manuscript revisions. As the discussion period is nearing its conclusion, we would be grateful to receive any additional feedback or concerns you may have. Your insights are invaluable to us. Thank you again for your time and support.

---

> ### Author Response · Authors · 2024-12-02
> **Gentle Reminder for Reviewer iQyZ**
>
> We sincerely thank you for thoroughly reviewing our manuscript and for your positive assessment of our contributions. As the discussion period approaches its conclusion, we kindly remind the reviewer that we welcome any additional feedback. We have carefully revised the manuscript based on your comments and would be happy to address any further suggestions you may have.

---

### Author Response · Authors · 2024-11-20
**General Response by Authors**

We sincerely thank the reviewers for their thoughtful feedback, detailed evaluations, and encouraging comments, including that the motivation to address the limitations of black-box metrics was well-founded and significant (**iQyZ, 7jq1, Nhgb**), despite the challenges in proposing reliable unlearning metrics (**7jq1, Nhgb**). We also appreciate the recognition of new evaluation metric (**7jq1, Nhgb, 33Yi**) and new unlearning method, which builds on this new metric to achieve strong unlearning outcomes (**7jq1, 33Yi, Nhgb**), as well as the clarity of presentation (**33Yi**), the technical soundness (**iQyZ, Nhgb**), and the significant potential of our work to advance the unlearning community (**iQyZ**).

We have uploaded the first revision of the manuscript (changes are marked by blue color). Note that correlation of IDI with t-SNE has been moved from Sec. 5.3 to App. E.4 to accommodate the additional information added in the revision.

---

### Author Response · Authors · 2024-11-24
**Second revision**

We have uploaded the second revision of the manuscript. The revision includes our experimental details for LiRA MIA [1], addressing the reviewer’s suggestion (**Nhgb**).

Summary of changes
* Add experimental details for LiRA MIA in App. C.1

[1] Hayes et al., Inexact unlearning needs more careful evaluations to avoid a false sense of privacy, arXiv, 2024

---

### Meta-Review · Area_Chair_7jad · 2024-12-05

**Metareview:**

This work proposes a new metric to evaluate performance of machine unlearning algorithms. The proposed approach is model dependent, namely needs to access internal states of the underlying machine learning algorithms. A rich set of experiments are included to demonstrate that the new metric can characterize the level of unlearning.

Reviewers agree that this work considers an important problem and appreciate the extensive empirical study.

However, reviewers also have concerns that the computational cost is high and needs to access the retrained model, and thus it is unclear how feasible it is to deploy such method to large-scale problems in practice.

**Additional Comments On Reviewer Discussion:**

Reviewers requested additional experiments ranging from existing metrics to unlearning algorithms. Authors provided some quick results and have included in the revised manuscript. Reviewers appreciate that the empirical study is comprehensive, but have reservations how practical this method is.

---

### Decision · Program_Chairs · 2025-01-22

Reject